# Natural Gradient VI: Guarantees for Non-Conjugate Models

**Fangyuan Sun**[1,*]    **Ilyas Fatkhullin**[1,2,3*]    **Niao He**[1]

[1]Department of Computer Science, ETH Zurich
[2]ETH AI Center
[3]Industrial and Systems Engineering, Georgia Institute of Technology

fansun@student.ethz.ch,  ilyas.fatkhullin@ai.ethz.ch,  niao.he@inf.ethz.ch

## Abstract

Stochastic Natural Gradient Variational Inference (NGVI) is a widely used method for approximating posterior distribution in probabilistic models. Despite its empirical success and foundational role in variational inference, its theoretical underpinnings remain limited, particularly in the case of non-conjugate likelihoods. While NGVI has been shown to be a special instance of Stochastic Mirror Descent, and recent work has provided convergence guarantees using relative smoothness and strong convexity for conjugate models, these results do not extend to the non-conjugate setting, where the variational loss becomes non-convex and harder to analyze. In this work, we focus on mean-field parameterization and advance the theoretical understanding of NGVI in three key directions. First, we derive sufficient conditions under which the variational loss satisfies relative smoothness with respect to a suitable mirror map. Second, leveraging this structure, we propose a modified NGVI algorithm incorporating non-Euclidean projections and prove its global non-asymptotic convergence to a stationary point. Finally, under additional structural assumptions about the likelihood, we uncover hidden convexity properties of the variational loss and establish fast global convergence of NGVI to a global optimum. These results provide new insights into the geometry and convergence behavior of NGVI in challenging inference settings.

## 1 Introduction

Variational Inference (VI) is a powerful framework for approximating Bayesian posteriors by casting inference as an optimization problem [JGJS99, BKM17]. Unlike sampling-based approaches such as Markov chain Monte Carlo (MCMC; [MRR+53]) VI enables scalable posterior approximation through stochastic optimization, often with significant computational advantages. In practice, however, the exact gradient of the variational objective—such as the evidence lower bound (ELBO)—is rarely available in closed form, especially for complex or non-conjugate models. Consequently, gradients must be estimated from data using Monte Carlo sampling. Since the advent of Black-Box VI [WW13, TLG14, RGB14], this form of gradient-based stochastic optimization in Euclidean parameter spaces has become the de facto standard in practical applications.

While gradient descent in the Euclidean space of parameters has become the standard tool in modern VI, it ignores the intrinsic geometry of the space of probability distributions. This has motivated the development of *Natural Gradient Variational Inference (NGVI)* [HTRK07], which replaces standard gradients with natural gradients [Ama98] that account for the underlying information geometry of the variational family. NGVI follows the steepest descent direction (on average) with respect to the Kullback-Leibler (KL) divergence, rather than the Euclidean norm. Empirical evidence

---

*F.S. is supported by ETH AI Center, I.F. is funded by ETH AI Center Doctoral Fellowship.

39th Conference on Neural Information Processing Systems (NeurIPS 2025).

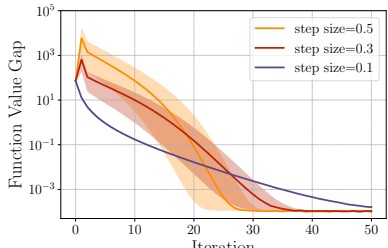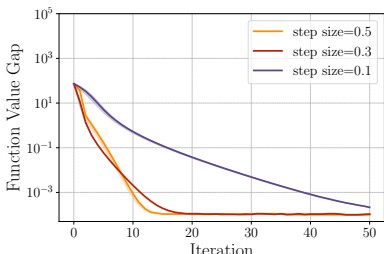

Figure 1: Convergence of SNGD on Poisson regression with different initialization. The function value gap refers to the difference between the objective value at iteration $t$ and the optimal value; performance is averaged over 10 runs for each algorithm. In the *left panel*, SNGD is initialized with $\sigma_0^2 = 2$. We observe instability at the initial iteration and slow convergence with step sizes 0.5 and 0.3. In contrast, with a different initialization $\sigma_0^2 = 0.4$ shown in the *right panel*, we observe stable behavior and fast convergence for larger step sizes.

suggests that this geometry-aware approach leads to faster or more stable convergence in various applications, including Bayesian neural networks [ZSDG18, OSK$^+$19] and probabilistic filtering [SEH18, LZM$^+$25]. However, theoretical understanding of NGVI remains limited—particularly in the case of *non-conjugate models*, where the posterior is not contained in the variational family and the optimization landscape becomes non-convex.

Recent work [WG24] established the first non-asymptotic convergence guarantees for stochastic natural gradient descent (SNGD), a basic variant of NGVI, under conjugate likelihoods, by framing SNGD as a special case of stochastic mirror descent (SMD; [NY83]). Their analysis hinges on the strong convexity and smoothness of the variational objective in the Bregman geometry induced by KL divergence. Unfortunately, these guarantees fail to extend to non-conjugate models such as logistic or Poisson regression, where the objective becomes non-convex and the geometry is not well-behaved. For instance, Figure 1 indicates that 1-smoothness of the objective as in conjugate models does not hold globally (see Section B for more detailed explanation). This observation serves as a motivation for our subsequent analysis of relative smoothness in non-conjugate models.

**Contributions.** This work develops a theoretical foundation for stochastic natural gradient variational inference (NGVI) in the *non-conjugate* setting. Our main contributions are as follows:

• **Relative Smoothness of the Variational Objective.** We derive sufficient conditions under which the negative ELBO is smooth relative to the Bregman geometry induced by the KL divergence on a compact set of parameters. Our results apply to a broad class of non-conjugate models in mean-field parameterization, including logistic regression, and provide explicit smoothness constants (polynomial in the problem dimension) over arbitrary compact subsets of the parameter space.

• **Projected Stochastic NGVI Algorithm.** Based on the relative smoothness analysis, we introduce a projected variant of NGVI called Proj-SNGD that enforces updates within a compact parameter domain using non-Euclidean projections. This approach leverages the dual structure of exponential families and admits efficient implementation in the mean-field Gaussian case. The resulting algorithm improves numerical stability while preserving the geometric fidelity of the natural gradient updates.

• **Convergence Guarantees via Mirror Descent.** We analyze Proj-SNGD through the lens of stochastic mirror descent (SMD), extending recent non-convex convergence theory [FH24]. We establish non-asymptotic convergence to a stationary point at rate $\mathcal{O}(1/\sqrt{T})$, and further show that under concavity of the log-likelihood, a hidden convexity structure [FHH25] allows us to prove global convergence at rate $\mathcal{O}(1/T)$.

By extending convergence guarantees to the non-conjugate regime, this work addresses a fundamental challenge in the theory of NGVI and provides a principled framework for variational inference in complex, non-linear Bayesian models.

## 1.1 Related Work

In the Euclidean setup, A large body of prior work provides convergence guarantees of standard VI algorithms. Building on smoothness and convexity properties established in [TLG14, Dom20], authors of [DGG23] propose two gradient descent-based algorithms that achieve convergence rates of $\mathcal{O}(1/\sqrt{T})$ using convexity, and $\mathcal{O}(1/T)$ using strong convexity. Concurrently, Kim et al. [KOW$^+$23] analyze the convergence behavior of gradient-based VI under various parameterization and prove similar convergence bounds. Furthermore, a linear rate in the case of conjugate likelihoods is established in [KMG24]. In the non-Euclidean setup, Wu and Gardner [WG24] establish the first $\mathcal{O}(1/T)$ convergence rate of NGVI assuming the model is conjugate. For general overview on natural gradient descent, we refer to the survey by Martens [Mar20].

Since its introduction in [Ama98], natural gradient descent has been extensively studied in the context of variational inference [HRL12, HBWP13, LT21], particularly for non-conjugate models [KBL$^+$16, KL17, SEH18, TR19]. Notably, [LKS19] and [ADY$^+$23] extend natural gradient methods to mixtures of exponential-family variational distributions, while [CAAK19] and [JCM24] focus on their application in online learning scenarios. In reinforcement learning, a series of works has focused on natural policy gradient [Kak01, AKLM21, Xia22, SLZ$^+$23]. Natural gradient variational inference (NGVI) has also been applied across various domains, including the training of variational Bayesian neural networks [ZSDG18, OSK$^+$19], Kalman filtering [LZM$^+$25], and multimodal optimization [MAM$^+$25].

There exists an extensive body of work on the convergence of stochastic mirror descent (SMD) under convexity assumptions [NJLS09]. Later, [Lan12, AZO17] show that mirror descent can be accelerated, similar to Nesterov's method. *Relative smoothness*, which was introduced concurrently in [LFN18, BBT17], has become an important tool in analysis of SMD [HR21, VYB$^+$22]. In the non-convex setting, [GLZ16] derive an $\mathcal{O}(1/\sqrt{T})$ convergence rate to a stationary point, albeit under very strong assumptions and with large mini-batches. This requirement is relaxed by [ZH18] and [DDM18], who establish similar convergence guarantees without relying on mini-batching. More recently, Fatkhullin and He [FH24] further relax the smoothness assumptions on the distance generating function using a distinct proof technique and improved convergence criterion. Non-convex SMD has also been studied in the context of reinforcement learning, where the value function is typically highly non-convex but admits certain structures analogous to our problem, see, e.g., [Lan23]. Additionally, alternative variance assumptions for SMD are explored by [Hen24].

## 2 Background

**Notation.** We write $\|\cdot\|$ for the Euclidean norm of a vector or the operator norm of a matrix, and $\langle\cdot,\cdot\rangle$ to denote the standard inner product in Euclidean space. For vectors $a, b \in \mathbb{R}^d$, let $a \odot b$ denote their entrywise (Hadamard) product. Let $\mathcal{S}_+^d$ denote the cone of $d \times d$ symmetric positive definite matrices. For symmetric matrices $A, B \in \mathbb{R}^{d \times d}$, we write $A \succeq B$ if $A - B \in \mathcal{S}_+^d$. For distributions $p$ and $q$, we write $D_{\mathrm{KL}}(q \,\|\, p) = \mathbb{E}_{q(z)} \log(q(z)/p(z))$ for the Kullback–Leibler divergence from $p$ to $q$. When needed, we write $q(z; \theta)$ to emphasize the parameterization of the distribution.

### 2.1 Variational Inference and Exponential Families

Let $z \in \mathbb{R}^d$ be a latent variable and $\mathcal{D} = \{(x_i, y_i)\}_{i=1}^n$ be the observed dataset. Given a prior $p(z)$ and likelihood model $p(\mathcal{D} \,|\, z)$, the goal of Bayesian inference is to compute the posterior $p(z \,|\, \mathcal{D}) \propto p(z)p(\mathcal{D} \,|\, z)$. Since this posterior is generally intractable, variational inference (VI) approximates it by minimizing the KL divergence from $p(z \,|\, \mathcal{D})$ to a tractable distribution $q(z)$ within a family $\mathcal{Q}$:

$$\min_{q \in \mathcal{Q}} D_{\mathrm{KL}}(q(z) \,\|\, p(z \,|\, \mathcal{D})).$$

This is equivalent to minimizing the negative evidence lower bound (negative ELBO) [BN06]:

$$\ell(q) = -\mathbb{E}_{q(z)}[\log p(\mathcal{D} \,|\, z)] + D_{\mathrm{KL}}(q(z) \,\|\, p(z)). \tag{1}$$

In this work, we assume $\mathcal{Q}$ is a minimal exponential family [WJ$^+$08], i.e., each $q \in \mathcal{Q}$ has the form

$$q(z; \eta) = h(z) \exp\left(\langle \phi(z), \eta \rangle - A(\eta)\right),$$

where $\eta$ is the natural parameter, $\phi(z)$ is the sufficient statistic, and $A(\eta)$ is the log-partition function. The dual expectation parameter is defined as $\omega := \mathbb{E}_{q(z;\eta)}[\phi(z)]$.

Let $\mathcal{D}_A := \{\eta \in \mathbb{R}^d : A(\eta) < \infty\}$ be the natural parameter domain, and let $\Omega$ be the set of corresponding expectation parameters. The gradient map $\nabla A : \mathcal{D}_A \to \Omega$ is bijective, with inverse $\nabla A^*$, where $A^*(\omega) := \sup_{\eta \in \mathcal{D}_A} (\langle \eta, \omega \rangle - A(\eta))$ is the convex conjugate of the log-partition function. This duality ensures that for any distribution $q \in \mathcal{Q}$, parameterized by either $\eta$ or $\omega$, we have

$$\nabla A(\eta) = \omega, \qquad \nabla A^*(\omega) = \eta.$$

For a comprehensive overview of exponential families and duality, we refer readers to [WJ+08].

**Remark.** With a slight abuse of notation, we denote the negative ELBO as $\ell(\eta) := \ell(q(z;\eta))$ or $\ell(\omega) := \ell(q(z;\eta))$ when working in either parameterization.

**Example: Gaussian Family.** For the multivariate Gaussian family $q = \mathcal{N}(\mu, \Sigma)$, the natural parameter is $\eta = (\lambda, \Lambda)$ with $\lambda = \Sigma^{-1}\mu$ and $\Lambda = -\frac{1}{2}\Sigma^{-1}$, and the expectation parameter is $\omega = (\xi, \Xi)$ with $\xi = \mu$ and $\Xi = \Sigma + \mu\mu^\top$. The standard parameterization $(\mu, \Sigma)$ and Cholesky form $(\mu, C)$ ($C$ is the Cholesky factor of $\Sigma$) are often used in practice.

## 2.2 NGVI as Stochastic Mirror Descent

Stochastic mirror descent (SMD) generalizes standard SGD to non-Euclidean geometries by replacing the squared Euclidean distance with a Bregman divergence.

**Definition 2.1** (Bregman Divergence). *Given a strictly convex and differentiable function $\Phi : \mathcal{U} \to \mathbb{R}$, the Bregman divergence between $u, v \in \mathcal{U}$ is $D_\Phi(u, v) := \Phi(u) - \Phi(v) - \langle \nabla\Phi(v), u - v \rangle$.*

The update rule for SMD on a differentiable objective $\ell : \mathcal{U} \to \mathbb{R}$ is in

$$\text{primal form:} \quad u_{t+1} = \underset{u \in \mathcal{U}}{\arg\min} \, \gamma_t \langle \hat{\nabla}\ell(u_t), u \rangle + D_\Phi(u, u_t), \qquad \text{or equivalently in} \quad (2)$$

$$\text{dual form:} \quad \nabla\Phi(v_{t+1}) = \nabla\Phi(u_t) - \gamma_t \hat{\nabla}\ell(u_t), \quad u_{t+1} = \underset{u \in \mathcal{U}}{\arg\min} \, D_\Phi(u, v_{t+1}), \quad (3)$$

where $\hat{\nabla}\ell(u_t)$ is a stochastic gradient and $\gamma_t$ is a step size.

In variational inference with exponential families, the natural parameter $\eta$ admits a geometry governed by the Fisher information matrix $\mathcal{I}(\eta) = \nabla^2 A(\eta)$, where $A(\eta)$ is the log-partition function. Stochastic NGVI in this space preconditions the gradient by the inverse Fisher matrix [RM15]. Applied to the variational objective $\ell$, the update becomes:

$$\eta_{t+1} = \eta_t - \gamma_t \mathcal{I}(\eta_t)^{-1} \nabla\hat{\ell}(\eta_t) = \eta_t - \gamma_t \hat{\nabla}\ell(\omega_t), \tag{4}$$

where $\omega_t = \nabla A(\eta_t)$ is the corresponding expectation parameter, see Section C.2 for details. Using the duality between $\eta$ and $\omega$, and the identity $\eta = \nabla A^*(\omega)$, we can re-write the update in the expectation parameter space as

$$\text{SNGD:} \qquad \nabla A^*(\omega_{t+1}) = \nabla A^*(\omega_t) - \gamma_t \hat{\nabla}\ell(\omega_t).$$

This matches the update rule of stochastic mirror descent with mirror map $A^*$. Hence, NGVI can be viewed as mirror descent over the expectation parameter $\omega$, with geometry induced by KL divergence. Indeed, for exponential families, the Bregman divergence associated with $A^*$ coincides with KL divergence: $D_{A^*}(\omega, \omega') = D_{\mathrm{KL}}(q(z;\omega) \| q(z;\omega'))$, see e.g., [WG24].

## 2.3 Mean-field Parameterization

In the following, we assume that the prior is standard Gaussian, $p(z) = \mathcal{N}(0, I)$, and the variational family $\mathcal{Q}$ is the mean-field Gaussian family, with mean $\mu = (\mu_i)_{1 \le i \le d}$ and diagonal covariance matrix $\Sigma = \mathrm{diag}((\sigma_i)_{1 \le i \le d})$. The corresponding expectation parameter $\omega = (\xi, \Xi)$ is given by $\xi = \mu$ and $\Xi = \Sigma + \mathrm{diag}(\mu \odot \mu)$, defined over the domain

$$\Omega = \left\{ (\xi, \Xi) : \xi \in \mathbb{R}^d, \ \Xi - \mathrm{diag}(\xi \odot \xi) \in \mathcal{S}_+^d \text{ and is diagonal} \right\}.$$

This parameterization is widely adopted due to its tractability and interpretability, allowing efficient coordinate-wise updates while still capturing key aspects of the posterior distribution. Moreover, it often provides a favorable trade-off between computational complexity and approximation quality in high-dimensional settings. It has been extensively studied in recent theoretical works, including applications to Bayesian deep neural networks [CE24], high-dimensional Bayesian linear models [CFLM23], and particle-based variational inference [DWZZ24].

## 3 Landscape Properties of Non-Convex NGVI

In this section, we investigate the landscape properties of variational objective $\ell(\omega)$ with a particular focus on the properties useful for establishing non-asymptotic convergence of NGVI. First, we note that $\ell(\omega)$ is coercive in the expectation parameter, i.e., it grows to infinity $\ell(\omega) \to \infty$ when the parameters approach the boundary of the set $\omega \to \partial\Omega$, see Section D for more details. This is a useful property which guarantees the existence of the minima of $\ell(\omega)$. Second, it is known that $\ell(\omega)$ is non-convex w.r.t. $\omega$ in the general non-conjugate likelihood setting, i.e., when likelihood does not belong to the variational family $\mathcal{Q}$, see Section E for details. This non-convexity is the main challenge for establishing non-asymptotic convergence of NGVI. In what follows we aim to use modern tools from non-convex optimization, which will allow us to provide a better understanding of NGVI landscape and characterize its convergence. The rest of the section is organized as follows. In Section 3.1, we will prove that, despite non-convexity, both the lower and upper curvature of $\ell(\omega)$ are bounded in the non-Euclidean geometry following the formalism of relative weak convexity and relative smoothness [LFN18]. In Section 3.2, we uncover the hidden convexity properties of the objective and connect them with Polyak-Lojasiewicz (PL) condition [Pol63, Loj63], which allows us to show fast convergence of NGVI despite non-convexity.

### 3.1 Relative Smoothness of Variational Objective

We start with the definition of $\alpha$-$\beta$ relative smoothness [BBT17, LFN18], which is a generalization of smoothness and weak convexity to Bregman geometry,

**Definition 3.1.** *Let $\Phi : \mathcal{U} \to \mathbb{R}$ be a differentiable and strictly convex function on a convex set $\mathcal{U}$. A differentiable function $\ell$ is said to be $\alpha$-$\beta$ smooth relative to $\Phi$ for some $\alpha \in \mathbb{R}, \beta > 0$ if*

$$\alpha D_\Phi(v, u) \leq \ell(v) - \ell(u) - \langle \nabla \ell(u), v - u \rangle \leq \beta D_\Phi(v, u), \quad \forall u, v \in \mathcal{U}.$$

*If $\alpha \geq 0$, $\ell$ is also called $\alpha$-strongly convex relative to $\Phi$.*

In our problem we will mostly deal with the case of negative curvature $\alpha < 0$. Then if $L = \beta = -\alpha$, we refer to $\ell(\cdot)$ as being $L$-smooth relative to $\Phi$, or relative smooth with parameter $L$.

According to Proposition 1.1 of [LFN18], $\alpha$-$\beta$ relative smoothness is equivalent to

$$\alpha \nabla^2 \Phi(u) \preceq \nabla^2 \ell(u) \preceq \beta \nabla^2 \Phi(u), \qquad \text{for any } u \in \mathcal{U}. \tag{5}$$

In NGVI setting, we hope to find conditions under which negative ELBO objective (1)

$$\ell(\omega) = \underbrace{-\mathbb{E}_q[\log p(\mathcal{D} \,|\, z)]}_{\text{log-likelihood term}} + \underbrace{D_{\mathrm{KL}}(q(z) \,\|\, p(z))}_{\text{KL divergence term}} \tag{6}$$

is $L$-relative smooth on $\Omega$ w.r.t. $A^*$.

**Relative Smoothness of KL Divergence Term**  Since $A^*(\xi, \Xi) = -\frac{1}{2} \log \det(\Xi - \mathrm{diag}(\xi \odot \xi))$ (see Section C.1), and the KL divergence between Gaussian distributions admits a closed-form

$$D_{\mathrm{KL}}(q(z; \xi, \Xi) \,\|\, p(z)) = -\frac{1}{2} \log \det(\Xi - \mathrm{diag}(\xi \odot \xi)) + \frac{1}{2} \mathrm{Tr}(\Xi),$$

the Hessians of the two functions coincide. Thus KL divergence term is 1-1 smooth relative to $A^*$.

**Relative Smoothness of Log-Likelihood Term**  Now we consider the log-likelihood term in (6). We explain the intuition for the derivation in the univariate case and state the general result in high dimensional case, deferring the formal proof to Theorem 3.2.[2]

---

[2]For simplicity, we assume that $\mathcal{D} = \{(x, y)\}$ only contains a single data point. If there are $n$ data points and each $-\mathbb{E}_q[\log p(x_i, y_i | z)]$ is $\alpha_i$-$\beta_i$-relative smooth, then $-\mathbb{E}_q[\log p(\mathcal{D}|z)]$ is $\sum_{i=1}^n \alpha_i$-$\sum_{i=1}^n \beta_i$-relative smooth, since $-\mathbb{E}_q[\log p(\mathcal{D}|z)] = -\sum_{i=1}^n \mathbb{E}_q[\log p(x_i, y_i | z)]$.

In order to compute the Hessian matrix, $\nabla^2 \mathbb{E}_{q(z;\omega)}[f(z)]$, for some four times differentiable function $f$, we need the following useful lemma [Pri58, Bon64, OA09].

**Lemma 3.1** (Bonnet's and Price's Gradients). *Let $q(z)$ be the probability density function (PDF) of the multivariate Gaussian, $\mathcal{N}(\mu, \Sigma)$, and assume $f$ is twice continuously differentiable, then*

$$\nabla_\mu \mathbb{E}_{q(z;\mu,\Sigma)}[f(z)] = \mathbb{E}_q[\nabla f(z)], \qquad \nabla_\Sigma \mathbb{E}_{q(z;\mu,\Sigma)}[f(z)] = \frac{1}{2}\mathbb{E}_q[\nabla^2 f(z)].$$

In the univariate case with standard parameters, $\mu = \xi$, $\sigma^2 = \Xi - \xi^2$, we apply Theorem 3.1 and chain rule to obtain

$$\nabla_\xi \mathbb{E}_{q(z;\xi,\Xi)}[f(z)] = \mathbb{E}_q[\nabla f(z)]\frac{\partial \mu}{\partial \xi} + \frac{1}{2}\mathbb{E}_q[\nabla^2 f(z)]\frac{\partial \sigma^2}{\partial \xi} = \mathbb{E}_q[\nabla f(z) - \nabla^2 f(z) \cdot \xi],$$

$$\nabla_\Xi \mathbb{E}_{q(z;\xi,\Xi)}[f(z)] = \mathbb{E}_q[\nabla f(z)]\frac{\partial \mu}{\partial \Xi} + \frac{1}{2}\mathbb{E}_q[\nabla^2 f(z)]\frac{\partial \sigma^2}{\partial \Xi} = \frac{1}{2}\mathbb{E}_q[\nabla^2 f(z)].$$

Using Theorem 3.1 and chain rule again and we get the following result.

**Proposition 1.** *When $d = 1$, let $f(z) := -\log p(\mathcal{D} \mid z)$, then the Hessian of the log-likelihood term equals*

$$\nabla^2 \mathbb{E}_{q(z;\xi,\Xi)}[f(z)] = \begin{pmatrix} \mathbb{E}_q[-2\nabla^3 f(z) \cdot \mu + \nabla^4 f(z) \cdot \mu^2] & \frac{1}{2}\mathbb{E}_q[\nabla^3 f(z) - \nabla^4 f(z) \cdot \mu] \\ \frac{1}{2}\mathbb{E}_q[\nabla^3 f(z) - \nabla^4 f(z) \cdot \mu] & \frac{1}{4}\mathbb{E}_q[\nabla^4 f(z)] \end{pmatrix}. \quad (7)$$

Moreover, it is straightforward to see that

$$\nabla^2 A^*(\omega) = \frac{1}{\sigma^4} \begin{pmatrix} 2\mu^2 + \sigma^2 & -\mu \\ -\mu & \frac{1}{2} \end{pmatrix}. \quad (8)$$

Therefore, proving $\alpha$-$\beta$ relative smoothness is equivalent to finding $\alpha$, $\beta$ such that for all $\omega$,

$$\alpha \nabla^2 A^*(\omega) \preceq -\nabla^2 \mathbb{E}_{q(z;\omega)}[\log p(\mathcal{D} \mid z)] \preceq \beta \nabla^2 A^*(\omega). \quad (9)$$

Using the same approach, we can compute the Hessian matrices for any $d > 1$ (see Theorems G.1 and G.2).

For any $U \geq 1$, $D > 1$, define the bounded sets in the spaces of standard and expectation parameters:

Standard: $\quad \tilde{\mathcal{P}} := \{(\mu, \Sigma) : |\mu_i| \leq U, \Sigma \text{ is diagonal}, D^{-1} \leq \Sigma_{ii} \leq D, 1 \leq i \leq d\}$,

Expectation: $\quad \tilde{\Omega} := \{(\xi, \Xi) : (\xi, \Xi - \text{diag}(\xi \odot \xi)) \in \tilde{\mathcal{P}}\} \subseteq \Omega$.

Then we present our main result on relative smoothness on such bounded sets for any $U$ and $D$.[3]

**Theorem 3.2** (Sufficient Conditions for Relative Smoothness). *Let $f(z) = -\log p(\mathcal{D} \mid z)$ be a four times continuously differentiable function in $z$ on $\mathbb{R}^d$. Assume $\sup_{z \in \mathbb{R}^d} \sup_{i=1,\cdots,d} |\nabla_i f(z)| \leq L_1$, $\sup_{z \in \mathbb{R}^d} \sup_{i=1,\cdots,d} \sup_{j=1,\cdots,d} |\nabla_{ij}^2 f(z)| \leq L_2$. Then the log-likelihood term is L-smooth relative to $A^*$ on $\tilde{\Omega}$ with*

$$L = \mathcal{O}(dD^2 U(L_1 + L_2 U) + dD^3(L_1 + L_2 U)).$$

*Proof sketch.* First, we use the approach mentioned above to compute the Hessian matrices $\nabla^2 \mathbb{E}_{q(z;\xi,\Xi)}[f(z)]$ and $\nabla^2 A^*(\omega)$. To handle high-order derivatives appearing in $\nabla^2 \mathbb{E}_{q(z;\xi,\Xi)}[f(z)]$, we apply Stein's Lemma (Theorem F.3) to bound them by lower-order derivatives. Finally, we find $\alpha$ and $\beta$ as in (9) by proving the positive semidefiniteness of the corresponding matrices.

**Example 1.** For logistic regression, $-\log p(y \mid x, z) = \log(1 + e^{-yx^\top z})$, where $x \in \mathbb{R}^d$ is a data point and $y \in \{-1, 1\}$ is the label. Since $-\nabla_i \log p(y \mid x, z) = -\sigma(-yx^\top z)yx_i$ and $-\nabla_{ij}^2 \log p(y \mid x, z) = \sigma(-yx^\top z)(1 - \sigma(-yx^\top z))x_i x_j$, by writing $\|x\|_\infty := \max_i |x_i|$, we have

$$\sup_{z \in \mathbb{R}^d} \sup_{i=1,\cdots,d} |-\nabla_i \log p(y \mid x, z)| \leq \|x\|_\infty = L_1,$$

$$\sup_{z \in \mathbb{R}^d} \sup_{i=1,\cdots,d} \sup_{j=1,\cdots,d} |-\nabla_{ij}^2 \log p(y \mid x, z)| \leq \|x\|_\infty^2 = L_2.$$

Therefore, $\ell(\omega)$ is smooth relative to $A^*$ with parameter $\mathcal{O}(dD^2 \|x\|_\infty (U + D)(1 + U\|x\|_\infty))$.

---

[3]A stronger statement with tight characterization for the case $d = 1$ can be found in Section F.

**Comparison to conjugate case.** For conjugate models, $\ell(\omega)$ is 1-smooth and 1-strongly convex relative to $A^*$ [WG24], which makes it very well-conditioned strictly (but not strongly) convex objective. For non-conjugate models including logistic regression and Poisson regression, however, $\alpha$ is typically negative, indicating that $\ell(\omega)$ is a non-convex objective. The relative smoothness constant in this case *scales polynomially* with the size of the set $\tilde{\mathcal{P}}$ and dimension $d$.

### 3.2 Hidden Convexity of Variational Objective under Log-concave Likelihood

We show that $\ell(\omega)$ exhibits hidden convexity when the log-likelihood is concave in $z$, as in logistic and Poisson regression. Formal proofs of statements in this section are relegated to Section G.3.

A function has hidden convexity if it becomes convex under a reparameterization $c(\cdot)$. Formally:

**Definition 3.2** (Hidden Convexity; [FHH25])**.** *Let* $\ell : \Omega \to \mathbb{R}$ *satisfy* $\ell(\omega) = H(c(\omega))$ *for an invertible map* $c : \Omega \to \Theta$. *Then* $\ell$ *is hidden convex with modulus* $\mu_C > 0$, $\mu_H \geq 0$ *if:*

1. *The set* $\Theta$ *is convex and* $H : \Theta \to \mathbb{R}$ *is* $\mu_H$-*strongly convex w.r.t. Euclidean geometry.*

2. *The map* $c$ *is invertible and* $\exists \mu_C > 0 : \|c(\omega_1) - c(\omega_2)\|_2 \geq \mu_C \|\omega_1 - \omega_2\|_2$, $\quad \forall \omega_1, \omega_2 \in \Omega$.

The result below follows from the fact that the strong convexity of $f(u)$ transfers to the expected value $\mathbb{E}_{q(u;\mu,C)}[f(u)]$ under the Cholesky parameterization $(\mu, C)$, where $C$ is lower triangular with positive diagonals and $CC^\top = \Sigma$ [Dom20, Theorem 9].

**Proposition 2.** *If* $\log p(\mathcal{D} \,|\, z)$ *is concave in* $z$, *then the restriction of* $\ell(\omega) : \Omega \to \mathbb{R}$ *on* $\tilde{\Omega}$ *is hidden convex with modulus* $\mu_C = (4U^2 + 4D + 1)^{-1/2}$ *and* $\mu_H = 1$.

Given hidden convexity, we establish the PL inequality based on the analysis in [FHH25].

**Proposition 3.** *If* $\log p(\mathcal{D} \,|\, z)$ *is concave and a stationary point* $\omega^*$ *lies in* $\tilde{\Omega}$, *then* $\omega^*$ *is a global minimum. Furthermore,* $\ell(\omega)$ *satisfies the PL inequality in the relative interior* $\mathrm{ri}(\tilde{\Omega})$:

$$\|\nabla\ell(\omega)\|^2 \geq 2\mu_C^2(\ell(\omega) - \ell^*), \quad \forall\, \omega \in \mathrm{ri}(\tilde{\Omega}), \qquad \textit{with } \mu_C \textit{ from Proposition 2.} \tag{10}$$

The assumption that $\omega^* \in \tilde{\Omega}$ is mild. Under weak conditions, $\ell(\omega)$ is coercive: $\ell(\omega) \to \infty$ as $\|(\mu, \Sigma)\| \to \infty$ or $\det(\Sigma) \to 0$ (see Section D), thus for large enough $U$ and $D$, $\tilde{\Omega}$ contains a stationary point. Note that although the PL condition holds, it does not immediately imply the function value convergence of NGVI, since for non-Euclidean algorithms the gradient norm may converge arbitrarily slow. In the next section, we introduce an additional mild assumption (Assumption 2), sufficient for the function value convergence.

## 4 Convergence of Non-Convex NGVI

This section analyzes convergence properties of NGVI. In Section 4.1, we introduce our projected stochastic natural gradient descent (Proj-SNGD) algorithm and prove an $\mathcal{O}(1/\sqrt{T})$ convergence rate under the relative smoothness condition in Section 4.2. Finally, if the log-likelihood is concave, we show in Section 4.3 that Proj-SNGD achieves $\mathcal{O}(1/T)$ convergence to the global minimum.

### 4.1 Projected Stochastic Natural Gradient Descent

Given an initial point $\omega_0 \in \Omega$, a number of iterations $T$ and step sizes $\{\gamma_t\}_{0 \leq t \leq T-1}$, we introduce the update rule of our projected variant of SNGD (with $\omega_0 \in \tilde{\Omega}$)

$$\textsf{Proj-SNGD:} \qquad \nabla A^*(\omega_{t+1,*}) = \nabla A^*(\omega_t) - \gamma_t \hat{\nabla}\ell(\omega_t), \qquad \omega_{t+1} = \mathrm{Proj}_{\tilde{\Omega}}(\omega_{t+1,*}). \tag{11}$$

Here $\mathrm{Proj}_{\tilde{\Omega}}(\omega)$ denotes the non-Euclidean projection of $\omega$ onto $\tilde{\Omega}$ induced by geometry of $A^*$. This involves 1) a transformation from the expectation parameter $(\xi, \Xi)$ to the standard parameter $(\mu, \Sigma)$, 2) a Euclidean projection of $(\mu, \Sigma)$ onto $\tilde{\mathcal{P}}$, and 3) a reverse transformation back to the expectation space. Note that in the mean field case, the projection in the second step can be performed efficiently by a simple entry-wise clipping. Denote the clipping function $\mathrm{clip}_{[a,b]}(x) = \min\{\max\{a, x\}, b\}$, then the second formula of (11) is equivalent to

$$(\mu_{t+1})_i = \mathrm{clip}_{[-U,U]}((\mu_{t+1,*})_i), \quad (\Sigma_{t+1})_{ii} = \mathrm{clip}_{[1/D,D]}((\Sigma_{t+1,*})_{ii}), \quad \forall 1 \leq i \leq d.$$

**Importance of projection.** The projection onto the bounded set $\tilde{\Omega}$ is necessary for two reasons. Theoretically, as shown in Section 3, both relative smoothness and hidden convexity hold on the set $\tilde{\Omega}$, and SNGD can potentially escape this set (as shown in Figure 1), invalidating its convergence guarantees. Empirically, we will show in Section 5 that projection improves the numerical stability.

## 4.2 Convergence using Relative Smoothness

Equipped with the relative smoothness of $\ell(\cdot)$, we can prove that Proj-SNGD converges to a stationary point with an $\mathcal{O}(1/\sqrt{T})$ rate. We use $\hat{\nabla}\ell(\omega)$ to denote the stochastic gradient and assume $\mathbb{E}[\hat{\nabla}\ell(\omega) \,|\, \omega] = \nabla\ell(\omega)$. Randomness may stem from using mini-batches rather than the entire dataset when computing the gradient. Next, we impose the following assumption on the variance of stochastic gradients $\hat{\nabla}\ell(\omega_t)$.

**Assumption 1.** *There exists $V \geq 0$ such that*

$$\gamma_t^{-1}\mathbb{E}[\langle \hat{\nabla}\ell(\omega_t) - \nabla\ell(\omega_t), \omega_{t+1}^+ - \omega_{t+1}\rangle \,|\, \omega_t] \leq V^2 \qquad \text{for all } \omega_t \in \tilde{\Omega}, \tag{12}$$

*where $\omega_{t+1}^+ := \operatorname{argmin}_{\omega\in\tilde{\Omega}} \gamma_t\langle\nabla\ell(\omega_t), \omega\rangle + D_{A^*}(\omega, \omega_t)$ is the output of a* Proj-NGD *step with exact gradient.*

**Remark.** Assumption 1, first proposed by [HR21], reduces to $\mathbb{E}\|\hat{\nabla}\ell(\omega_t) - \nabla\ell(\omega_t)\|^2$ in standard gradient descent case, however, it does not depend on any particular norm in general. This assumption was justified and used in [WG24] to show convergence of SNGD in conjugate setting. In Section I, we prove Assumption 1 is satisfied for our (non-conjugate) logistic regression models.

For a general setting, we will use the Bregman Forward-Backward Envelope (BFBE; [ATP21, FH24]) as the convergence criterion to a stationary point.

**Definition 4.1** (Bregman Forward-Backward Envelope (BFBE)). *For some $\rho > 0$, the BFBE at $\omega \in \tilde{\Omega}$ is defined as $\mathcal{E}_\rho(\omega) := -2\rho\min_{\omega'\in\tilde{\Omega}}[\langle\nabla\ell(\omega), \omega' - \omega\rangle + \rho D_{A^*}(\omega', \omega)]$.*

In the Euclidean case with unconstrained domain, BFBE becomes the squared norm of the gradient $\mathcal{E}_\rho(\omega) = \|\nabla\ell(\omega)\|^2$. In non-Euclidean case it is a natural generalization of stationarity criteria to the geometry induced by $A^*$, and if $\omega \in \operatorname{ri}(\tilde{\Omega})$, then $\mathcal{E}_\rho(\omega) = 0$ if and only if $\|\nabla\ell(\omega)\| = 0$. It is shown in [FH24] that BFBE is the strongest criterion available for non-convex SMD. Next, we establish non-asymptotic convergence of Proj-SNGD using BFBE criterion.

**Theorem 4.1** (Convergence of Proj-SNGD). *Assume $\ell(\omega)$ has a bounded domain $\tilde{\Omega} \subseteq \Omega$ and $\operatorname{ri}(\tilde{\Omega})$ contains at least one stationary point $\omega^*$. Suppose $\ell$ is smooth with respect to $A^*$ with parameter $L$. Then under Assumption 1, for constant step size $\gamma_t = \gamma = \min\left\{\frac{1}{2L}, \sqrt{\frac{\lambda_0}{V^2 LT}}, \right\}$, Proj-SNGD satisfies*

$$\mathbb{E}[\mathcal{E}_{3L}(\bar{\omega}_T)] \leq 18\frac{L\lambda_0}{T} + 9\sqrt{\frac{LV^2\lambda_0}{T}},$$

*where $\lambda_0 := \ell(\omega_0) - \ell^*$ and $\bar{\omega}_T$ is sampled from $\{\omega_0, \ldots, \omega_{T-1}\}$ with probabilities $\gamma_t / \sum_{i=0}^{T-1}\gamma_i$.*

The proof is in Section H. Theorem 4.1 implies that Proj-SNGD converges at a rate of $\mathcal{O}(1/\sqrt{T})$ in the number of iterations with the presence of randomness in gradients. In noiseless setting ($V = 0$), we obtain a faster $\mathcal{O}(1/T)$ convergence rate. With Theorem 4.1, one may plug in the relative smoothness parameter derived from Theorem 3.2 to obtain the corresponding convergence rates.

## 4.3 Fast Convergence under Log-concave Likelihood

In this section, we prove that Proj-SNGD attains the global minimum at a fast $\mathcal{O}(1/T)$ rate if the negative log-likelihood is convex. This fast rate relies on the PL inequality established in Proposition 3, but requires the following additional technical assumption.

**Assumption 2.** *For any $\omega \in \partial\tilde{\Omega}$, let $\mathbf{n}_\omega$ be an outward normal direction at $\omega$.[4] Then it holds that*

$$\langle -(\nabla^2 A^*(\omega))^{-1}\nabla\ell(\omega), \mathbf{n}_\omega\rangle < 0.$$

---

[4]For a compact set $P := \{\omega \in \mathbb{R}^d : p_i(\omega) \leq 0, 1 \leq i \leq k\}$, the set of outward normal directions at $\omega \in \partial P$ is defined as the normalized cone of the gradients of the active constraints, i.e., $\operatorname{cone}\{\nabla p_i(\omega) : p_i(\omega) = 0, 1 \leq i \leq k\} \cap \{v \in \mathbb{R}^d, \|v\| = 1\}$.

In the proof of fast convergence of Proj-SNGD (see Theorem 4.2 below), we require that for every $\omega \in \tilde{\Omega}$,

$$\omega_* := \operatorname*{argmin}_{\omega' \in \Omega} \{ \langle \nabla \ell(\omega), \omega' \rangle + 2\rho D_{A^*}(\omega', \omega) \} \in \tilde{\Omega}, \tag{13}$$

where $\rho > 0$ is a constant that can be chosen arbitrarily large. When $\omega$ lies in the interior of $\tilde{\Omega}$, the condition (13) is satisfied for sufficiently large $\rho$. When $\omega$ lies on the boundary, Assumption 2 ensures that the gradient $\nabla \ell(\omega)$ points towards the interior of $\tilde{\Omega}$ under the geometry induced by $A^*$. It further guarantees that, for sufficiently large $\rho$, $\omega_*$ defined in (13) lies in $\tilde{\Omega}$.

The example below shows that Assumption 2 can be satisfied for a univariate Bayesian linear regression model.

**Example 2.** We consider a univariate Bayesian linear regression model with a single data point $(x, y)$, where $x, y, z \in \mathbb{R}$, $p(z) = \mathcal{N}(0, 1)$ and $p(y \mid x, z) = \mathcal{N}(xz, 1)$. It can be shown that (see Section H.4) Assumption 2 is satisfied if the following set of inequalities holds:

$$-U(x^2 + 1) < xy < U(x^2 + 1), \quad D^{-1} < x^2 + 1 < D. \tag{14}$$

Therefore, by carefully choosing $U$ and $D$, Assumption 2 can be satisfied for any data point $(x, y)$.

Now we are ready to present the $\mathcal{O}(1/T)$ global convergence guarantee of Proj-SNGD algorithm.

**Theorem 4.2** (Fast Convergence of Proj-SNGD). *Suppose* $\operatorname{ri}(\tilde{\Omega})$ *contains a stationary point* $\omega^*$ *and $\ell$ is smooth with respect to* $A^*$ *with parameter $L$. Suppose* $\log p(\mathcal{D} \mid z)$ *is concave in $z$. Let* $\mu_B := \frac{\mu_C^2}{9U^2 D^2}$. *Under Assumptions 1 and 2, for the step size scheme*

$$\gamma_t = \begin{cases} \frac{1}{2L}, & \text{if } t \leq T/2 \text{ and } T \leq \frac{6L}{\mu_B}, \\ \frac{6}{\mu_B(t - \lceil T/2 \rceil) + 12L}, & \text{otherwise}, \end{cases} \tag{15}$$

*the iterates of* Proj-SNGD *satisfy*

$$\mathbb{E}[\ell(\omega_{T,*}) - \ell^*] \leq \frac{192 L \lambda_0}{\mu_B} \exp\left(-\frac{\mu_B T}{12L}\right) + \frac{648 L V^2}{\mu_B^2 T},$$

*where* $\ell^* = \ell(\omega^*)$, $\omega_{t,*} := \operatorname{argmin}_{\omega' \in \tilde{\Omega}} \langle \nabla \ell(\omega_t), \omega' \rangle + L D_{A^*}(\omega', \omega_t)$ *and* $\lambda_0 := \ell(\omega_0) - \ell^*$.

The proof of Theorem 4.2 can be found in Section H.3. In the noiseless setting, $V = 0$, Theorem 4.2 implies linear convergence and in the stochastic case we have the $\mathcal{O}(1/T)$ rate. Notably, this rate matches the rate in [WG24] under conjugate assumption, and the rates in [DGG23, KOW+23] under strong convexity conditions. However, we work in much more general setting of non-conjugate models and in non-Euclidean geometry.

One limitation is our technical Assumption 2 that can be difficult to verify for a more complex model than a Bayesian linear regression. However, empirical results (see Section 5) indicate that with moderate values of $U$ and $D$, even if Assumption 2 may fail, the performance of Proj-SNGD is no worse (and sometimes better) than SNGD.

## 5 Experiments

**Numerical Stability of** SNGD **and** Proj-SNGD. In this experiment, we again focus on Poisson regression as discussed in Figure 1. We consider data $(x, y) = (0.9, 24)$ where $y$ is sampled from $\operatorname{Poisson}(e^{4x})$, fixed initialization of variance parameter $\sigma_0^2 = 2$, and stochastic initialization of mean parameter $\mu_0 \sim \operatorname{Unif}([-3, 0])$. We aim to demonstrate numerical stability of Proj-SNGD and SNGD.[5] The results of 10 independent runs are shown in Figure 2. The solid line represents the median and the shaded area indicates the first and third quartiles. In the left panel of Figure 2, we observe an increase in $\ell$ at the initial iteration when using SNGD with larger step sizes. However, with the same initialization, the increase is absent for Proj-SNGD, as shown in the right panel of Figure 2. Moreover, Proj-SNGD exhibits faster convergence and, despite the constraint an additional $\tilde{\Omega}$, it converges to the same optimal solution as SNGD.

---

[5] Poisson regression admits a closed-form gradient, and thus no randomness is involved in the training process. The only source of randomness is the initialization of $\mu_0$.

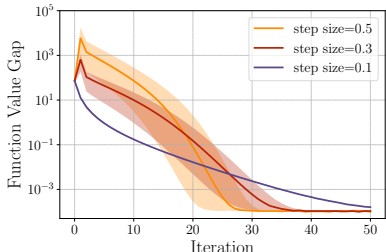
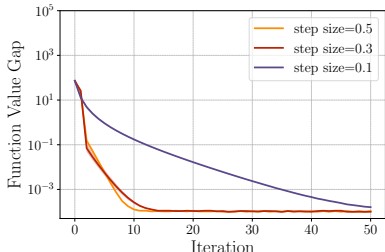

Figure 2: SNGD (left panel) and Proj-SNGD (right panel) applied to Poisson regression problem with different step-sizes and initialized with $\sigma_0^2 = 2$. Proj-SNGD is used with $U = 4$ and $D = 25$. *Non-Euclidean projection* improves convergence and sensitivity of SNGD. The left panel here is the same as the left panel in Figure 1. The right panel shows that projection fixes SNGD even when starting with the same (large) initial point $\sigma_0^2 = 2$.

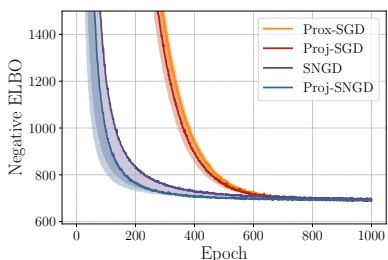
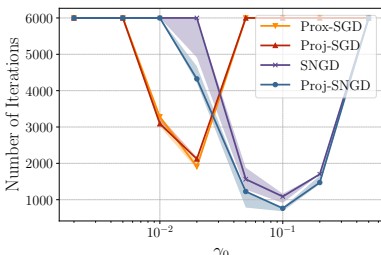

Figure 3: Euclidean and non-Euclidean algorithms on MNIST dataset. Left: Objective during optimization with tuned step size. Right: Number of iterations before the objective falls below $\ell(\omega) \leq 700$ for different initial step-sizes $\gamma_0$. Non-Euclidean algorithms show consistently better performance, tolerate larger step-sizes and are more robust to step-size tuning.

**Experiment on MNIST Dataset.** In this experiment, we compare non-Euclidean algorithms (Proj-SNGD and SNGD) with Euclidean algorithms (Prox-SGD and Proj-SGD) proposed in [Dom20]. Details of implementation and additional results can be found in Section J. We consider logistic regression on a subset of MNIST dataset [LeC98] with labels 6 or 8 ($n = 11769$, $d = 784$). We use mini-batches of size 2000 and set the step size $\gamma_t = \gamma_0/\sqrt{t}$, where $\gamma_0$ is a hyperparameter to be finetuned. We run the algorithm for 1000 epochs (6000 iterations). The results averaged over 5 independent runs are shown in Figure 3. In the left panel of Figure 3, we observe that non-Euclidean algorithms converge faster than Euclidean ones with fine-tuned step-size. The right panel illustrates robustness to the initial step size $\gamma_0$, with lower iteration counts indicating faster convergence. Non-Euclidean algorithms reach the threshold in around 1000–2000 iterations for a wide range $0.05 \leq \gamma_0 \leq 0.2$, while Euclidean algorithms only do so at $\gamma_0 = 0.02$. We also observe that Proj-SNGD slightly outperforms SNGD. Therefore, non-Euclidean algorithms, especially Proj-SNGD, are more robust to step-size, potentially reducing the tuning burden in practice.

Additional experimental results on other datasets can be found in Section J.2.

## 6 Limitations and Future Work

While our work makes a significant progress in understanding NGVI in non-conjugate models, there are certain limitation we want to discuss. First, our study of landscape properties is restricted to a compact domain. Whether global relative smoothness and/or PL inequality holds remains an open question. Second, our analysis is limited to the mean-field variational family; extending the analysis to full-covariance Gaussian family is an important direction for future work. Third, to obtain $\mathcal{O}(1/T)$ convergence rate, we invoke an assumption on the behavior of objective on the boundary of the domain, which might be challenging to verify, and it would be important to remove this assumption.

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

# Contents

# Appendix

## A  Summary of Assumptions in this Paper

Below is a summary of the assumptions used in this paper for establishing the convergence of Proj-SNGD:

1. The objective $\ell$ is smooth relative to $A^*$. This assumption is required for both Theorem 4.1 and Theorem 4.2, and can be verified using Theorem 3.2.

2. The stochastic gradient is unbiased with bounded variance (Assumption 1). This is needed for both Theorem 4.1 and Theorem 4.2, and holds for logistic regression models (see Section I).

3. The relative interior $\mathrm{ri}(\tilde{\Omega})$ contains a stationary point $\omega^*$. This condition is required for both Theorem 4.1 and Theorem 4.2. It is a mild assumption, since $\ell$ is coercive under very weak conditions (see Section D).

4. The descent direction points inward on the boundary of $\tilde{\Omega}$. This assumption is only required in Theorem 4.2, and it is satisfied, for example, in a univariate Bayesian linear regression model (see Section H.4).

5. The log-likelihood $\log p(\mathcal{D} \mid z)$ is concave in $z$. This is only needed in Theorem 4.2, and holds for many models, including logistic regression and Poisson regression.

# B  Descent Lemma for Mirror Descent

In this section, we prove a descent lemma for mirror descent.

**Lemma B.1.** *Let $\Phi$ be a strictly convex distance generating function on a closed domain $\Omega \subseteq \mathbb{R}^d$ with induced Bregman divergence $D_\Phi$. Assume that $\ell : \mathcal{U} \to \mathbb{R}$ is L-smooth relative to $\Phi$. Consider (deterministic) mirror descent update with step size $\gamma$*

$$u_{t+1} = \underset{u \in \mathcal{U}}{\operatorname{argmin}}\ \gamma \langle \nabla \ell(u_t), u \rangle + D_\Phi(u, u_t),$$

*then with step size $\gamma = 1/L$, the objective decreases after each iteration, i.e.,*

$$\ell(u_{t+1}) \le \ell(u_t).$$

Theorem B.1 indicates that if a mirror descent update with step size $\gamma$ causes an increase of the objective, the relative smoothness parameter must be greater than $1/\gamma$. Thus in the left panel of Figure 1, the smoothness parameter is greater than $1/0.3$, hence 1-relative smoothness fails in the non-conjugate model.

In addition, the necessary condition of $\alpha$-$\beta$ relative smoothness in univariate case (Theorem F.1) implies

$$\beta \ge \frac{\sigma^4}{2} x^4 e^{\frac{\sigma^2 x^2}{2} - 2x},$$

thus the smoothness parameter should grow exponentially with respect to $\sigma_0^2$. Therefore, Figure 1 is consistent with our theoretical findings.

*Proof.* By the second part of Lemma F.1 in [FH24] and the definition of $u_{t+1}$, we have that for all $v \in \mathcal{U}$,

$$\langle \nabla \ell(u_t), v \rangle + \frac{1}{\gamma} D_\Phi(v, u_t) \ge \langle \nabla \ell(u_t), u_{t+1} \rangle + \frac{1}{\gamma} D_\Phi(u_{t+1}, u_t) + \frac{1}{\gamma} D_\Phi(v, u_{t+1}).$$

Then, we use the relative smoothness of $\ell$ to get that for all $v \in \mathcal{U}$,

$$
\begin{aligned}
\ell(u_{t+1}) &\le \ell(u_t) + \langle \nabla \ell(u_t), u_{t+1} - u_t \rangle + L D_\Phi(u_{t+1}, u_t) \\
&= \ell(u_t) - \langle \nabla \ell(u_t), u_t \rangle + \left( \langle \nabla \ell(u_t), u_{t+1} \rangle + \frac{1}{\gamma} D_\Phi(u_{t+1}, u_t) \right) \\
&\le \ell(u_t) - \langle \nabla \ell(u_t), u_t \rangle + \langle \nabla \ell(u_t), v \rangle + \frac{1}{\gamma} D_\Phi(v, u_t) - \frac{1}{\gamma} D_\Phi(v, u_{t+1}).
\end{aligned}
$$

Finally, we choose $v = u_t$ and the result follows. $\qquad\square$

# C  Background on Exponential Family

In this section, we will provide background information about the exponential family. In Section C.1, we will derive the natural and expectation parameters of a Gaussian distribution. We will prove the simple form of NGD update (formula (4)) in Section C.2.

## C.1  Facts about Gaussian Variational Family

Recall that $q$ belongs to exponential family if it takes the following form:

$$q(z; \eta) = h(z) \exp\left( \langle \phi(z), \eta \rangle - A(\eta) \right),$$

where $\eta$ is the natural parameter, $\phi(z)$ is the sufficient statistic, and $A(\eta)$ is the log-partition function. The expectation parameter is defined as $\omega = \mathbb{E}_{q(z;\eta)}[\phi(z)]$. Now we let $q(z)$ be the PDF of Gaussian random vector $\mathcal{N}(\mu, \Sigma)$, then

$$q(z) \propto \exp\left( (z - \mu)^\top \Sigma^{-1} (z - \mu) - \frac{1}{2} \log \det(\Sigma) \right)$$

$$\propto \exp\left( \langle zz^\top, -\frac{1}{2} \Sigma^{-1} \rangle + \langle z, \Sigma^{-1} \mu \rangle - \langle \mu, \Sigma^{-1} \mu \rangle - \frac{1}{2} \log \det(\Sigma) \right).$$

Therefore, the sufficient statistics are $z$ and $zz^\top$, and we have the following fact.

**Fact 1.** *For multivariate Gaussian distribution $q$, the expectation parameters of $q$ are given by $\xi := \mathbb{E}[z] = \mu$ and $\Xi := \mathbb{E}[zz^\top] = \Sigma + \mu\mu^\top$. The natural parameters of $q$ are given by $\lambda := \Sigma^{-1}\mu$ and $\Lambda := -\frac{1}{2}\Sigma^{-1}$.*

Moreover, by plugging the definition of $\lambda$ and $\Lambda$, one have

$$A(\lambda, \Lambda) = \langle \mu, \Sigma^{-1} \mu \rangle + \frac{1}{2} \log \det(\Sigma) = -\frac{1}{4} \lambda^\top \Lambda^{-1} \lambda - \frac{1}{2} \log \det(-2\Lambda).$$

Next, we compute $A^*$ via the definition of convex conjugate:

$$A^*(\xi, \Xi) = \max_{\lambda, \Lambda \prec 0} \left\{ \langle \xi, \lambda \rangle + \langle \Xi, \Lambda \rangle + \frac{1}{4} \lambda^\top \Lambda^{-1} \lambda + \frac{1}{2} \log \det(-2\Lambda) \right\}.$$

It's easy to see that the optimal solution of the previous problem is given by

$$\lambda^* = -2\Lambda^* \xi, \quad \Lambda^* = (\xi\xi^\top - \Xi)^{-1}.$$

Moreover, we have the following result.

**Fact 2.** *The convex conjugate $A^*$ equals*

$$A^*(\xi, \Xi) = \begin{cases} -\frac{1}{2} \log \det(\Xi - \xi\xi^\top) + d + \frac{d}{2} \log 2, & \text{if } \Xi - \xi\xi^\top \succ 0, \\ +\infty, & \text{otherwise.} \end{cases}$$

Based on Fact 2, we can check that

$$\nabla_\lambda A(\lambda, \Lambda) = -\frac{1}{2} \Lambda^{-1} \lambda = \xi, \quad \nabla_\Lambda A(\lambda, \Lambda) = \frac{1}{4} \Lambda^{-1} \lambda \lambda^\top \Lambda^{-1} - \frac{1}{2} \Lambda^{-1} = \Xi,$$

$$\nabla_\xi A^*(\xi, \Xi) = (\Xi - \xi\xi^\top)^{-1} \xi = \lambda, \quad \nabla_\Xi A^*(\xi, \Xi) = -\frac{1}{2}(\Xi - \xi\xi^\top)^{-1} = \Lambda.$$

This also indicates that $\nabla A$ and $\nabla A^*$ are inverse operators of each other.

Moreover, when $d = 1$, it's straightforward to check that

$$\nabla^2 A^*(\omega) = \frac{1}{\sigma^4} \begin{pmatrix} 2\mu^2 + \sigma^2 & -\mu \\ -\mu & \frac{1}{2} \end{pmatrix}.$$

This result will be useful in the proof of relative smoothness, e.g., in Section 3.1 (8).

Next, we consider the mean field family,

$$q(z) \propto \exp\left( \sum_{i=1}^d -\frac{1}{2\Sigma_{ii}} z_i^2 + \frac{\mu_i}{\Sigma_{ii}} z_i + \frac{\mu_i^2}{\Sigma_{ii}} - \frac{1}{2} \log \Sigma_{ii} \right).$$

**Fact 3.** *For multivariate Gaussian distribution $q$ under mean-field parameterization, the expectation parameters of $q$ are given by $\xi_i := \mathbb{E}[z_i] = \mu_i$ and $\Xi_{ii} := \mathbb{E}[z_i^2] = \Sigma_{ii} + \mu_i^2$ entry-wise. Equivalently, we have*

$$\xi = \mu, \quad \Xi = \Sigma + \mathrm{diag}(\mu \odot \mu).$$

*Similarly, we also have $\lambda := \Sigma^{-1}\mu$ and diagonal matrix $\Lambda = -\frac{1}{2}\Sigma^{-1}$.*

In addition, Fact 3 implies that the transformation between any two of the natural, expectation and standard parameters can be performed efficiently in $\mathcal{O}(d)$ time. Therefore, every update of Proj-SNGD in (11) (given stochastic gradient $\hat{\nabla}\ell(\omega_t)$) can be performed in $\mathcal{O}(d)$ time.

## C.2   Derivation of SNGD

In this section, we show that formula (4) holds, i.e.,

$$\mathcal{I}(\eta_t)^{-1}\nabla\ell(\eta_t) = \nabla\ell(\omega_t),$$

where $\eta_t$ and $\omega_t$ are the natural and expectation parameter of the same distribution. The proof is adapted from [RM15] and [WG24].

*Proof of Equation (4).*   We first prove that $\mathcal{I}(\eta) = \nabla^2 A(\eta)$. Since

$$q(z;\eta) = h(z)\exp(\langle\phi(z),\eta\rangle - A(\eta)),$$

we have

$$\nabla_\eta \log q(z;\eta) = \phi(z) - \nabla A(\eta) = \phi(z) - \mathbb{E}[\phi(z)].$$

By definition of Fisher information and duality between $\eta$ and $\omega$,

$$\mathcal{I}(\eta) = \mathbb{E}[\nabla_\eta \log q(z;\eta)\nabla_\eta \log q(z;\eta)^\top] = \mathrm{Cov}(\phi(z)).$$

Moreover, we have

$$
\begin{aligned}
\nabla^2 A(\eta) &= \nabla_\lambda(\mathbb{E}[\phi(z)]) \\
&= \nabla_\lambda \int q(z)\phi(z)\,\mathrm{d}z \\
&= \int \nabla_\lambda(q(z))\phi(z)\,\mathrm{d}z \\
&= \int q(z)\nabla_\lambda(\log q(z))\phi(z)\,\mathrm{d}z \\
&= \mathbb{E}[\phi(z)(\phi(z) - \mathbb{E}[\phi(z)])^\top] \\
&= \mathrm{Cov}(\phi(z)).
\end{aligned}
$$

Then, we use the chain rule to get

$$\nabla_\eta \ell_\eta(\eta) = \nabla_\eta \ell_\omega(\nabla A(\eta)) = \nabla_\eta^2 A(\eta) \cdot \nabla_\omega \ell_\omega(\omega) = \mathcal{I}(\eta)\nabla\ell(\omega).$$

Multiplying $\mathcal{I}(\eta)^{-1}$ on both sides gives the desired result. $\qquad\square$

# D  Coerciveness of the Variational Objective for Gaussian Prior and Gaussian Variational Family

In this subsection, we aim to investigate the coerciveness of the variational loss $\ell(\omega)$ with Gaussian variational family.

**Definition D.1** (Coerciveness on Bounded Domain). *Let $f : \mathcal{U} \to \mathbb{R}$ be a real-valued function defined on an open set $\mathcal{U}$. We say $f$ is coercive if $f(\omega) \to +\infty$ as $u \in \mathcal{U}$, $u \to \partial\mathcal{U}$ or $\|u\| \to \infty$, where $\partial\mathcal{U}$ denotes the boundary of $\mathcal{U}$.*

Notice that coerciveness may potentially depend on specific parameterization. We consider 4 different parameterizations of Gaussian variational family in our work:

$$
\begin{aligned}
\text{Standard:} & \qquad (\mu, \Sigma) \\
\text{Expectation:} & \qquad \omega = (\mu, \Sigma + \mu\mu^\top) \\
\text{Cholesky:} & \qquad c(\omega) = (\mu, C) \\
\text{Natural:} & \qquad \eta = \left(\Sigma^{-1}\mu, -\frac{1}{2}\Sigma^{-1}\right)
\end{aligned}
$$

In Cholesky parameterization, $C$ is defined as the Cholesky factor of $\Sigma$. Since our focus is the convergence of NGVI, we are mainly interested in the landscape properties of $\ell(\omega)$ in expectation parameterization. We first show objective $\ell(\mu, \Sigma)$ is coercive in standard parameterization $(\mu, \Sigma) \in \mathbb{R}^d \times \mathcal{S}_+^d$.

**Theorem D.1** (Coerciveness of the Variational Objective in Standard Parameterization). *Let $q(z) = \mathcal{N}(z; \mu, \Sigma)$ be a Gaussian variational distribution with mean parameter $\mu \in \mathbb{R}^d$ and covariance matrix $\Sigma \in \mathcal{S}_+^d$. Let the prior be the standard normal distribution $p(z) = \mathcal{N}(z; 0, I)$. Suppose the log-likelihood function satisfies a* sub-quadratic growth *condition: there exist constants $c_0, c_1 \geq 0$ and an exponent $r \in [1, 2)$ such that for any observed data $\mathcal{D}$ and all $z \in \mathbb{R}^d$,*

$$\log p(\mathcal{D} \mid z) \leq c_0 + c_1 \|z\|^r. \tag{16}$$

*Then the variational objective $\ell : \mathbb{R}^d \times \mathcal{S}_+^d \to \mathbb{R}$ defined in (6) is coercive, in the sense that*

$$\|(\mu, \Sigma)\| \to \infty \quad or \quad \det(\Sigma) \to 0 \quad \implies \quad \ell(\mu, \Sigma) \to \infty.$$

The sub-quadratic growth assumption (16) is a very mild condition which is satisfied by most problems in practice. For example, the log-likelihood function $\log p(\mathcal{D} \mid z) = \log p(y \mid x, z)$ for dataset $\mathcal{D} = \{(x_i, y_i)\}_{i=1}^n$ diverges to $-\infty$ when $\|z\| \to \infty$ in Bayesian regression and logistic regression, and has at most linear growth in Poisson regression for bounded $\{x_i\}_{i=1}^n$.

With this result, we can prove that the objective is coercive in all parameterizations as listed below.

1. **Expectation parameterization.** The domain of expectation parameter is $\Omega = \{(\xi, \Xi) : \xi \in \mathbb{R}^d, \Xi - \xi\xi^\top \in \mathcal{S}_+^d\}$. For $\xi$, $\|\xi\| \to \infty$ if and only if $\|\mu\| \to \infty$. For unboundedness of $\Xi$, since $\|\Xi\| \leq \|\Sigma\| + \|\mu\|^2$, $\|\Xi\| \to \infty$ implies at least one of $\|\Sigma\| \to \infty$ and $\|\mu\| \to \infty$ holds. Moreover, if $(\xi, \Xi) \to \partial\Omega$, i.e., $\det(\Xi - \xi\xi^\top) \to 0$, this is equivalent to $\det(\Sigma) \to 0$.

2. **Cholesky parameterization.** Since $C$ is defined as a lower-triangle matrix with positive diagonal entries, $\|C\| \to \infty$ if and only if $\|\Sigma\| \to \infty$, and $\det(C) \to 0$ if and only if $\det(\Sigma) \to 0$.

3. **Natural parameterization.** The domain of natural parameter is $\mathcal{D}_A = \{(\lambda, \Lambda) : \lambda \in \mathbb{R}^d, -\Lambda \in \mathcal{S}_+^d\}$. For $\lambda$, $\|\lambda\| \leq \|\Sigma^{-1}\|\|\mu\| = (\lambda_{\min}(\Sigma))^{-1}\|\mu\|$, thus $\|\lambda\| \to 0$ implies either $\det(\Sigma) \to 0$ or $\|\mu\| \to 0$. The proof of $\Lambda$ is obvious.

*Proof of Theorem D.1.* It's clear that the boundary of $\mathbb{R}^d \times \mathcal{S}_+^d$ is $\mathbb{R}^d \times \{\Sigma : \Sigma \text{ is symmetric and } \det(\Sigma) = 0\}$. To show coerciveness, we need to prove that under the following 3 cases, the objective $\ell(\mu, \Sigma)$ will diverge to infinity:

1. $\|\mu\| \to \infty$,

2. $\det(\Sigma) \to 0$,

3. $\|\Sigma\| \to \infty$.

Recall from (6) that the variational objective is given by

$$\ell(\mu, \Sigma) = D_{\mathrm{KL}}(q(z) \,\|\, p(z)) - \mathbb{E}_q[\log p(\mathcal{D} \,|\, z)].$$

For the first term, we can find the closed-form solution of KL divergence between to Gaussian distributions:

$$D_{\mathrm{KL}}(q(z) \,\|\, p(z)) = \frac{1}{2} \left[ -\log \det(\Sigma) + \mathrm{Tr}(\Sigma) + \|\mu\|^2 - d \right].$$

For the second term, the sub-quadratic growth condition (16) implies that

$$-\mathbb{E}_q[\log p(\mathcal{D} \,|\, z)] \geq -c_0 - c_1 \mathbb{E}[\|z\|^r].$$

It remains to upper bound $\mathbb{E}[\|z\|^r]$. Let $x \sim \mathcal{N}(0, I)$ be a standard Gaussian random vector, then by Minkowski inequality and Jensen's inequality we have

$$\begin{aligned}
(\mathbb{E}[\|z\|^r])^{1/r} &= (\mathbb{E}[\|\mu + \Sigma^{1/2} x\|^r])^{1/r} \\
&\leq \|\mu\| + (\mathbb{E}[(\|\Sigma^{1/2} x\|^2)^{r/2}])^{1/r} \\
&\leq \|\mu\| + ((\mathbb{E}[\|\Sigma^{-1/2} x\|^2])^{r/2})^{1/r} \\
&= \|\mu\| + \mathrm{Tr}(\Sigma)^{1/2}.
\end{aligned}$$

Therefore, since $1 \leq r < 2$, it can be shown that for some constant $C > 0$, it holds that

$$\mathbb{E}[\|z\|^r] \leq C(\|\mu\|^r + \mathrm{Tr}(\Sigma)^{r/2}).$$

Finally, we have

$$\ell(\mu, \Sigma) \geq \frac{1}{2} \left[ -\log \det(\Sigma) + \mathrm{Tr}(\Sigma) + \|\mu\|^2 - d \right] - c_0 - c_1 C(\|\mu\|^r + \mathrm{Tr}(\Sigma)^{r/2}).$$

If $\det(\Sigma) \to 0$, the first term will diverge to $+\infty$. If $\|\mu\| \to \infty$ or $\|\Sigma\| \to \infty$, the terms in the square brackets dominate as $r < 2$, hence $\ell(\mu, \Sigma)$ will diverge to $+\infty$. $\qquad\square$

# E   Examples of Non-convex Objectives in Non-conjugate Models

In this section, we present two non-conjugate models where the objective is non-convex. These examples are also discussed in Section 5 of [WG24], but we include them here for completeness.

**Logistic Regression.** We consider 1-dimensional logistic regression model with data $\mathcal{D} = \{(x_i, y_i) : x_i \in [-1, 1], y_i \in \{-1, 1\}\}_{i=1}^n$ and latent variable $z = (w, b) \in \mathbb{R}^2$. The prior $p(z)$ is standard Gaussian. Then we have

$$\ell(\omega) = \sum_{i=1}^m \mathbb{E}_{q(w,b)}\left[\log(1 + \exp(-y_i(wx_i + b)))\right] + D_{\mathrm{KL}}(q(w,b) \,\|\, p(w,b)).$$

In the following we will write $s_i := \sigma(-y_i(wx_i + b))$ for $1 \le i \le n$, where $\sigma(\cdot)$ is the sigmoid function. On the convex subset

$$\Omega_1 := \{\omega = (0, \Xi) : \Xi \in \mathcal{S}_+^2 \text{ and } \Xi = \mathrm{diag}(\sigma_1^2, \sigma_2^2)\} \subseteq \Omega,$$

we can compute the second derivative with respect to $\sigma_2^2$ using Theorem 3.1 as

$$\nabla_{\sigma_2^2}^2 \ell(\omega) = \frac{1}{4} \sum_{i=1}^n \mathbb{E}_{q(w,b)}[s_i(1 - s_i)(6s_i^2 - 6s_i + 1)] + \frac{1}{2\sigma_2^4}.$$

A necessary condition of convexity of $\ell(\omega)$ is that the second derivative w.r.t. $\sigma_2^2$ is non-negative.

Note that as $\sigma_1^2, \sigma_2^2 \to 0$, $w, b \to 0$ and $s_i \to 1/2$ in probability for all $1 \le i \le n$, thus

$$\lim_{\sigma_1^2, \sigma_2^2 \to 0} \frac{1}{4} \mathbb{E}_{q(w,b)}[s_i(1 - s_i)(6s_i^2 - 6s_i + 1)] = -\frac{1}{32} < 0.$$

Then there exists $\delta > 0$ such that when $\sigma_1^2 = \sigma_2^2 = \delta$,

$$s_i(1 - s_i)(6s_i^2 - 6s_i + 1) \le -\frac{1}{64}$$

holds for all $1 \le i \le n$. Then we can choose $n > 32/\delta^2$ to show that $\ell(\omega)$ is non-convex when $\sigma_1^2 = \sigma_2^2 = \delta$, i.e.,

$$\nabla_{\sigma_2^2}^2 \ell(\omega)\Big|_{\xi=0, \Xi=\mathrm{diag}(\delta,\delta)} < 0.$$

**Poisson Regression.** In this example, we are given dataset $\mathcal{D} = \{(x_i, y_i) : x_i \in \mathbb{R}^d, y_i \in \mathbb{N}\}_{i=1}^n$ with latent variable $z \in \mathbb{R}^d$. We assume that $y \,|\, x \sim \mathrm{Poisson}(\lambda)$, where

$$\lambda = \exp(z^\top x).$$

The expected log-likelihood admits the following closed-form solution:

$$-\mathbb{E}_{q(z)} \log(y|x, z) = \sum_{i=1}^n \left[-y_i x_i^\top \xi + \exp\left(x_i^\top \xi + \frac{1}{2} x_i^\top (\Xi - \xi\xi^\top) x_i\right)\right].$$

Then we can compute the Hessian w.r.t. $\xi$:

$$\nabla_\xi^2 \ell(\omega) = \sum_{i=1}^n \exp\left(x_i^\top \xi + \frac{1}{2} x_i^\top (\Xi - \xi\xi^\top) x_i\right) x_i^\top \xi (x_i^\top \xi - 2) x_i x_i^\top + \nabla_\xi^2 A(\omega).$$

We consider the following convex subset where the covariance matrix equals identity, i.e.,

$$\Omega_2 := \{\omega = (\xi, \Xi) : \xi \in \mathbb{R}^d, \Xi = \xi\xi^\top + 2I\}.$$

We can find $x_i$ and $\xi$ such that $0 < x_i^\top \xi < 2$ for all $1 \le i \le n$. Then we have $\exp\left(x_i^\top \xi + \frac{1}{2} x_i^\top (\Xi - \xi\xi^\top) x_i\right) > 1$, and thus

$$\nabla_\xi^2 \ell(\omega) \preceq \sum_{i=1}^n x_i^\top \xi (x_i^\top \xi - 2) x_i x_i^\top + \nabla_\xi^2 A(\omega),$$

where we used the fact that $D_{\mathrm{KL}}(q(z;\omega)\,\|\,q(z;\omega')) = D_{A^*}(\omega,\omega')$. Moreover,

$$\nabla_\xi^2 A^*(\omega) = (1 + \xi^\top \Sigma^{-1}\xi)\Sigma^{-1} + \Sigma^{-1}\xi\xi^\top\Sigma^{-1}.$$

Therefore, for some $c > 0$, if rescale the dataset $\mathcal{D}' = \{(cx_i, y_i) : x_i \in \mathbb{R}^d, y_i \in \mathbb{N}\}_{i=1}^n$ and evaluate $\ell(\omega)$ at $\xi' = \xi/c$, we have

$$\nabla_\xi^2 \ell(\omega)\Big|_{\xi'=\xi/c} \preceq \sum_{i=1}^n c^2 x_i^\top \xi (x_i^\top \xi - 2) x_i x_i^\top + (1 + c^{-2}\|\xi\|^2)I + c^{-2}\xi\xi^\top \prec 0,$$

if $c$ is sufficiently large.

# F Tighter Relative Smoothness in Univariate Case ($\mu, \sigma \in \mathbb{R}$)

In this section, we will provide tighter relative smoothness guarantees in univariate case, i.e., $\mu, \sigma \in \mathbb{R}$, using the equivalent conditions for relative smoothness in (9).

## F.1 Necessary and Sufficient Conditions for Relative Smoothness

Let $f(z) := -\log p(\mathcal{D} \,|\, z)$. Inequalities (9) give the necessary and sufficient conditions under which relative smoothness holds. We first find $\beta$ such that the inequality on the right hand side of (9) holds, i.e.,

$$\beta \nabla^2 A^*(\omega) - \nabla^2 \mathbb{E}_{q(z;\omega)}[f(z)] \succeq 0. \tag{17}$$

Let $B = \mathbb{E}_q[\nabla^3 f(z)]$, $C = \mathbb{E}_q[\nabla^4 f(z)]$. Using the explicit form of the Hessians (7) and (8), the relative smoothness condition (17) is equivalent to

$$M(\beta) = \begin{pmatrix} \frac{\beta}{\sigma^2} + \frac{2\mu^2\beta}{\sigma^4} + [2\mu B - \mu^2 C] & -\frac{\mu\beta}{\sigma^4} - \frac{1}{2}[B - \mu C] \\ -\frac{\mu\beta}{\sigma^4} - \frac{1}{2}[B - \mu C] & \frac{\beta}{2\sigma^4} - \frac{1}{4}C \end{pmatrix} \succeq 0. \tag{18}$$

**Theorem F.1** (Necessary and Sufficient Conditions for Relative Smooth Condition (17), Univariate). *Let $f$ be a four times continuously differentiable function on $\mathbb{R}$. Then for Gaussian variational family $\mathcal{Q} = \{q(z;\omega) : \omega = (\xi, \Xi) \in \Omega\}$ and some positive constant $\beta$, condition (17) holds* if and only if *for all $\mu \in \mathbb{R}, \sigma^2 > 0$,*

$$\begin{cases} \sigma^4 \mathbb{E}_q[\nabla^4 f(z)] \leq 2\beta, \\ \dfrac{\beta}{\sigma^2} + \dfrac{2\mu^2\beta}{\sigma^4} \geq -2\mu \mathbb{E}_q[\nabla^3 f(z)] + \mu^2 \mathbb{E}_q[\nabla^4 f(z)], \\ \dfrac{\beta^2}{2\sigma^6} - \dfrac{\mathbb{E}_q[\nabla^4 f(z)]\beta}{4\sigma^2} - \dfrac{(\mathbb{E}_q[\nabla^3 f(z)])^2}{4} \geq 0. \end{cases} \tag{19}$$

**Remark.** Theorem F.1 is the immediate result of Sylvester's criterion of positive-semidefiniteness of matrix $M(\beta)$.

The first condition corresponds to the non-negativeness of the bottom right entry.

The second condition corresponds to the non-negativeness of the top left entry. Note that this is not a quadratic function of $\mu$ since $B$ and $C$ implicitly depend on $\mu$. However, if we impose a uniform bound $|C| \leq \beta_4$ and $|B| \leq \beta_3$ for some $\beta_4, \beta_3 \geq 0$ for all $\mu \in \mathbb{R}, \sigma^2 > 0$, the following inequality will be a sufficient condition for it to hold:

$$\left( \frac{2\beta}{\sigma^4} - \beta_4 \right) \mu^2 - 2\beta_3\mu + \frac{\beta}{\sigma^2} \geq 0. \tag{20}$$

(20) holds for all $\mu \in \mathbb{R}$ if and only if: either the coefficients of quadratic and linear terms are both zero ($\nabla^3 f(z) = 0$), or the discriminant is non-positive, which gives

$$\frac{\beta}{\sigma^2} \left( \frac{\beta}{2\sigma^4} - \frac{\beta_4}{4} \right) - \frac{\beta_3^2}{4} \geq 0. \tag{21}$$

The last condition corresponds to the non-negativeness of the determinant. A direct computation of the determinant gives

$$\det(M(\beta)) = \mu^2 \left[ \left( \frac{2\beta}{\sigma^4} - C \right) \left( \frac{\beta}{2\sigma^4} - \frac{C}{4} \right) - \left( \beta - \frac{C}{2} \right)^2 \right]$$
$$+ \mu \left[ 2B \left( \frac{\beta}{2\sigma^4} - \frac{C}{4} \right) - B \left( \frac{\beta}{\sigma^4} - \frac{C}{2} \right) \right] + \frac{\beta}{\sigma^2} \left( \frac{\beta}{2\sigma^4} - \frac{C}{4} \right) - \frac{B^2}{4}.$$

Note that the coefficients of the quadratic and linear terms are both 0, therefore we only need to guarantee that the constant term is non-negative. Interestingly, this condition is very similar to the non-positive discriminant condition (21): one substitutes $B, C$ with the upper bounds of $|B|, |C|$ to obtain (21).

We also need to find conditions under which the inequality on the left side of (9) holds, i.e.,

$$\nabla^2 \mathbb{E}_{q(z;\omega)}[f(z)] - \alpha \nabla^2 A^*(\omega) \succeq 0, \tag{22}$$

which is equivalent to

$$-M(\alpha) = \begin{pmatrix} -2\mu B + \mu^2 C - \frac{\alpha}{\sigma^2} - \frac{2\mu^2\alpha}{\sigma^4} & \frac{1}{2}[B - \mu C] + \frac{\mu\alpha}{\sigma^4} \\ \frac{1}{2}[B - \mu C] + \frac{\mu\alpha}{\sigma^4} & \frac{1}{4}C - \frac{\alpha}{2\sigma^4} \end{pmatrix} \succeq 0.$$

We apply Sylvester's criterion as before to obtain the following result.

**Theorem F.2** (Necessary and Sufficient Conditions for Relative Weak Convexity (22), Univariate)**.**
*Let f be a (a.e.) fourth differentiable function on $\mathbb{R}$. Then for Gaussian variational family $\mathcal{Q} = \{q(z;\omega) : \omega = (\xi, \Xi) \in \Omega\}$ and some constant $\alpha$, condition (22) holds if and only if for all $\mu \in \mathbb{R}, \sigma^2 > 0$,*

$$\begin{cases} \sigma^4 \mathbb{E}_q[\nabla^4 f(z)] \geq 2\alpha, \\ \dfrac{2\mu^2\alpha}{\sigma^4} + \dfrac{\alpha}{\sigma^2} \leq -2\mu \mathbb{E}_q[\nabla^3 f(z)] + \mu^2 \mathbb{E}_q[\nabla^4 f(z)], \\ \dfrac{\alpha^2}{2\sigma^6} - \dfrac{\mathbb{E}_q[\nabla^4 f(z)]\alpha}{4\sigma^2} - \dfrac{(\mathbb{E}_q[\nabla^3 f(z)])^2}{4} \geq 0. \end{cases} \tag{23}$$

**Remark.** The first condition implies an important necessary condition for relative convexity (i.e., $\alpha \geq 0$): $\nabla^4 f(z) \geq 0$ for all $z \in \mathbb{R}$. Indeed, if $\nabla^4 f(z_0) < 0$ for some $z_0$ and $\nabla^4 f(z)$ is continuous, we can always pick some $(\mu, \sigma^2)$ such that $\mathbb{E}_q[\nabla^4 f(z)] < 0$ (e.g. $\mu = z_0$ and $\sigma^2$ is sufficiently small), then we cannot find any $\alpha \geq 0$ such that the first condition holds.

### F.2  Simple Sufficient Conditions for Relative Smoothness

In the following, we aim to provide sufficient conditions for (17) and (22) with lower-order derivative conditions by applying Stein's lemma.

**Lemma F.3** (Stein's Lemma [Ste81])**.** *Let $Z \sim \mathcal{N}(\mu, \Sigma)$, then for any differentiable function g, we have*

$$\mathbb{E}[g(Z)(Z - \mu)] = \Sigma \mathbb{E}[\nabla g(Z)].$$

Theorem F.3 can be used to reduce the order of derivatives. More specifically, we can show that (see Theorem G.3 and Section G.2 for a more general proof in multivariate case)

$$|\mathbb{E}_q[\nabla^4 f(z)]| \leq \sigma^{-2} \sup_z |\nabla^2 f(z)|, \qquad |\mathbb{E}_q[\nabla^3 f(z)]| \leq \sigma^{-2} \sup_z |\nabla f(z)|. \tag{24}$$

Combining the parts above, we can get the following sufficient conditions.

**Corollary F.3.1** (Sufficient Conditions for Relative Smoothness, Univariate)**.** *Under the assumptions of Theorem F.1, if we further assume that $\sup_z |\nabla f(z)| \leq L_1$ and $\sup_z |\nabla^2 f(z)| < L_2$ for some constant $L_1, L_2 \geq 0$, then on the restriction of the parameter set $\{(\mu, \sigma^2) : 0 < \sigma^2 \leq D\}$ for some $D > 0$, condition (17) holds with*

$$\beta = \frac{D}{2}L_2 + \frac{\sqrt{2D}}{2}L_1.$$

*Moreover, relative weak convexity condition (22) holds with parameter*

$$\alpha = -\frac{D}{2}L_2 - \frac{\sqrt{2D}}{2}L_1.$$

*Proof.* For the choice of $\beta$, we first check three conditions in Theorem F.1 one by one:

1. Since $\sigma^2 \leq D$ and $\sup_z |\nabla^2 f(z)| \leq L_2$, use the upper bound on $|\mathbb{E}_q[\nabla^4 f(z)]|$ in (24) and we have

$$\sigma^4 \mathbb{E}_q[\nabla^4 f(z)] \leq \sigma^2 \sup_z |\nabla^2 f(z)| < DL_2 \leq \frac{\beta}{2}.$$

2. Thanks to our global bound on $|\mathbb{E}[\nabla^4 f(z)]|$ and $|\mathbb{E}[\nabla^3 f(z)]|$ in (24), we have

$$\left(\frac{2\beta}{\sigma^4} - \mathbb{E}_q[\nabla^4 f(z)]\right)\mu^2 + 2\mu\mathbb{E}_q[\nabla^3 f(z)] + \frac{\beta}{\sigma^2} \geq \left(\frac{2\beta}{\sigma^4} - \frac{L_2}{\sigma^2}\right)\mu^2 - \frac{2L_1}{\sigma^2}|\mu| + \frac{\beta}{\sigma^2}$$

holds for any $\mu \in \mathbb{R}$.

In order to show the RHS is non-negative for any $\mu \in \mathbb{R}$, we first assume $\mu \geq 0$ (similar results follow if $\mu \leq 0$). In this case, we only need to prove

$$(2\beta - L_2\sigma^2)\mu^2 - 2L_1\sigma^2\mu + \beta\sigma^2 \geq 0.$$

This is a quadratic function of $\mu$ with positive quadratic term coefficient (this is the first condition), hence a sufficient condition is that the discriminant is non-positive, that is,

$$4L_1^2\sigma^4 - 4\beta\sigma^2(2\beta - L_2\sigma^2) \leq 0. \tag{25}$$

This is again a quadratic function of $\beta$, and the inequality holds if we pick a sufficiently large $\beta$. More specifically, the larger root of (25) is

$$\beta^* = \frac{L_2\sigma^2 + \sqrt{L_2^2\sigma^4 + 8L_1^2\sigma^2}}{4} \leq \frac{L_2\sigma^2 + L_2\sigma^2}{4} + \frac{2\sqrt{2}L_1\sigma^2}{4} \leq \frac{D}{2}L_2 + \frac{\sqrt{2D}}{2}L_1 = \beta.$$

Therefore, $\beta$ satisfies the inequality (25).

3. We again use the upper bound of $|\mathbb{E}_q[\nabla^4 f(z)]|$ and $|\mathbb{E}_q[\nabla^3 f(z)]|$ in (24),

$$\frac{\beta^2}{2\sigma^6} - \frac{\mathbb{E}_q[\nabla^4 f(z)]\beta}{4\sigma^2} - \frac{(\mathbb{E}_q[\nabla^3 f(z)])^2}{4} \geq \frac{\beta^2}{2\sigma^6} - \frac{L_2\beta}{4\sigma^4} - \frac{L_1^2}{4\sigma^4}.$$

In fact, RHS is non-negative if and only if (25) holds.

The proof of relative weak convexity is similar. $\qquad\square$

**Remark.** In order to obtain a relative strong convexity guarantee of the objective $\ell(\omega)$, since the KL divergence term in (6) is 1-relatively convex, we need to guarantee that $-\frac{D}{2}L_2 - \frac{\sqrt{2D}}{2}L_1 > -1$.

As a result, under the assumptions of Corollary F.3.1, we conclude that $\mathbb{E}_q[f(z)]$ is relatively smooth with respect to $A^*$ with parameter

$$L = \frac{D}{2}L_2 + \frac{\sqrt{2D}}{2}L_1.$$

**Example 3.** Below are some examples where the likelihood $p(\mathcal{D} \mid z) = p(y \mid x, z)$ for $\mathcal{D} = \{(x, y)\}$ is specified:

**Linear Bayesian Regression.** In linear Bayesian regression where $p(y \mid x, z) = \mathcal{N}(xz, \sigma^2)$, it's easy to see that $-\log p(y \mid x, z)$ is a quadratic function in $z$, hence the necessary and sufficient conditions in Theorem F.1 and Theorem F.2 are satisfied with $\alpha = \beta = 0$. Therefore, the negative ELBO is relatively smooth with respect to $A^*$ with parameter 1, and is also relatively strongly convex with parameter 1. This coincides with the result in [WG24] when the model is conjugate.

**Logistic Regression.** For logistic regression, let $f(z) = -\log p(y \mid x, z) = \log(1 + e^{-xyz})$, then it can be shown that $|\nabla f(z)| \leq |x|$, $|\nabla^2 f(z)| \leq \frac{x^2}{4}$ for all $z \in \mathbb{R}$. Therefore, we can set $\sigma^2 \leq D$ and $\mathbb{E}_q[f(z)]$ is relatively smooth with parameter $L = \frac{x^2 D}{8} + \frac{\sqrt{2D}|x|}{2}$ according to Corollary F.3.1.

**Poisson Regression** In Poisson regression, we write $f(z) = -\log p(y \mid x, z) = e^{zx} - xyz + c$ for $y \in \mathbb{N}_+$ and some constant $c$. Then $\mathbb{E}_q[\nabla^4 f(z)] = x^4 e^{\mu x + \sigma^2 x^2/2}$. In order to satisfy the first necessary and sufficient condition $\sigma^4 \mathbb{E}_q[\nabla^4 f(z)] \leq 2\beta$ in the necessary and sufficient conditions Theorem F.1, we have to upper-bound $|x|, |\mu|$ and $\sigma^2$. Therefore, it is in general difficult to prove relative smoothness for Poisson regression without further assumptions (e.g., bounded domain).

# G  Missing Proofs in Section 3

## G.1  Proof of Theorem 3.2: Sufficient Conditions for Relative Smoothness

We begin with two lemmas which compute the Hessian matrices $\nabla^2 \mathbb{E}_{q(z;\omega)}[f(z)]$ and $\nabla^2 A^*(\omega)$.

**Lemma G.1.** *Let $f \in C^4(\mathbb{R}^d)$, then for $\omega = (\xi_1, \Xi_{11}, \cdots, \xi_d, \Xi_{dd}) \in \mathbb{R}^{2d}$, we have for $p, q \in \{1, \cdots, d\}$,*

$$(\nabla^2 \mathbb{E}_{q(z;\omega)}[f(z)])_{ij} = \begin{cases} -2B_{pp}\mu_p + C_{pp}\mu_p^2, & i = j = 2p-1, \\ \frac{1}{4}C_{pp}, & i = j = 2p, \\ \frac{1}{2}(B_{pp} - C_{pp}\mu_p), & (i,j) = (2p-1, 2p) \\ & \quad \text{or } (i,j) = (2p, 2p-1), \\ H_{pq} - B_{pq}\mu_q - B_{qp}\mu_p + C_{pq}\mu_p\mu_q, & (i,j) = (2p-1, 2q-1), p \neq q, \\ \frac{1}{2}(B_{pq} - C_{pq}\mu_p), & (i,j) = (2p-1, 2q) \\ & \quad \text{or } (i,j) = (2p, 2q-1), p \neq q, \\ \frac{1}{4}C_{pq}, & (i,j) = (2p, 2q), p \neq q. \end{cases}$$

*where $H_{ij} = \mathbb{E}_q[\nabla^2_{ij}f(z)]$, $B_{ij} = \mathbb{E}_q[\nabla^3_{ijj}f(z)]$ and $C_{ij} = \mathbb{E}_q[\nabla^4_{iijj}f(z)]$.*

*Proof.* We apply Bonnet's and Price's gradients (Theorem 3.1), we can easily get the following partial derivatives

$$\nabla_{\xi_{ii}}\mathbb{E}[f(z)] = \mathbb{E}_q[\nabla_i f(z) - \nabla^2_{ii}f(z)\xi_i],$$

$$\nabla_{\Xi_{ii}}\mathbb{E}[f(z)] = \frac{1}{2}\mathbb{E}_q[\nabla^2_{ii}f(z)].$$

Next, we use Theorem 3.1 again to get for $i \neq j$,

$$\begin{aligned} \nabla^2_{\xi_i\xi_j}\mathbb{E}[f(z)] &= \nabla_{\xi_j}\mathbb{E}_q[\nabla_i f(z) - \nabla^2_{ii}f(z)\xi_i] \\ &= \mathbb{E}_q[\nabla^2_{ij}f(z) - \nabla^3_{ijj}f(z)\xi_j - \nabla^3_{iij}f(z)\xi_i + \nabla^4_{iijj}f(z)\xi_i\xi_j] \\ &= H_{ij} - B_{ij}\mu_j - B_{ij}\mu_i + C_{ij}\mu_i\mu_j. \end{aligned}$$

And for $i = j$,

$$\begin{aligned} \nabla^2_{\xi_i\xi_i}\mathbb{E}[f(z)] &= \nabla_{\xi_i}\mathbb{E}_q[\nabla_i f(z) - \nabla^2_{ii}f(z)\xi_i] \\ &= \mathbb{E}_q[\nabla^2_{ii}f(z) - \nabla^3_{iii}f(z)\xi_i - \nabla^3_{iii}f(z)\xi_i + \nabla^4_{iiii}f(z)\xi_i^2 - \nabla^2_{ii}f(z)] \\ &= -2B_{ii}\mu_i + C_{ii}\mu_i^2. \end{aligned}$$

Moreover, for either $i = j$ or $i \neq j$, we have

$$\nabla^2_{\xi_i\Xi_{jj}}\mathbb{E}[f(z)] = \nabla_{\Xi_{jj}}\mathbb{E}_q[\nabla_i f(z) - \nabla^2_{ii}f(z)\xi_i] = \frac{1}{2}\mathbb{E}_q[\nabla^3_{ijj}f(z) - \nabla^4_{iijj}f(z)\xi_i] = \frac{1}{2}(B_{ij} - C_{ij}\mu_i).$$

$$\nabla^2_{\Xi_{ii}\Xi_{jj}}\mathbb{E}[f(z)] = \frac{1}{2}\nabla_{\Xi_{jj}}\mathbb{E}_q[\nabla^2_{ii}f(z)] = \frac{1}{4}\mathbb{E}_q[\nabla^4_{iijj}f(z)] = \frac{1}{4}C_{ij}.$$

$\square$

**Lemma G.2.** *For mean field family $q_\omega(z)$, we have*

$$\nabla^2 A^*(\omega) = \mathrm{diag}(\nabla^2 A_1^*(\omega_1), \nabla^2 A_2^*(\omega_2), \cdots, \nabla^2 A_d^*(\omega_d)),$$

*where each $\nabla^2 A_i^*(\omega_i)$ is a $2 \times 2$ matrix corresponding to univariate case, i.e.,*

$$\nabla^2 A_i^*(\omega_i) = \frac{1}{\sigma_i^4}\begin{pmatrix} 2\mu_i^2 + \sigma_i^2 & -\mu_i \\ -\mu_i & \frac{1}{2} \end{pmatrix}.$$

*Proof.* From Section C.1 we know that

$$\nabla_\xi A^*(\omega) = (\Xi - \xi\xi^\top)^{-1}\xi = \Sigma^{-1}\mu,$$

$$\nabla_\Xi A^*(\omega) = -\frac{1}{2}(\Xi - \xi\xi^\top)^{-1} = -\frac{1}{2}\Sigma^{-1}.$$

Since the coordinates are independent, $\nabla^2_{\omega_i \omega_j} A^*(\omega) = 0$ for all $i \neq j$. Therefore, $\nabla^2 A^*(\omega)$ is a block diagonal matrix, and it's easy to show that each block

$$\nabla^2_{\omega_i \omega_i} A^*(\omega) = \nabla^2 A_i^*(\omega_i) = \frac{1}{\sigma_i^4} \begin{pmatrix} 2\mu_i^2 + \sigma_i^2 & -\mu_i \\ -\mu_i & \frac{1}{2} \end{pmatrix}.$$

$\square$

In order to prove the theorem, we still need a lemma to transform high-order derivative conditions to lower-order ones with Stein's Lemma F.3. The proof of this lemma is deferred to Section G.2.

**Lemma G.3.** *For $f \in C^2(\mathbb{R}^d)$, if $Z \sim \mathcal{N}(\mu, \Sigma)$ where $\Sigma$ is a diagonal matrix, we have that for any $i, j \in \{1, \cdots, d\}$,*

$$|\mathbb{E}[\nabla_{ij} f(Z)]| \leq \sigma_i^{-1} \sigma_j^{-1} \sup_z |f(z)|.$$

With the three lemmas above, we are ready to prove Theorem 3.2.

*Proof of Theorem 3.2.* First, we will show that with $\beta = \mathcal{O}(dD^2 U(L_1 + L_2 U) + dD^3(L_1 + L_2 U))$, $\nabla^2 \mathbb{E}_{q(z;\omega)}[f(z)] \preceq \beta \nabla^2 A^*(\omega)$ holds, i.e., $M(\beta) := \beta \nabla^2 A^*(\omega) - \nabla^2 \mathbb{E}_{q(z;\omega)}[f(z)]$ is positive semidefinite. According to Theorems G.1 and G.2, for $p, q \in \{1, \cdots, d\}$,

$$M(\beta)_{ij} = \begin{cases} \frac{\beta}{\sigma_p^4}(2\mu_p^2 + \sigma_p^2) + 2B_{pp}\mu_p - C_{pp}\mu_p^2, & i = j = 2p-1, \\ \frac{\beta}{2\sigma_p^4} - \frac{1}{4}C_{pp}, & i = j = 2p, \\ -\frac{\mu_p}{\sigma_p^4}\beta - \frac{1}{2}(B_{pp} - C_{pp}\mu_p), & (i,j) = (2p-1, 2p) \text{ or } (i,j) = (2p, 2p-1), \\ -H_{pq} + B_{pq}\mu_q + B_{qp}\mu_p - C_{pq}\mu_p\mu_q, & (i,j) = (2p-1, 2q-1), p \neq q, \\ -\frac{1}{2}(B_{pq} - C_{pq}\mu_p), & (i,j) = (2p-1, 2q) \\ & \text{or } (i,j) = (2p, 2q-1), p \neq q, \\ -\frac{1}{4}C_{pq}, & (i,j) = (2p, 2q), p \neq q. \end{cases}$$

(26)

Then, we will prove that with $\beta = \mathcal{O}(dD^2 U(L_1 + L_2 U) + dD^3(L_1 + L_2 U))$, $M$ is positive semidefinite. In the following, we will drop the dependence of $\beta$ for convenience.

Note that the assumptions in the theorem and Theorem G.3 imply that $|B_{pq}| \leq \sigma_q^{-2} \sup_z |\nabla_p f(z)| \leq DL_1$ and $|C_{pq}| \leq \sigma_q^{-2} \sup_z |\nabla_{pp}^2 f(z)| \leq DL_2$. It is obvious that $|H_{pq}| \leq L_2$.

Define matrix $\tilde{M} = M - \frac{2}{3}\text{diag}(M)$, where $\text{diag}(M)$ is the diagonal matrix by setting all non-diagonal entries of $M$ to 0. Then for any $v \in \mathbb{R}^{2d}$,

$$v^\top M v = \sum_{i=1}^{2d} \sum_{j=1}^{2d} v_i v_j M_{ij} = 2^{2-d} \sum_{B \in \mathcal{B}} \sum_{i \in B} \sum_{j \in B} v_i v_j \tilde{M}_{ij}$$

$$+ \sum_{p=1}^{d} \sum_{i=1}^{2} \sum_{j=1}^{2} v_{2p-2+i} v_{2p-2+j} M_{2p-2+i, 2p-2+j}.$$

(27)

Here $\mathcal{B}$ is the set of sets of cardinality $d$, which contain exactly one element of $\{1, 2\}$, one element of $\{3, 4\}$, ..., one element of $\{2d-1, 2d\}$. The set $\mathcal{B}$ has cardinality $2^d$. Equation (27) can be proved by comparing the coefficient of each term $v_i v_j M_{ij}$ on both sides. Therefore, to prove $v^\top M v \geq 0$ for any $v \in \mathbb{R}^{2d}$, we only need to show each term on the RHS is non-negative. In other words, we only need to show

- For any $B \in \mathcal{B}$, the matrix $\tilde{M}^B$ is PSD, where $\tilde{M}^B \in \mathbb{R}^{d \times d}$ is the submatrix of $\tilde{M}$ by choosing the $i$-th rows and columns where $i \in B$.

- For any $p \in \{1, \cdots, d\}$, the submatrices of $M$

$$M^{(p)} := \begin{pmatrix} M_{2p-1, 2p-1} & M_{2p-1, 2p} \\ M_{2p, 2p-1} & M_{2p, 2p} \end{pmatrix}$$

are PSD.

For the first set of conditions, we aim to find $\beta$ such that the matrices $\tilde{M}^B$ are diagonally dominant and the diagonal entries are positive. For positiveness, if $\beta \geq \mathcal{O}(dD^2U(L_1 + L_2U) + dD^3(L_1 + L_2U))$, then

$$M_{2p-1,2p-1} = \frac{\beta}{\sigma_p^4}(2\mu_p^2 + \sigma_p^2) + 2B_{pp}\mu_p - C_{pp}\mu_p^2$$
$$\geq \sigma_p^{-2}\beta - 2|B_{pp}||\mu_p| - |C_{pp}||\mu_p|^2$$
$$\geq D^{-1}\beta - 2DL_1U - DL_2U^2$$
$$\geq 0.$$

$$M_{2p,2p} = \frac{\beta}{2\sigma_p^4} - \frac{1}{4}C_{pp} \geq \frac{1}{2}D^{-2}\beta - \frac{1}{4}DL_2 \geq 0.$$

For diagonal dominance, we fix $i = 2p - 1$ for some $p \in 1, \cdots, d$, and show that for every $\tilde{M}^B$ containing the $i$-th row and $i$-th column, the diagonal element $\tilde{M}_{ii}^B$ is dominating in this matrix. Indeed, since the $2p$-th row does not appear in $\tilde{M}^B$, we only need to guarantee that $\tilde{M}_{2p-1,2p-1} \geq \sum_{i=1, i \notin \{2p-1,2p\}}^{2d} |\tilde{M}_{2p-1,i}|$:

$$\tilde{M}_{2p-1,2p-1} \geq \sum_{i=1, i \notin \{2p-1,2p\}}^{2d} |\tilde{M}_{2p-1,i}|$$
$$\iff \frac{\beta}{3\sigma_p^4}(2\mu_p^2 + \sigma_p^2) + \frac{2}{3}B_{pp}\mu_p - \frac{1}{3}C_{pp}\mu_p^2 \geq \sum_{i \neq k} |-H_{pi} + B_{pi}\mu_i + B_{ip}\mu_p - C_{pi}\mu_p\mu_i|$$
$$+ \frac{1}{2}|B_{pi} - C_{pi}\mu_p|$$
$$\impliedby \frac{\beta}{3\sigma_p^2} \geq d(L_2 + 2DL_1U + DL_2U^2) + \frac{d}{2}(DL_1 + DL_2U)$$
$$\impliedby \beta = \mathcal{O}(dD^2U(L_1 + L_2U)).$$

Now, applying the same reasoning for $i = 2p$, we obtain

$$\tilde{M}_{2p,2p} \geq \sum_{i=1, i \notin \{2p-1,2p\}}^{2d} |\tilde{M}_{2p,i}|$$
$$\iff \frac{\beta}{6\sigma_p^4} - \frac{1}{12}C_{pp} \geq \sum_{i \neq p} \frac{1}{2}|B_{ip} - C_{ip}\mu_i| + \frac{1}{4}|C_{pi}|$$
$$\impliedby \frac{\beta}{6\sigma_p^4} \geq \frac{d}{2}(DL_1 + DL_2U) + dDL_2$$
$$\impliedby \beta \geq \mathcal{O}(dD^3(L_1 + L_2U)).$$

The second set of conditions can be easily verified. Since we have proved the non-negativeness of diagonal entries, to prove PSD we only need to show that the determinant of $M^{(p)}$ is non-negative. For every fixed $p \in \{1, \cdots, d\}$ the determinant is given by

$$\left(\frac{\beta}{2\sigma_p^4}(2\mu_p^2 + \sigma_p^2) + 2B_{pp}\mu_p - C_{pp}\mu_p^2\right)\left(\frac{\beta}{2\sigma_p^4} - \frac{1}{4}C_{pp}\right) - \left[\frac{\mu_p}{\sigma_p^4}\beta - \frac{1}{2}(B_{pp} - C_{pp}\mu_p)\right]^2.$$

Let the determinant be non-negative and we get

$$\beta^2 \geq \mathcal{O}(D^2U(L_1 + L_2U))\beta + \mathcal{O}(D^3U(L_1 + L_2U)).$$

Since the larger root of $\beta^2 = a\beta + b$ ($a, b \geq 0$) is given by $\beta = \frac{-a+\sqrt{a^2+4b}}{2}$, the condition $\beta \geq a + \sqrt{b} = \frac{a}{2} + \frac{a+\sqrt{4b}}{2} \geq \frac{a+\sqrt{a^2+4b}}{2}$ guarantees that $\beta^2 \geq a\beta + b$ always holds. Therefore, the determinant is non-negative if $\beta \geq \mathcal{O}(D^2U(L_1 + L_2U) + \sqrt{D^3U(L_1 + L_2U)}) = \mathcal{O}(D^2U(L_1 + L_2U))$.

Clearly, $\beta = \mathcal{O}(dD^2U(L_1 + L_2U) + dD^3(L_1 + L_2U))$ satisfies all the conditions.

Finally, we need to prove that $\alpha = -\mathcal{O}(dD^2U(L_1 + L_2U) + dD^3(L_1 + L_2U))$ satisfies $\nabla^2 \mathbb{E}_{q(z;\omega)}[f(z)] \succeq \alpha \nabla^2 A^*(\omega)$, i.e., $\nabla^2 \mathbb{E}_{q(z;\omega)}[f(z)] - \alpha \nabla^2 A^*(\omega) = -M(\alpha) \succeq 0$. We can again choose $\alpha < 0$ and $|\alpha|$ so large that the matrix $-M(\alpha)$ is diagonally dominant with positive diagonal entries. Since the previous proof provides upper bounds on the absolute value of non-diagonal entries of $M$, the same bound also applies to the absolute value of non-diagonal entries of $-M(\alpha)$.

Therefore, using the same approach, one can show that $-M(\alpha) \succeq 0$ with $-\alpha$ taking the same value as $\beta$. As a result, $-\mathbb{E}_q[\log p(\mathcal{D} \mid z)] = \mathbb{E}_q[f(z)]$ is $L$-smooth relative to $A^*$ on $\tilde{\Omega}$, where $L = \mathcal{O}(dD^2U(L_1 + L_2U) + dD^3(L_1 + L_2U))$. $\qquad\square$

## G.2  Proof of Theorem G.3

We first restate Theorem G.3 below.

**Lemma.** For $f \in C^2(\mathbb{R}^d)$, if $Z \sim \mathcal{N}(\mu, \Sigma)$ where $\Sigma$ is a diagonal matrix, we have that for any $i, j \in \{1, \cdots, d\}$,
$$|\mathbb{E}[\nabla_{ij} f(Z)]| \leq \sigma_i^{-1} \sigma_j^{-1} \sup_z |f(z)|.$$

*Proof.* For $i = j$, we have
$$\begin{aligned}
\mathbb{E}[\nabla_{ii}^2 f(Z)] &= \int_{\mathbb{R}} q(z_i) \int_{\mathbb{R}^{d-1}} q(z_{-i} \mid z_i) \nabla_{ii}^2 f(z) \, dz_{-i} \, dz_i \\
&= \int_{\mathbb{R}} q(z_i) \int_{\mathbb{R}^{d-1}} q(z_{-i}) \nabla_{ii}^2 f(z_{-i}, z_i) \, dz_{-i} \, dz_i \\
&= \int_{\mathbb{R}} q(z_i) \nabla_{ii} \left( \int_{\mathbb{R}^{d-1}} q(z_{-i}) f(z_{-i}, z_i) \, dz_{-i} \right) dz_i \\
&= \mathbb{E}[g''(Z_i)],
\end{aligned}$$
where $g(Z_i)$ is a univariate function and
$$g(Z_i) := \int_{\mathbb{R}^{d-1}} q(z_{-i}) f(z_{-i}, Z_i) \, dz_{-i}, \qquad |g(Z_i)| \leq \sup_{z \in \mathbb{R}} |f(z)|.$$

Since $Z_i \sim \mathcal{N}(\mu_i, \sigma_i^2)$, we apply Stein's lemma (Theorem F.3) twice and we have
$$\begin{aligned}
\mathbb{E}[g''(Z_i)] &= \sigma_i^{-2} \mathbb{E}[g'(Z_i)(Z_i - \mu_i)] \\
&= \sigma_i^{-2} \mathbb{E}[g'(Z_i)(Z_i - \mu_i) + g(Z_i)] - \sigma_i^{-2} \mathbb{E}[g(Z_i)] \\
&= \sigma_i^{-4} \mathbb{E}[g(Z_i)(Z_i - \mu_i)^2] - \sigma_i^{-2} \mathbb{E}[g(Z_i)] \\
&= \sigma_i^{-2} \mathbb{E}[g(Z_i)(X_i^2 - 1)],
\end{aligned}$$
where $X_i := (Z_i - \mu_i)/\sigma_i \sim \mathcal{N}(0, 1)$. Using Lemma G.4 and we get the following bound on $\mathbb{E}[\nabla_{ii}^2 f(Z)]$:
$$\mathbb{E}[\nabla_{ii}^2 f(Z)] = \mathbb{E}[g''(Z_i)] \leq \sigma_i^{-2} \mathbb{E}[g(Z_i)(X_i^2 - 1)] \leq \sigma_i^{-2} \sup_z |f(z)| \mathbb{E}|X_i^2 - 1| \leq \sigma_i^{-2} \sup_z |f(z)|.$$

For $i \neq j$, we define $g(Z) = (Z_j - \mu_j)f(Z)$ and use Theorem F.3 to get
$$\mathbb{E}[f(Z)(Z_j - \mu_j)(Z - \mu)] = \mathbb{E}[g(Z)(Z - \mu)] = \Sigma \mathbb{E}[\nabla g(Z)].$$
Reading the $i$-th coordinate gives
$$\mathbb{E}[f(Z)(Z_j - \mu_j)(Z_i - \mu_i)] = \mathbb{E}[g(Z)(Z_i - \mu_i)] = \sigma_i^2 \mathbb{E}[\nabla_i g(Z)] = \sigma_i^2 \mathbb{E}[(Z_j - \mu_j)\nabla_i f(Z)]. \quad (28)$$
Then we define $h(Z) = \nabla_i f(Z)$ and Theorem F.3 gives
$$\mathbb{E}[h(Z)(Z - \mu)] = \Sigma \mathbb{E}[\nabla h(Z)].$$
Reading the $j$-th coordinate gives
$$\mathbb{E}[(Z_j - \mu_j)\nabla_i f(Z)] = \mathbb{E}[h(Z)(Z_j - \mu_j)] = \sigma_j^2 \mathbb{E}[\nabla_j h(Z_j)] = \sigma_j^2 \mathbb{E}[\nabla_{ij}^2 f(Z)]. \quad (29)$$

Combining (28) and (29) and we have the following bound of $\mathbb{E}[\nabla_{ij} f(Z)]$:

$$\mathbb{E}[\nabla_{ij} f(Z)] = \sigma_i^{-1}\sigma_j^{-1}\mathbb{E}[f(Z)X_iX_j],$$

where $X_i := (Z_i - \mu_i)/\sigma_i$ and same for $X_j$. Note that $X_i, X_j$ are independent because we assume $\Sigma$ is diagonal. Finally, we use the standard result that $\mathbb{E}|X| \le 1$ for standard Gaussian variable $X$ to get that

$$\mathbb{E}[\nabla_{ij} f(Z)] = \sigma_i^{-1}\sigma_j^{-1}\mathbb{E}[f(Z)X_iX_j] \le \sigma_i^{-1}\sigma_j^{-1}\sup_z |f(z)|\mathbb{E}|X_i|\mathbb{E}|X_j| \le \sigma_i^{-1}\sigma_j^{-1}\sup_z |f(z)|.$$

By symmetry, $-\mathbb{E}[\nabla_{ij}^2 f(Z)] \le \sigma_i^{-1}\sigma_j^{-1}\sup_z |f(z)|$ holds as well. $\square$

**Lemma G.4.** *Let* $X \sim \mathcal{N}(0,1)$*, then*

$$\mathbb{E}[|X^2 - 1|] = 2\sqrt{\frac{2}{\pi e}} \le 1.$$

*Proof.* First, we have

$$\mathbb{E}[|X^2 - 1|] = \mathbb{E}[X^2 - 1] + 2\mathbb{E}[\mathbf{1}_{|X|\le 1}(1 - X^2)] = 4\mathbb{E}[\mathbf{1}_{X\in[0,1]}(1 - X^2)].$$

Note that

$$\frac{\mathrm{d}}{\mathrm{d}x}\left(xe^{-\frac{x^2}{2}}\right) = (1 - x^2)e^{-\frac{x^2}{2}},$$

then we can compute the expectation in closed form

$$\mathbb{E}[\mathbf{1}_{X\in[0,1]}(1 - X^2)] = \int_0^1 \frac{1}{\sqrt{2\pi}}(1 - x^2)e^{-x^2/2}\,\mathrm{d}x = \frac{1}{\sqrt{2\pi}}xe^{-\frac{x^2}{2}}\Big|_0^1 = \frac{1}{\sqrt{2\pi e}}.$$

Hence $\mathbb{E}[|X^2 - 1|] = 4\mathbb{E}[\mathbf{1}_{X\in[0,1]}(1 - X^2)] = 2\sqrt{\frac{2}{\pi e}} \le 1$. $\square$

### G.3 Proof of Proposition 2: Hidden Convexity of $\ell(\omega)$

We define $\Theta := \{(\mu, C) : C \in \mathcal{S}_+^d \text{ and is diagonal}\}$. This set corresponds to the Cholesky parameterization of the expectation parameters contained in $\Omega$. Furthermore, we define $c : \Omega \to \Theta$ to be the one-to-one map from expectation parameter to Cholesky parameter of the same distribution.

Expectation parameters, $\Omega$  Cholesky parameters, $\Theta$

mapping $c(\omega)$

$$\omega = (\mu,\ \Sigma + \mathrm{diag}(\mu \odot \mu)) \qquad\longrightarrow\qquad c(\omega) = (\mu,\ C),$$
$$\text{where } CC^\top = \Sigma$$

For bounded expectation parameter domain $\tilde{\Omega}$, we also define its counterpart in Cholesky parameterization $\tilde{\Theta} := \{(\mu, C) : C \text{ is diagonal and } D^{-1} \le C_{ii}^2 \le D,\ \forall\, 1 \le i \le d\}$. With a slight abuse of notation, we also use $c$ to denote the one-to-one map from $\tilde{\Omega}$ to $\tilde{\Theta}$.

In the following, we will verify that the negative ELBO $\ell(\omega)$ is hidden convex as defined in Definition 3.2 on the bounded domain $\tilde{\Omega}$.

*Proof of Proposition 2.* Define $L : \tilde{\Theta} \to \mathbb{R}$,

$$L(\theta) := \ell(c(\omega)) = \mathbb{E}_{q(z;\theta)}[-\log p(y\,|\,z) - \log p(z)] + \mathbb{E}_{q(z;\theta)}[\log q(z;\theta)], \tag{30}$$

where $q(z;\theta) = q(z;\mu, C) = \mathcal{N}(\mu, CC^\top)$ is the Gaussian vector with Cholesky parameterization.

**Proof of the First Property in Definition 3.2.** We first verify the strong convexity of $L$ on $\tilde{\Theta}$. It is obvious that the domain $\tilde{\Theta}$ is convex. For the second term in (30), we note that it is simply the negative entropy of Gaussian distribution:

$$\mathbb{E}_{q(z;\theta)}[\log q(z;\theta)] = -\frac{d}{2}(1 + \log(2\pi)) - \frac{1}{2}\log\det(CC^\top) = Const - \sum_{i=1}^{d}\log C_{ii}.$$

Since each $-\log C_{ii}$ is convex in $C_{ii}$, the entropy is convex in $C$. Then we can conclude that the second term $\mathbb{E}_{q_\theta(z)}[\log q_\theta(z)]$ is convex in $\theta$.

For the first term of (30), we apply Theorem 9 of [Dom20], which states that the $\mu_H$-strong convexity of $f(z)$ for some function $f$ implies the $\mu_H$-strong convexity of $\mathbb{E}_{q(z;\theta)}[f(z)]$ w.r.t. the Cholesky parameter $\theta$. Therefore, we only need to prove the strong convexity of $-\log p(y \mid z) - \log p(z)$ in $z$. Indeed, $-\log p(y \mid z) - \log p(z)$ is 1-strongly convex in $z$ because $\log p(y \mid z)$ is concave in $z$ and

$$-\log p(z) = \frac{d}{2}\log(2\pi) + \frac{1}{2}z^\top z$$

is 1-strongly convex in $z$.

Therefore, $L(\theta)$, as the sum of a 1-strongly convex function (first term) and a convex function (second term), is 1-strongly convex. Then we conclude that the first property in Definition 3.2 is satisfied with $\mu_H = 1$.

**Proof of the Second Property in Definition 3.2.** Now we compute $\mu_c$ by computing the Lipschitz coefficient of the inverse map $c^{-1}$ on $\tilde{\Theta}$. We reparameterize $\omega$ and $\theta$ as vectors: $\omega = (\xi_1, \Xi_{11}, \cdots, \xi_d, \Xi_{dd}) \in \mathbb{R}^{2d}$ and $\theta = (\mu_1, C_{11}, \cdots, \mu_d, C_{dd}) \in \mathbb{R}^{2d}$, respectively. Note that $\omega = c^{-1}(\theta) = (\mu_1, \mu_1^2 + C_{11}^2, \cdots, \mu_d, \mu_d^2 + C_{dd}^2)$, and $\nabla c^{-1}(\theta) = \text{diag}(G_1, \cdots, G_d) \in \mathbb{R}^{2d \times 2d}$, where

$$G_i := \begin{pmatrix} 1 & 0 \\ 2\mu_i & 2C_{ii} \end{pmatrix}.$$

For each block $G_i$, the largest singular value of $G_i$ is bounded by

$$\sigma_{\max}(G_i) = \sqrt{\lambda_{\max}(G_iG_i^\top)} \leq \sqrt{\text{Tr}(G_iG_i^\top)} \leq \sqrt{4U^2 + 4D + 1}.$$

The last inequality above holds since

$$G_iG_i^\top = \begin{pmatrix} 1 & 2\mu_i \\ 2\mu_i & 4\mu_i^2 + 4C_{ii}^2 \end{pmatrix},$$

and $\text{Tr}(G_iG_i^\top) = 1 + 4\mu_i^2 + 4C_{ii}^2 \leq 4U^2 + 4D + 1$ using the fact that $(\mu, C) \in \tilde{\Theta}$.

Hence we conclude that $\|\nabla c^{-1}(\theta)\| \leq \sqrt{4U^2 + 4D + 1}$, and $c^{-1}(\cdot)$ is $\sqrt{4U^2 + 4D + 1}$-Lipschitz continuous. Then the second property is satisfied with $\mu_C = (4U^2 + 4D + 1)^{-1/2}$. $\qquad\square$

# H    Missing Proofs in Section 4

In this section, we will prove the convergence guarantees of Proj-SNGD in Section 4. We first prove an auxiliary result of smoothness and strong convexity of $A^*$ in the Euclidean geometry in Section H.1. Next, we prove Theorem 4.1 and Theorem 4.2 in Section H.2 and Section H.3, respectively.

## H.1    Smoothness and Strong Convexity of $A^*$

In this subsection, we will establish the smoothness and strong convexity properties of $A^*$ in Euclidean geometry in the bounded set

$$\tilde{\Omega} = \{(\xi, \Xi) : |\xi_i| \leq U, \Xi - \mathrm{diag}(\xi \odot \xi) \text{ is diagonal}, D^{-1} \leq (\Xi - \mathrm{diag}(\xi \odot \xi))_{ii} \leq D, \forall 1 \leq i \leq d\}.$$

It is obvious that $A^*$ is globally 1-strongly convex in the non-Euclidean geometry induced by $A^*$ itself. However, it is only globally strictly convex in the Euclidean geometry, and strongly convexity only holds in a bounded domain of $\Omega$. In order to transform the PL inequality in Euclidean geometry (10) to the PL inequality in non-Euclidean geometry (see Section H.3), such smooth and strong convexity results are necessary.

**Lemma H.1.** $A^*$ is strongly convex with parameter $C_S := (D(4U^2 + 2D + 1))^{-1}$ with respect to Euclidean norm. Moreover, it is smooth with parameter $C_L := \frac{9U^2 D^2}{2}$.

*Proof.* As shown in Lemma G.2,

$$\nabla^2 A^*(\omega) = \mathrm{diag}(\nabla^2 A_1^*(\omega_1), \cdots, \nabla^2 A_d^*(\omega_d)),$$

$$\nabla^2 A_i^*(\omega_i) = \frac{1}{\sigma_i^4} \begin{pmatrix} 2\mu_i^2 + \sigma_i^2 & -\mu_i \\ -\mu_i & \frac{1}{2} \end{pmatrix}, \quad \text{for } 1 \leq i \leq d.$$

For each block $\nabla^2 A_i^*(\omega_i)$, its smaller eigenvalue satisfies

$$\lambda_{\min}(\nabla^2 A_i^*(\omega_i)) \geq \frac{\det(\nabla^2 A_i^*(\omega_i))}{\mathrm{Tr}(\nabla^2 A_i^*(\omega_i))} = \frac{1}{\sigma_i^2(4\mu_i^2 + 2\sigma_i^2 + 1)} \geq \frac{1}{D(4U^2 + 2D + 1)}.$$

Since $\nabla^2 A^*(\omega)$ is a block diagonal matrix, we conclude that $\lambda_{\min}(\nabla^2 A^*(\omega)) \geq \frac{1}{D(4U^2+2D+1)}$, hence

$$\nabla^2 A^*(\omega) \succeq \frac{1}{D(4U^2 + 2D + 1)} I.$$

For the smoothness part, we solve for the larger eigenvalue directly to get

$$
\begin{aligned}
\lambda_{\max}(\nabla^2 A_i^*(\omega_i)) &= \frac{2\mu_i^2 + \sigma_i^2 + 0.5 + \sqrt{(2\mu_i^2 + \sigma_i^2 - 0.5)^2 + 4\mu_i^2}}{2\sigma_i^4} \\
&\leq \frac{2\mu_i^2 + \sigma_i^2 + 0.5 + |2\mu_i^2 + \sigma_i^2 - 0.5| + 2|\mu_i|}{2\sigma_i^4} \\
&\leq \frac{2\mu_i^2 + \sigma_i^2 + 0.5 + 2\mu_i^2 + \sigma_i^2 + 0.5 + 2|\mu_i|}{2\sigma_i^4} \\
&\leq 2U^2 D^2 + D + \frac{1}{2}D^2 + U D^2 \\
&\leq \frac{9}{2} U^2 D^2,
\end{aligned}
$$

where we used $\sqrt{a + b} \leq \sqrt{a} + \sqrt{b}$ in the first inequality, and $2\mu_i^2 + \sigma_i^2 > 0$ in the second inequality. Then we conclude that $A^*$ is smooth with parameter $C_L := \frac{9}{2}U^2 D^2$.    $\square$

## H.2    Proof of Theorem 4.1: Convergence of Proj-SNGD

We first introduce the standard assumption on gradient variance, which is widely used in optimization.

**Assumption 3.** *For all $\omega_t \in \tilde{\Omega}$, there exists $V \geq 0$ such that*

$$\mathbb{E}[\|\hat{\nabla}\ell(\omega_t) - \nabla\ell(\omega_t)\|^2] \leq V^2. \tag{31}$$

We also define the Bregman Moreau envelope of function $\ell$ as

$$\ell_{1/\rho}(\omega) := \min_{\omega' \in \Omega}[\ell(\omega') + \rho D_{A^*}(\omega', \omega)].$$

In order to prove Theorem 4.1, we will rely on the following theorem on the convergence of SMD under relative smoothness condition. Note that this theorem assumes Assumption 3.

**Theorem H.2.** *[Theorem 4.3 in [FH24]] Suppose $A^*$ is 1-strongly convex and $\ell$ is smooth with respect to $A^*$ with parameter $L$. Let $\{\gamma_t\}_{0 \leq t \leq T-1}$ be non-increasing with $\gamma_0 \leq 1/(2L)$. For $\{\omega_i\}_{i=0}^{T-1}$ generated by (32), let $\bar{\omega}_T$ be randomly chosen from $\omega_0, \cdots, \omega_{T-1}$ with probability $p_t = \tilde{\gamma}_t / \sum_{i=0}^{n-1} \tilde{\gamma}_i$, then under Assumption 3, we have*

$$\mathbb{E}[\mathcal{E}_{3L}(\bar{\omega}_T)] \leq \frac{3\lambda_0 + 6LV^2 \sum_{t=0}^{T-1} \gamma_t^2}{\sum_{t=0}^{T-1} \gamma_t}$$

*where $\lambda_0 = 2(\ell(\omega_0) - \ell^*)$ and $\ell^* := \inf_{\omega \in \tilde{\Omega}}(\omega)$. If we use constant step size $\gamma_t = \min\left\{\frac{1}{2L}, \sqrt{\frac{\lambda_0}{V^2 LT}}\right\}$ then*

$$\mathbb{E}[\mathcal{E}_{3L}(\bar{\omega}_T)] \leq 18\frac{L\lambda_0}{T} + 9\sqrt{\frac{LV^2\lambda_0}{T}}.$$

Note that $\ell_{1/\rho}(\omega) \leq \ell(\omega)$ for all $\omega \in \Omega$. Thus $\tilde{\lambda}_0 \leq 2(\ell(\omega_0) - \ell^*)$. Theorem 4.1 is essentially the direct result of Theorem H.2 except two minor differences:

1.  Instead of the update rules of Proj-SNGD as defined in (11), Theorem H.2 assumes a more direct update rule

    $$\omega_{t+1} = \operatorname*{argmin}_{\omega \in \tilde{\Omega}} \gamma_t \langle \hat{\nabla}\ell(\omega_t), \omega \rangle + D_{A^*}(\omega, \omega_t). \tag{32}$$

    We will prove the equivalence of two expression in Theorem H.3.

2.  Theorem H.2 considers a different assumption on gradient variance, and Theorem H.2 assumes 1-strong convexity of the distance generating function $A^*$. In Proposition 4 we will show that the same result holds also under Assumption 1, and that 1-strong convexity assumption can be removed with this new assumption.

**Lemma H.3.** *Let $\tilde{\omega}_{t+1}$ be the output of Proj-SNGD as in (11), and let $\omega_{t+1}$ be the direct update rule as in (32). Then $\tilde{\omega}_{t+1} = \omega_{t+1}$. Moreover, (11) can be performed entry-wise in $\mathcal{O}(d)$ time.*

*Proof.* Recall that in the update rule of Proj-SNGD as in (11), we define $\omega_{t+1,*} \in \Omega$ such that

$$\nabla A^*(\omega_{t+1,*}) = \nabla A^*(\omega_t) - \gamma_t \hat{\nabla}\ell(\omega_t). \tag{33}$$

By the duality of SMD, $\omega_{t+1,*}$ is also the optimal solution without constraints, i.e.,

$$\omega_{t+1,*} = \operatorname*{argmin}_{\omega \in \Omega} \gamma_t \langle \hat{\nabla}\ell(\omega_t), \omega \rangle + D_{A^*}(\omega, \omega_t).$$

Use the definition of $\omega_{t+1}$ and (33), we have

$$\begin{aligned}
\omega_{t+1} &= \operatorname*{argmin}_{\omega \in \tilde{\Omega}} \gamma_t \langle \hat{\nabla}\ell(\omega_t), \omega \rangle + D_{A^*}(\omega, \omega_t) \\
&= \operatorname*{argmin}_{\omega \in \tilde{\Omega}} \langle \nabla A^*(\omega_t) - \nabla A^*(\omega_{t+1,*}), \omega \rangle + A^*(\omega) - A^*(\omega_t) - \langle \nabla A^*(\omega_t), \omega - \omega_t \rangle \\
&= \operatorname*{argmin}_{\omega \in \tilde{\Omega}} D_{A^*}(\omega, \omega_{t+1,*}) + Const \\
&= \operatorname*{argmin}_{\omega \in \tilde{\Omega}} D_{A^*}(\omega, \omega_{t+1,*}) + Const.
\end{aligned}$$

Therefore, to find $\omega_{t+1}$, we only need to compute $\omega_{t+1,*}$ and then project it onto $\tilde{\Omega}$ under the geometry induced by $A^*$. Next we show that $\tilde{\omega}_{t+1}$ solves the optimization problem above. Use the fact that for mean-field parameterization (see Section C.1),

$$\begin{cases} \nabla_\xi A^*(\xi, \Xi) = (\Xi - \mathrm{diag}(\xi \odot \xi))^{-1}\xi, \\ \nabla_\Xi A^*(\xi, \Xi) = -\frac{1}{2}(\Xi - \mathrm{diag}(\xi \odot \xi))^{-1}, \end{cases}$$

in the standard parameter space we have

$$\begin{aligned}
(\mu_{t+1}, \Sigma_{t+1}) =& \operatorname*{argmin}_{(\mu,\Sigma)\in\tilde{\mathcal{P}}} -\frac{1}{2}\log\det(\Sigma) + \frac{1}{2}\log\det(\Sigma_{t+1,*}) - \langle \Sigma_{t+1,*}^{-1}\mu_{t+1,*}, \mu \rangle \\
&+ \frac{1}{2}\langle \Sigma_{t+1,*}^{-1}, \Sigma + \mathrm{diag}(\mu \odot \mu)\rangle \\
=& \operatorname*{argmin}_{(\mu,\Sigma)\in\tilde{\mathcal{P}}} \sum_{i=1}^{d}\left[ -\frac{1}{2}\log\Sigma_{ii} - (\Sigma_{t+1,*})_{ii}^{-1}(\mu_{t+1,*})_i\mu_i + \frac{1}{2}(\Sigma_{t+1,*})_{ii}^{-1}(\Sigma_{ii} + \mu_i^2)\right].
\end{aligned}$$
$$(34)$$

The last equality holds due to the mean field assumption. Therefore, we can solve $\omega_{t+1}$ by optimizing over $(\mu_i, \Sigma_{ii})$ independently.

For each entry $i$, (34) is a quadratic function in $\mu_i$ with positive quadratic term and the (unconstrained) minimum is attained at $\mu_i = (\mu_{t+1,*})_i$. If $(\mu_{t+1,*})_i < -U$, (34) is an increasing function on $[-U, U]$, and if $(\mu_{t+1,*})_i > U$, the function is decreasing on $[-U, U]$. Therefore, the minimum on $[-U, U]$ is always attained at $(\mu_{t+1})_i = \mathrm{clip}_{[-U,U]}((\mu_{t+1,*})_i)$.

Similarly, (34) is a convex function in $\Sigma_{ii}$ and the (unconstrained) minimum is attained at $\Sigma_{ii} = (\Sigma_{t+1,*})_{ii}$. It's also straightforward to check that the minimizer is given by $(\Sigma_{t+1})_{ii} = \mathrm{clip}_{[D^{-1},D]}((\Sigma_{t+1,*})_{ii})$. The closed-form solution of $(\mu_{t+1}, \Sigma_{t+1})$ coincides with the standard projection of $(\mu_{t+1,*}, \Sigma_{t+1,*})$ onto $\tilde{\mathcal{P}}$ under Euclidean geometry. By transforming $(\mu, \Sigma)$ back to expectation parameter space, we conclude that $\tilde{\omega}_{t+1} = \omega_{t+1}$.

Since the computation of $\nabla A^*(\cdot)$ and its inverse $(\nabla A^*(\cdot))^{-1}$ involves only entry-wise calculation (see Section C.1), one can also compute

$$\omega_{t+1,*} = (\nabla A^*)^{-1}(\nabla A^*(\omega_t) - \tilde{\gamma}_t\hat{\nabla}\ell(\omega_t))$$

in $\mathcal{O}(d)$ time. Finally, the transformation between $(\mu, \Sigma)$ and $(\xi, \Xi)$, and the projection both require $\mathcal{O}(d)$ time. Therefore, $\omega_{t+1}$ can be calculated in $\mathcal{O}(d)$ time. This completes the proof. $\qquad\square$

**Proposition 4.** *The results of Theorem H.2 also hold if Assumption 3 is replaced by Assumption 1, without requiring $A^*$ to be 1-strongly convex.*

*Proof.* For $t \geq 0$, define

$$\tilde{\lambda}_t := \ell_{1/\rho}(\omega_t) - \ell^* + \gamma_{t-1}\rho(\ell(\omega_t) - \ell^*).$$

In step II (one step progress on the Lyapunov function), formula (15) of the proof in [FH24], we have

$$\begin{aligned}
\tilde{\lambda}_{t+1} \leq& \tilde{\lambda}_t - \gamma_t\rho(\rho - L)D_{A^*}(\hat{\omega}_t, \omega_t) - \frac{\gamma_t\rho}{2(\rho+L)}\mathcal{E}_{\rho+L}(\omega_t) + \rho\gamma_t\langle\hat{\nabla}\ell(\omega_t) - \nabla\ell(\omega_t), \hat{\omega}_t - \omega_t\rangle \\
&+ \rho\gamma_t\langle\hat{\nabla}\ell(\omega_t) - \nabla\ell(\omega_t), \omega_t - \omega_{t+1}\rangle - \rho(1 - \gamma_t L)D_{A^*}(\omega_{t+1}, \omega_t) \\
=& \tilde{\lambda}_t - \gamma_t\rho(\rho - L)D_{A^*}(\hat{\omega}_t, \omega_t) - \frac{\gamma_t\rho}{2(\rho+L)}\mathcal{E}_{\rho+L}(\omega_t) + \rho\gamma_t\langle\hat{\nabla}\ell(\omega_t) - \nabla\ell(\omega_t), \hat{\omega}_t - \omega_{t+1}^+\rangle \\
&+ \rho\gamma_t\langle\hat{\nabla}\ell(\omega_t) - \nabla\ell(\omega_t), \omega_{t+1}^+ - \omega_{t+1}\rangle - \rho(1 - \gamma_t L)D_{A^*}(\omega_{t+1}, \omega_t)
\end{aligned}$$

here we define $\hat{\omega} := \mathrm{prox}_{l/\rho}(\omega)$.

Note that $\mathbb{E}[\langle\hat{\nabla}\ell(\omega_t) - \nabla\ell(\omega_t), \hat{\omega}_t - \omega_{t+1}^+\rangle \mid \omega_t] = 0$ because the $\hat{\omega}_t - \omega_{t+1}^+$ has no randomness given $\omega_t$, and the gradient estimator is unbiased. Moreover, by bounded variance assumption (12),

$\mathbb{E}[\langle \hat{\nabla}\ell(\omega_t) - \nabla\ell(\omega_t), \omega_{t+1}^+ - \omega_{t+1}\rangle \mid \omega_t] \leq \gamma_t V^2$. Therefore, we use the same assumptions as in the proof that $\rho = 2L$ and $\gamma_t \leq 1/(2L)$ and we have

$$\mathbb{E}[\tilde{\lambda}_{t+1} \mid \omega_t] \leq \tilde{\lambda}_t - \gamma_t \rho(\rho - L)D_{A^*}(\hat{\omega}_t, \omega_t) - \frac{\gamma_t \rho}{2(\rho + L)}\mathcal{E}_{\rho+L}(\omega_t) + \rho\gamma_t^2 V^2$$
$$- \rho(1 - \gamma_t L)D_{A^*}(\omega_{t+1}, \omega_t)$$
$$\leq \tilde{\lambda}_t - \frac{\gamma_t}{3}\mathcal{E}_{3L}(\omega_t) + 2L\gamma_t^2 V^2.$$

This is the same formula as formula (17) in [FH24], where the last inequality holds due to the non-negativeness of Bregman divergence. Note that in the original proof of Theorem H.2, 1-strong convexity assumption is used only when deriving (17) from (15). However, in our new proof, we can arrive at (17) without invoking the strong convexity condition. Therefore, the same results hold following the proof of the theorem. $\qquad\square$

### H.3  Proof of Theorem 4.2: Fast Convergence of Proj-SNGD

To establish Theorem 4.2, we rely on Theorem 4.7 from [FH24] (restated as Theorem H.4 below). Before doing so, we introduce the Bregman Prox-PL condition, which serves as an assumption in Theorem H.4. It is similar to PL inequality with a difference that the gradient norm is replaced with BFBE, a non-Euclidean first-order stationarity measure. Recall that BFBE is defined in Definition 4.1 as

$$\mathcal{E}_\rho(\omega) := -2\rho \min_{\omega' \in \tilde{\Omega}}[\langle \nabla\ell(\omega), \omega' - \omega\rangle + \rho D_{A^*}(\omega', \omega)]. \tag{35}$$

**Assumption 4** (Bregman Prox-PL condition). *There exists some constant $\rho > 3L$ and $\mu_B > 0$ such that for all $\omega \in \tilde{\Omega}$,*

$$\mathcal{E}_\rho(\omega) \geq 2\mu_B(\ell(\omega) - \ell^*).$$

**Theorem H.4.** *[Theorem 4.7 in [FH24]] Suppose $A^*$ is 1-strongly convex and $\ell$ is smooth with respect to $A^*$ with parameter $L$. Let Assumptions 3 and 4 hold. For step size scheme*

$$\gamma_t = \begin{cases} \frac{1}{2L}, & \text{if } t \leq T/2 \text{ and } T \leq \frac{6L}{\mu_B}, \\ \frac{6}{\mu_B(t-\lceil T/2\rceil)+12L}, & \text{otherwise,} \end{cases} \tag{36}$$

*we have*

$$\min_{t \leq T-1} \mathbb{E}[\ell(\omega_{t,*}) - \ell^*] \leq \frac{192L\lambda_0}{\mu_B}\exp\left(-\frac{\mu_B T}{12L}\right) + \frac{648LV^2}{\mu_B^2 T}.$$

We observe that there are slight differences between Theorem 4.2 and Theorem H.4 in assumptions and upper bounds. Specifically, Theorem 4.2 requires Assumptions 1 and 2, whereas Theorem H.4 relies on Assumptions 3 and 4 together with the 1-strong convexity of $A^*$. In addition, Theorem 4.2 gives the last-iterate bound, while Theorem H.4 does not. In the first step, we will prove that the assumptions in Theorem H.4 can be derived from those required in Theorem 4.2. In the second step, we will prove a stronger last-iterate bound.

**Step 1.**  First, following the same approach as in Proposition 4, we can show that Assumption 3 can be replaced by Assumption 1, without requiring $A^*$ to be 1-strongly convex.

Second, we will prove that Assumption 4 is implied by the PL inequality (10) in $\tilde{\Omega}$ under Assumption 2, by proving that BFBE (non-Euclidean measure of stationarity) is greater than $\|\nabla\ell(\omega)\|^2$ (Euclidean measure of stationarity) up to a constant in the following proposition.

**Proposition 5** (PL implies Bregman Prox-PL under Assumption 2). *Under Assumption 2, for sufficiently large $\rho$ which may depend on $\omega$, it holds that*

$$\mathcal{E}_\rho(\omega) \geq \frac{1}{2C_L}\|\nabla\ell(\omega)\|^2, \quad \forall\omega \in \tilde{\Omega}.$$

*Together with (10), we conclude that Assumption 4 holds with parameter*

$$\mu_B = \frac{\mu_H \mu_C^2}{2C_L}.$$

**Remark.** In Theorem 4.2, we require that Proposition 5 holds for all iterates $\{\omega_t : 0 \le t \le T-1\}$. Since this set is finite, we can take the maximum value of $\rho$ corresponding to all $\omega_t$, ensuring that Proposition 5 holds uniformly along the entire trajectory.

The proof of Proposition 5 relies on another metric of first-order condition, which is closely related to BFBE:

**Definition H.1** (Bregman Gradient Mapping (BGM)). *For some $\rho > 0$, the BGM at $\omega \in \Omega$ is defined as*

$$\Delta_\rho^+(\omega) := \rho^2 D_{A^*}^{\mathrm{sym}}(\omega, \omega^+), \tag{37}$$

*where $D_{A^*}^{\mathrm{sym}}(\omega, \omega^+) := D_{A^*}(\omega, \omega^+) + D_{A^*}(\omega^+, \omega)$ is the symmetrized Bregman divergence, and*

$$\omega^+ = \operatorname*{argmin}_{\omega' \in \Omega} \langle \nabla \ell(\omega), \omega' \rangle + \rho D_{A^*}(\omega', \omega).$$

The following lemma from [FH24] reveals the connection between the two measures:

**Lemma H.5.** *[Lemma 4.2 in [FH24]] For any $\omega \in \Omega$ and any $\rho > 0$, we have*

$$2\mathcal{E}_{\rho/2}(\omega) \ge \Delta_\rho^+(\omega).$$

*Proof of Proposition 5.* Recall in Theorem H.1, we have shown that $A^*$ is $C_L$-smooth and $C_S$-strongly convex with respect to the Euclidean norm in the bounded set $\tilde{\Omega}$. By Lemma H.5 and smoothness of $A^*$, we obtain the following lower bound of BFBE

$$\begin{aligned}
\mathcal{E}_\rho(\omega) \ge& \frac{1}{2}\Delta_{2\rho}^+(\omega) \\
=& 2\rho^2 D_{A^*}^{\mathrm{sym}}(\omega, \omega^+) \\
=& 2\rho^2 \langle \nabla A^*(\omega) - \nabla A^*(\omega^+), \omega - \omega^+ \rangle \\
\ge& \frac{2\rho^2}{C_L} \|\nabla A^*(\omega) - \nabla A^*(\omega^+)\|^2,
\end{aligned} \tag{38}$$

where $\omega^+ = \operatorname*{argmin}_{\omega' \in \tilde{\Omega}} \langle \nabla \ell(\omega), \omega' \rangle + 2\rho D_{A^*}(\omega', \omega)$, following the definition in Definition H.1. Then, we can show that without domain constraint, $\omega_* := \operatorname*{argmin}_{\omega' \in \Omega} \langle \nabla \ell(\omega), \omega' \rangle + 2\rho D_{A^*}(\omega', \omega)$ satisfies

$$\nabla A^*(\omega_*) = \nabla A^*(\omega) - \frac{1}{2\rho} \nabla \ell(\omega). \tag{39}$$

By strong convexity of $A^*$, we have

$$\|\omega_* - \omega\| \le \frac{1}{C_S} \|\nabla A^*(\omega_*) - A^*(\omega)\| = \frac{1}{2\rho C_S} \|\nabla \ell(\omega)\|.$$

If $\omega^+ = \omega_*$ holds, we can plug (39) into the lower bound of BFBE (38) and we conclude that

$$\mathcal{E}_\rho(\omega) \ge \frac{2\rho^2}{C_L} \|\nabla A^*(\omega) - \nabla A^*(\omega^+)\|^2 = \frac{1}{2C_L} \|\nabla \ell(\omega)\|^2.$$

This completes the proof of Proposition 5. Therefore, it remains to prove $\omega^+ = \omega_*$.

For $\omega$ lying in the interior of $\tilde{\Omega}$, the condition holds for sufficient large $\rho$ as $\|\nabla \ell(\omega)\|$ is bounded on $\tilde{\Omega}$ and $2\rho D_{A^*}(\omega', \omega)$ dominates in the optimization problem. The strong convexity of $D_{A^*}$ then ensures that $\|\omega_* - \omega\|$ is very small.

For $\omega \in \partial\tilde{\Omega}$, we will need Assumption 2. Since $\nabla A(\eta)$ and $\nabla A^*(\omega)$ are inverse operators of one another (see Section C.1), (39) implies

$$\begin{aligned}
\omega_* &= (\nabla A^*)^{-1} \left( \nabla A^*(\omega) - \frac{1}{2\rho} \nabla \ell(\omega) \right) \\
&= \omega - \frac{1}{2\rho} \nabla^2 A(\eta) \nabla \ell(\omega) + o(1/\rho) \\
&= \omega - \frac{1}{2\rho} (\nabla^2 A^*(\omega))^{-1} \nabla \ell(\omega) + o(1/\rho)
\end{aligned}$$

as $\rho \to \infty$.

Then under Assumption 2, for any $\omega \in \partial\tilde{\Omega}$ and any outward normal direction $\mathbf{n}_\omega$, there exists some $\varepsilon > 0$ such that

$$\langle -(\nabla^2 A^*(\omega))^{-1}\nabla\ell(\omega), \mathbf{n}_\omega \rangle < -\varepsilon.$$

Then we have

$$\langle \mathbf{n}_\omega, \omega_* - \omega \rangle < -\frac{\varepsilon}{2\rho} + o(1/\rho) < 0$$

for some sufficiently large $\rho$. Hence $\omega_* \in \tilde{\Omega}$ and $\omega^+ = \omega_*$. $\square$

**Step 2.** Now we aim to strengthen the bound in Theorem H.4 to a last-iterate bound. In the proof of Theorem H.4 (which can be found in Appendix D in [FH24]), we can derive a recursion of form

$$\begin{aligned}
\Lambda_{t+1} &\leq \Lambda_t - \frac{\gamma_t \mu_B}{3}\Lambda_t^{2/2} + 2LV^2\gamma_t^2 \\
&= \left(1 - \frac{\gamma_t \mu_B}{3}\right)\Lambda_t + 2LV^2\gamma_t^2.
\end{aligned} \tag{40}$$

Therefore, we do not need to assume that $\Lambda_\tau \geq \varepsilon$ for all $\tau = 0, \cdots, t$ to obtain the same recursion. As a result, the same approach directly gives the upper bound of the last iterate.

### H.4  Proof of Example 2

In this section, we prove that Assumption 2 can be satisfied for a univariate Bayesian linear regression model, if the parameters $U$ and $D$ satisfy (14).

*Proof.* Using the explicit form of $\nabla^2 A^*(\omega)$ in Section C.1, we can compute

$$(\nabla^2 A^*(\xi, \Xi))^{-1} = \begin{pmatrix} \sigma^2 & 2\mu\sigma^2 \\ 2\mu\sigma^2 & 4\mu^2\sigma^2 + 2\sigma^4 \end{pmatrix}.$$

Since

$$\begin{aligned}
\ell(\xi, \Xi) &= -\mathbb{E}_q[\log p(y \mid x, z)] + D_{\mathrm{KL}}(q \| p) \\
&= \frac{1}{2}\mathbb{E}_q[(y - xz)^2] + \frac{1}{2}(-\log\sigma^2 + \sigma^2 + \mu^2 - 1) \\
&= \frac{1}{2}[(y - x\mu)^2 + x^2\sigma^2 - \log\sigma^2 + \sigma^2 + \mu^2 - 1] \\
&= \frac{1}{2}[(y - x\xi)^2 + (x^2 + 1)(\Xi - \xi^2) - \log(\Xi - \xi^2) + \xi^2 - 1].
\end{aligned}$$

Then we have

$$\nabla\ell(\xi, \Xi) = \begin{pmatrix} -xy + \frac{\xi}{\Xi - \xi^2} \\ \frac{x^2 + 1}{2} - \frac{1}{2(\Xi - \xi^2)} \end{pmatrix} = \begin{pmatrix} -xy + \frac{\mu}{\sigma^2} \\ \frac{x^2 + 1}{2} - \frac{1}{2\sigma^2} \end{pmatrix},$$

and

$$\begin{aligned}
(\nabla^2 A^*(\xi, \Xi))^{-1}\nabla\ell(\xi, \Xi) &= \sigma^2\begin{pmatrix} \mu(x^2 + 1) - xy \\ (2\mu^2 + \sigma^2)(x^2 + 1) - 1 - 2\mu xy \end{pmatrix} \\
&= \sigma^2\begin{pmatrix} \xi(x^2 + 1) - xy \\ (\Xi + \xi^2)(x^2 + 1) - 1 - 2\xi xy \end{pmatrix}.
\end{aligned}$$

Next, we consider the 4 constraints which define the boundary of $\tilde{\Omega}$.

1. $\xi = U$. If this constraint is active, the unique outward normal direction is $\mathbf{n}_\omega = (1, 0)$, and Assumption 2 requires $U(x^2 + 1) > xy$.

2. $\Xi - \xi^2 = D$. If this constraint is active, the unique unnormalized outward normal direction is $\mathbf{n}_\omega = (-2\xi, 1)$, and Assumption 2 requires $D(x^2 + 1) - 1 > 0$.

3. $\xi = -U$. If this constraint is active, the unique outward normal direction is $\mathbf{n}_\omega = (-1, 0)$, hence Assumption 2 requires $-U(x^2 + 1) < xy$.

4. $\Xi - \xi^2 = D^{-1}$. If this constraint is active, the unique unnormalized outward normal direction is $\mathbf{n}_\omega = (2\xi, -1)$, and Assumption 2 requires $-D^{-1}(x^2 + 1) + 1 > 0$.

For $\omega$ at corners (multiple constraints are active), $\mathbf{n}_\omega$ can be any normalized linear combination of the outward normal directions shown above. Then (14) remains sufficient to ensure Assumption 2.

# I Variance of the Gradient Estimator in Logistic Regression

In this section, we aim to show that Assumption 1 is satisfied for logistic regression in mean-field setting. We assume that the dataset $\mathcal{D} = \{(x_i, y_i) : x_i \in \mathbb{R}, y_i \in \{-1, 1\}\}_{i=1}^n$. We begin with Theorem I.1 to bound the gradient of the KL divergence term. In Theorem I.2 we will bound the gradient associated with the log-likelihood term.

**Lemma I.1.** *For any $\omega \in \tilde{\Omega}$,*

$$\nabla_\omega D_{\mathrm{KL}}(q(z; \omega) \,\|\, p(z)) \leq \frac{3\sqrt{d}UD}{2}.$$

*Proof.* Using the closed-form solution of KL divergence between two multivariate Gaussian distributions and the chain rule, we have

$$\nabla_{\xi_i} D_{\mathrm{KL}}(q(z; \omega) \,\|\, p(z)) = \frac{\xi_i}{\Xi_{ii} - \xi_i^2},$$

$$\nabla_{\Xi_{ii}} D_{\mathrm{KL}}(q(z; \omega) \,\|\, p(z)) = \frac{1}{2}\left(1 - \frac{1}{\Xi_{ii} - \xi_i^2}\right).$$

Therefore, we have

$$\|\nabla_\omega D_{\mathrm{KL}}(q(z; \omega) \,\|\, p(z))\|$$
$$\leq \|\nabla_\xi D_{\mathrm{KL}}(q(z; \omega) \,\|\, p(z))\| + \|\nabla_\Xi D_{\mathrm{KL}}(q(z; \omega) \,\|\, p(z))\|$$
$$\leq \sqrt{d}\left(\sup_{1 \leq i \leq d} \|\nabla_{\xi_i} D_{\mathrm{KL}}(q(z; \omega) \,\|\, p(z))\| + \sup_{1 \leq i \leq d} \|\nabla_{\Xi_{ii}} D_{\mathrm{KL}}(q(z; \omega) \,\|\, p(z))\|\right)$$
$$\leq \sqrt{d}\left(UD + \frac{1}{2}\max\{|1 - D|, |1 - D^{-1}|\}\right)$$
$$\leq \sqrt{d}\left(UD + \frac{D}{2}\right)$$
$$\leq \frac{3\sqrt{d}UD}{2},$$

where we write $\nabla_\xi D_{\mathrm{KL}}(q(z; \omega) \,\|\, p(z))$ as a vector in $\mathbb{R}^d$ and use the fact that $U, D \geq 1$. $\qquad\square$

**Lemma I.2.** *In $\mathcal{D}$, denote $s_1 = \max_i \|x_i\|$ and $s_2 = \max_i \|x_i \odot x_i\|$. Then for all $1 \leq i \leq n$ and $z \in \mathbb{R}^d$ we have*

$$\| -\nabla_z \log p(y_i \,|\, x_i, z) + \nabla_z^2 \log p(y_i \,|\, x_i, z) \odot \xi\| \leq s_1 + \frac{Us_2}{4}, \quad \| -\nabla_z^2 \log p(y_i \,|\, x_i, z)\| \leq \frac{s_2}{4}.$$

*Here $\nabla_z^2 \log p(y_i \,|\, x_i, z)$ is defined as a $d$-dimensional vector where $j$-th entry equals $\nabla_{z_j z_j}^2 \log p(y_i \,|\, x_i, z)$.*

*Proof.* For the $j$-th entry,

$$|-\nabla_{z_j} \log p(y_i | x_i, z) + \xi_j \nabla_{z_j z_j}^2 \log p(y_i \,|\, x_i, z)|$$
$$= |\nabla_{z_j} \log(1 + e^{-y_i x_i^\top z}) - \xi_j \nabla_{z_j z_j}^2 \log(1 + e^{-y_i x_i^\top z})|$$
$$= |-\sigma(-y_i x_i^\top z) y_i x_{ij} - \xi_j \sigma(-y_i x_i^\top z)(1 - \sigma(-y_i x_i^\top z)) x_{ij}^2|$$
$$\leq |-\sigma(-y_i x_i^\top z) y_i x_{ij}| + |\xi_j \sigma(-y_i x_i^\top z)(1 - \sigma(-y_i x_i^\top z)) x_{ij}^2|$$
$$\leq |x_{ij}| + \frac{U}{4} x_{ij}^2,$$

$$|-\nabla_{z_j z_j}^2 \log p(y_i | x_i, z)| = |\nabla_{z_j z_j}^2 \log(1 + e^{-y_i x_i^\top z})|$$
$$= |\sigma(-y_i x_i^\top z)(1 - \sigma(-y_i x_i^\top z)) x_{ij}^2|$$
$$\leq \frac{1}{4} x_{ij}^2,$$

where $\sigma(\cdot)$ is the sigmoid function. Then we have

$$
\begin{aligned}
\| - \nabla_z \log p(y_i \,|\, x_i, z) &+ \nabla_z^2 \log p(y_i \,|\, x_i, z) \odot \xi \| \\
&= \|(| - \nabla_{z_j} \log p(y_i \,|\, x_i, z) + \xi_j \nabla_{z_j z_j}^2 \log p(y_i \,|\, x_i, z)|)_j\| \\
&\leq \|(|x_{ij}| + \frac{U}{4} x_{ij}^2)_j\| \\
&\leq \|x_i\| + \frac{U}{4}\|x_i \odot x_i\| \\
&\leq s_1 + \frac{U s_2}{4},
\end{aligned}
$$

$$
\| - \nabla_z^2 \log p(y_i \,|\, x_i, z) \| = \|(-\nabla_{z_j z_j}^2 \log p(y_i \,|\, x_i, z))_j\| \leq \frac{1}{4}\|x_i \odot x_i\| \leq \frac{s_2}{4}.
$$

$\qquad\qquad\qquad\qquad\qquad\qquad\qquad\qquad\qquad\qquad\qquad\qquad\qquad\qquad\qquad\qquad\square$

With the same approach as in Lemma I.2, we have the following bounds on the log-likelihood term and the gradient $\nabla\ell(\omega)$.

$$
\|\nabla_\xi \mathbb{E}[-\log p(y_i \,|\, x_i, z)]\| = \|\mathbb{E}[-\nabla_z \log p(y_i \,|\, x_i, z) + \nabla_z^2 \log p(y_i \,|\, x_i, z) \odot \xi]\| \leq s_1 + \frac{U s_2}{4},
$$

$$
\|\nabla_\Xi \mathbb{E}[-\log p(y_i \,|\, x_i, z)]\| = \frac{1}{2}\|\mathbb{E}[-\nabla_z^2 \log p(y_i \,|\, x_i, z)]\| \leq \frac{s_2}{8},
$$

$$
\begin{aligned}
\|\nabla\ell(\omega)\| &\leq \sum_{i=1}^n \left[\|\nabla_\xi \mathbb{E}[-\log p(y_i \,|\, x_i, z)]\| + \|\nabla_\Xi \mathbb{E}[-\log p(y_i \,|\, x_i, z)]\|\right] \\
&\qquad + \|\nabla_\omega D_{\mathrm{KL}}(q(z;\omega) \,\|\, p(z))\| \\
&\leq n\left(s_1 + \frac{U s_2}{4} + \frac{s_2}{8}\right) + \frac{3\sqrt{d}UD}{2}.
\end{aligned} \tag{41}
$$

Next, we define our gradient estimator. Consider mini-batch gradient estimator

$$
\begin{aligned}
\hat{\nabla}_{\xi, MB}\ell(\omega) &= \frac{n}{m}\sum_{k=1}^m \nabla_\xi \mathbb{E}[-\log p(y_{i_k} \,|\, x_{i_k}, z)] + \nabla_\xi D_{\mathrm{KL}}(q(z) \,\|\, p(z)) \\
&= \frac{n}{m}\sum_{i=1}^k \mathbb{E}[-\nabla_z \log p(y_{i_k} \,|\, x_{i_k}, z) + \nabla_z^2 \log p(y_{i_k} \,|\, x_{i_k}, z) \odot \xi] \\
&\qquad + \nabla_\xi D_{\mathrm{KL}}(q(z) \,\|\, p(z)), \\
\hat{\nabla}_{\Xi, MB}\ell(\omega) &= \frac{n}{m}\sum_{k=1}^m \nabla_\Xi \mathbb{E}[-\log p(y_{i_k} \,|\, x_{i_k}, z)] + \nabla_\Xi D_{\mathrm{KL}}(q(z) \,\|\, p(z)) \\
&= \frac{n}{2m}\sum_{k=1}^k \mathbb{E}[-\nabla_z^2 \log p(y_{i_k} \,|\, x_{i_k}, z)] + \nabla_\Xi D_{\mathrm{KL}}(q(z) \,\|\, p(z)),
\end{aligned}
$$

where each $i_k$ is sampled uniformly from $\{1, \cdots, n\}$, and $m$ is the batch size. There is usually no closed-form solution of the expectation of the log-likelihood, thus we approximate the expectation with $N$ samples, i.e.,

$$
\begin{aligned}
\hat{\nabla}_\xi \ell(\omega) &= \frac{n}{mN}\sum_{k=1}^m \sum_{l=1}^N [-\nabla_z \log p(y_{i_k} \,|\, x_{i_k}, z_l) + \nabla_z^2 \log p(y_{i_k} \,|\, x_{i_k}, z_l) \odot \xi] \\
&\qquad + \nabla_\xi D_{\mathrm{KL}}(q(z) \,\|\, p(z)), \\
\hat{\nabla}_\Xi \ell(\omega) &= \frac{n}{2mN}\sum_{k=1}^m \sum_{l=1}^N [-\nabla_z^2 \log p(y_{i_k} \,|\, x_{i_k}, z_l)] + \nabla_\Xi D_{\mathrm{KL}}(q(z) \,\|\, p(z)),
\end{aligned} \tag{42}
$$

where $z_1, \cdots, z_N$ are iid samples of the variational distribution $q$. It is obvious that $\mathbb{E}[\hat{\nabla}_{\mathrm{MC}}\ell(\omega)] = \hat{\nabla}\ell(\omega)$, so the gradient estimator is unbiased.

Similar to (41), we can obtain the following bound on the stochastic gradient $\hat{\nabla}\ell(\omega)$.

$$
\begin{aligned}
\|\hat{\nabla}\ell(\omega)\| &\le \frac{n}{mN}\sum_{k=1}^{m}\sum_{l=1}^{N}[\|-\nabla_\xi \log p(y_{i_k}\,|\,x_{i_k},z_l)\| + \|-\nabla_\Xi \log p(y_{i_k}\,|\,x_{i_k},z_l)\|] \\
&\quad + \|\nabla_\omega D_{\mathrm{KL}}(q(z;\omega)\,\|\,p(z))\| \\
&\le n\left(s_1 + \frac{Us_2}{4} + \frac{s_2}{8}\right) + \frac{3\sqrt{d}UD}{2}.
\end{aligned}
\tag{43}
$$

Next we prove that this estimator satisfies bounded variance assumption (Assumption 3), which will also be useful for the proof of Assumption 1.

**Lemma I.3.** *The Monte Carlo gradient estimator satisfies Assumption 3, i.e.,*

$$
\mathbb{E}[\|\hat{\nabla}\ell(\omega) - \nabla\ell(\omega)\|^2] \le 16n^2\left(\left(s_1 + \frac{Us_2}{4}\right)^2 + \frac{s_2^2}{64}\right).
$$

*Proof.*

$$
\begin{aligned}
&\mathbb{E}[\|\hat{\nabla}\ell(\omega) - \nabla\ell(\omega)\|^2] \\
&\le 2\mathbb{E}[\|\hat{\nabla}\ell(\omega) - \hat{\nabla}_{MB}\ell(\omega)\|^2] + 2\mathbb{E}[\|\hat{\nabla}_{MB}\ell(\omega) - \nabla\ell(\omega)\|^2] \\
&\le 2\mathbb{E}\left[\left\|\frac{1}{N}\sum_{l=1}^{n}\Big[\frac{n}{m}\sum_{k=1}^{m}[-\nabla_z \log p(y_{i_k}\,|\,x_{i_k},z_l) + \xi\nabla_z^2 \log p(y_{i_k}\,|\,x_{i_k},z_l)] \right.\right. \\
&\qquad\qquad \left.\left. -\frac{n}{m}\sum_{i=1}^{k}\mathbb{E}[-\nabla_z \log p(y_{i_k}\,|\,x_{i_k},z) + \xi\nabla_z^2 \log p(y_{i_k}\,|\,x_{i_k},z)]\Big]\right\|^2\right] \\
&\quad + 2\mathbb{E}\left[\left\|\frac{1}{N}\sum_{l=1}^{N}\Big[\frac{n}{2m}\sum_{k=1}^{m}[-\nabla_z^2 \log p(y_{i_k}\,|\,x_{i_k},z_l)] - \frac{n}{2m}\sum_{k=1}^{k}\mathbb{E}[-\nabla_z^2 \log p(y_{i_k}\,|\,x_{i_k},z)]\Big]\right\|^2\right] \\
&\quad + 2\mathbb{E}\left[\left\|\frac{n}{m}\sum_{k=1}^{m}\nabla\mathbb{E}[-\log p(y_{i_k}\,|\,x_{i_k},z)] - \sum_{i=1}^{n}\nabla\mathbb{E}[-\log p(y_{i_k}\,|\,x_{i_k},z)]\right\|^2\right] \\
&\le 4\max_{i,z}\left\|\frac{n}{m}\sum_{k=1}^{m}[-\nabla_z \log p(y_i\,|\,x_i,z) + \nabla_z^2 \log p(y_i\,|\,x_i,z)\odot\xi]\right\|^2 \\
&\quad + 4n^2\max_i\|\nabla_\xi\mathbb{E}[-\log p(y_i\,|\,x_i,z)]\|^2 \\
&\quad + 4\max_{i,z}\left\|\frac{n}{2m}\sum_{k=1}^{m}[-\nabla_z^2 \log p(y_{i_k}\,|\,x_{i_k},z)]\right\|^2 + 4n^2\max_i\|\nabla_\Xi\mathbb{E}[-\log p(y_i\,|\,x_i,z)]\|^2 \\
&\quad + 8n^2(\max_i\|\nabla_\xi\mathbb{E}[-\log p(y_i\,|\,x_i,z)]\|^2 + \max_i\|\nabla_\Xi\mathbb{E}[-\log p(y_i\,|\,x_i,z)]\|^2) \\
&\le 4n^2\left(\max_{i,z}\|-\nabla_z \log p(y_i\,|\,x_i,z) + \nabla_z^2 \log p(y_i\,|\,x_i,z)\odot\xi\|^2 + \max_{i,z}\left\|-\frac{1}{2}\nabla_z^2 \log p(y_i\,|\,x_i,z)\right\|^2\right) \\
&\quad + 12n^2(\max_i\|\nabla_\xi\mathbb{E}[-\log p(y_i\,|\,x_i,z)]\|^2 + \max_i\|\nabla_\Xi\mathbb{E}[-\log p(y_i\,|\,x_i,z)]\|^2) \\
&\le 16n^2\left(\left(s_1 + \frac{Us_2}{4}\right)^2 + \frac{s_2^2}{64}\right).
\end{aligned}
$$

$\square$

Finally, we are ready to prove that Assumption 1 holds for logistic regression.

**Theorem I.4.** *Consider stochastic gradient estimator* (42). *Then with* $s_1 = \max_i \|x_i\|$ *and* $s_2 = \max_i \|x_i \odot x_i\|$, *Assumption* 1 *is satisfied with some* $V^2 > 0$, *which is at most polynomial in* $n, d, U, D, s_1$ *and* $s_2$.

*Proof.* We first summarize our notations.

Here $\omega_{t+1,*}^+$ (and $\omega_{t+1,*}$) are obtained by doing an (S)MD update from $\omega_t$ with step size $\gamma_t$, $\omega_{t+1} = \operatorname{argmin}_{\omega \in \tilde{\Omega}} D_{A^*}(\omega, \omega_{t+1,*})$ and $\omega_{t+1}^+ = \operatorname{argmin}_{\omega \in \tilde{\Omega}} D_{A^*}(\omega, \omega_{t+1,*}^+)$.

We decompose (12) into the sum of two terms.

$$
\frac{1}{\gamma_t} \mathbb{E}[\langle \hat{\nabla}\ell(\omega_t) - \nabla\ell(\omega_t), \omega_{t+1}^+ - \omega_{t+1} \rangle \,|\, \omega_t]
$$
$$
= \frac{1}{\gamma_t} \mathbb{E}[\langle \hat{\nabla}\ell(\omega_t) - \nabla\ell(\omega_t), \omega_{t+1}^+ - \omega_t \rangle \,|\, \omega_t] + \frac{1}{\gamma_t} \mathbb{E}[\langle \hat{\nabla}\ell(\omega_t) - \nabla\ell(\omega_t), \omega_t - \omega_{t+1} \rangle \,|\, \omega_t]
$$
$$
\leq \frac{1}{\gamma_t} \mathbb{E}[\|\hat{\nabla}\ell(\omega_t) - \nabla\ell(\omega_t)\| \|\omega_{t+1}^+ - \omega_t\| \,|\, \omega_t] + \frac{1}{\gamma_t} \mathbb{E}[\|\hat{\nabla}\ell(\omega_t) - \nabla\ell(\omega_t)\| \|\omega_{t+1} - \omega_t\| \,|\, \omega_t].
$$
(44)

Next, we aim to bound $\|\omega_{t+1} - \omega_t\|$. By $C_S$-strong convexity of $A^*$ and definition of $\omega_{t+1}^*$, we have

$$
\|\omega_{t+1,*} - \omega_{t+1}\| \leq \sqrt{\frac{2}{C_S} D_{A^*}(\omega_{t+1}, \omega_{t+1,*})}
$$
$$
\leq \sqrt{\frac{2}{C_S} D_{A^*}(\omega_t, \omega_{t+1,*})}
$$
$$
\leq \sqrt{\frac{2}{C_S} (D_{A^*}(\omega_t, \omega_{t+1,*}) + D_{A^*}(\omega_{t+1,*}, \omega_t))}
$$
$$
= \sqrt{\frac{2}{C_S} \langle \nabla A^*(\omega_{t+1,*}) - \nabla A^*(\omega_t), \omega_{t+1,*} - \omega_t \rangle}
$$
$$
\leq \frac{\sqrt{2}}{C_S} \|\nabla A^*(\omega_{t+1,*}) - \nabla A^*(\omega_t)\|.
$$

Moreover, by strong convexity we have

$$
\|\omega_{t+1,*} - \omega_t\| \leq \frac{1}{C_S} \|\nabla A^*(\omega_{t+1,*}) - \nabla A^*(\omega_t)\|.
$$

Then by triangle inequality we get

$$
\|\omega_{t+1} - \omega_t\| \leq \|\omega_{t+1} - \omega_{t+1,*}\| + \|\omega_{t+1,*} - \omega_t\| \leq \frac{\sqrt{2}+1}{C_S} \|\nabla A^*(\omega_{t+1,*}) - \nabla A^*(\omega_t)\|.
$$

Then we use the definition of SMD step in (11) to get

$$
\|\omega_{t+1} - \omega_t\| \leq \frac{\sqrt{2}+1}{C_S} \|\nabla A^*(\omega_{t+1,*}) - \nabla A^*(\omega_t)\| \leq \frac{\sqrt{2}+1}{C_S} \gamma_t \|\hat{\nabla}\ell(\omega_t)\|.
$$
(45)

Similarly, we have

$$
\|\omega_{t+1}^+ - \omega_t\| \leq \frac{\sqrt{2}+1}{C_S} \gamma_t \|\nabla\ell(\omega_t)\|.
$$
(46)

Plug (45) and (46) into (44) and we get

$$\frac{1}{\gamma_t}\mathbb{E}[\langle \hat{\nabla}\ell(\omega_t) - \nabla\ell(\omega_t), \omega_{t+1}^+ - \omega_{t+1}\rangle \mid \omega_t]$$

$$\leq \frac{\sqrt{2}+1}{C_S}\mathbb{E}[\|\hat{\nabla}\ell(\omega_t) - \nabla\ell(\omega_t)\|\|\nabla\ell(\omega_t)\| \mid \omega_t] + \frac{\sqrt{2}+1}{C_S}\mathbb{E}[\|\hat{\nabla}\ell(\omega_t) - \nabla\ell(\omega_t)\|\|\hat{\nabla}\ell(\omega_t)\| \mid \omega_t]$$

$$\leq \frac{\sqrt{2}+1}{C_S}\sqrt{\mathbb{E}[\|\hat{\nabla}\ell(\omega_t) - \nabla\ell(\omega_t)\|^2]} \sup_{\omega_t \in \tilde{\Omega}}\left(\|\nabla\ell(\omega_t)\| + \|\hat{\nabla}\ell(\omega_t)\|\right)$$

$$\leq \frac{\sqrt{2}+1}{C_S} \times 4n\sqrt{\left(s_1 + \frac{Us_2}{4}\right)^2 + \frac{s_2^2}{64}} \times \left(2n\left(s_1 + \frac{Us_2}{4} + \frac{s_2}{8}\right) + 3\sqrt{d}UD\right)$$

$$\leq \frac{\sqrt{2}+1}{C_S}[(8s_1^2 + 2Us_2 + s_2)n^2 + 12\sqrt{d}UDn]\sqrt{\left(s_1 + \frac{Us_2}{4}\right)^2 + \frac{s_2^2}{64}}.$$

$\square$

# J  Experiment Details and Additional Results

## J.1  Pseudocode of the Algorithms

We first present the pseudocode of Proj-SNGD algorithm proposed in Section 4.1.

---

**Algorithm 1** Proj-SNGD

---

**Require:** Initialization $\omega_0 \in \tilde{\Omega}$, number of iterations $T$, step sizes $\{\gamma_t\}_{0 \leq t \leq T-1}$
 1: **for** $t = 0, 1, \ldots, T-1$ **do**
 2:     Compute stochastic gradient $\hat{\nabla}\ell(\omega_t)$
 3:     Compute $\omega_{t+1,*} \in \Omega$ such that

$$\nabla A^*(\omega_{t+1,*}) = \nabla A^*(\omega_{t+1}) - \gamma_t \hat{\nabla}\ell(\omega_t) \tag{47}$$

 4:     Set

$$\omega_{t+1} = \mathrm{Proj}_{\tilde{\Omega}}(\omega_{t+1,*}) \tag{48}$$

 5: **end for**
 6: Sample $\bar{\omega}_T$ from $\{\omega_t\}_{0 \leq t \leq T-1}$ with probability $p_t = \gamma_t / \sum_{i=0}^{T-1} \gamma_i$
 7: **Return** $\bar{\omega}_T$

---

Next, we provide the pseudocode of Prox-SGD and Proj-SGD from [Dom20, DGG23]. We write

$$\tilde{L}(\theta) = \tilde{L}(\mu, C) := \mathbb{E}_{q(z;\theta)}[-\log p(y \,|\, z) - \log p(z)].$$

Therefore, using the definition of $L(\theta)$ in (30), we have

$$L(\theta) = \tilde{L}(\theta) + \mathbb{E}_{q(z;\theta)}[\log q(z;\theta)].$$

---

**Algorithm 2** Prox-SGD

---

**Require:** Initialization $\mu_0 \in \mathbb{R}^d, C_0 \in \mathcal{S}_+^d$ and diagonal, iterations $T$, step sizes $\{\gamma_t\}_{0 \leq t \leq T-1}$,
 1: **for** $t = 0, 1, \ldots, T-1$ **do**
 2:     Compute stochastic gradient $\hat{\nabla}_\mu \tilde{L}(\mu_t, C_t), \hat{\nabla}_C \tilde{L}(\mu_t, C_t)$
 3:     Set $\mu_{t+1} = \mu_t - \gamma_t \hat{\nabla}_\mu \tilde{L}(\mu_t, C_t), \quad C_{t+1,*} = C_t - \gamma_t \hat{\nabla}_C \tilde{L}(\mu_t, C_t)$
 4:     For each $1 \leq i \leq d$, set

$$(C_{t+1})_{ii} = \frac{1}{2}\left((C_t)_{ii} + \sqrt{(C_t)_{ii}^2 + 4\gamma_t}\right) \tag{49}$$

 5: **end for**
 6: **Return** $(\mu_T, C_T)$

---

In Algorithm 3, $M$ is defined as the smoothness parameter of the joint log-likelihood $\log p(z, \mathcal{D})$ with respect to $z$. Recall that the clipping function in (50) is defined as $\mathrm{clip}_{[a,b]}(x) = \min\{\max\{a, x\}, b\}$.

---

**Algorithm 3** Proj-SGD

---

**Require:** Initialization $\mu_0 \in \mathbb{R}^d, C_0 \in \mathcal{S}_+^d$ and diagonal, iterations $T$, step sizes $\{\gamma_t\}_{0 \leq t \leq T-1}$,
    smoothness parameter $M$
 1: **for** $t = 0, 1, \ldots, T-1$ **do**
 2:     Compute stochastic gradient $\hat{\nabla}_\mu L(\mu_t, C_t), \hat{\nabla}_C L(\mu_t, C_t)$
 3:     Set $\mu_{t+1} = \mu_t - \gamma_t \hat{\nabla}_\mu L(\mu_t, C_t), \quad C_{t+1,*} = C_t - \gamma_t \hat{\nabla}_C L(\mu_t, C_t)$
 4:     For each $1 \leq i \leq d$, set

$$(C_{t+1})_{ii} = \mathrm{clip}_{[1/\sqrt{M}, +\infty)}((C_{t+1,*})_{ii}) \tag{50}$$

 5: **end for**
 6: **Return** $(\mu_T, C_T)$

---

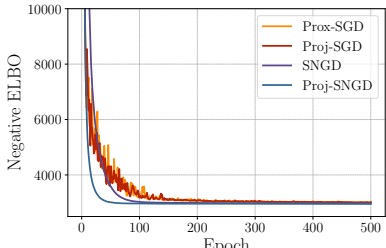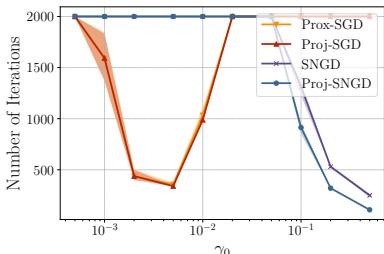

Figure 4: Euclidean and non-Euclidean algorithms on Madelon dataset. Left: Objective during optimization with tuned step size. Right: Number of iterations before the objective falls below $\ell(\omega) \leq 3000$ for different initial step sizes $\gamma_0$. Non-Euclidean algorithms show consistently better performance, tolerate larger step sizes and are more robust to step size tuning.

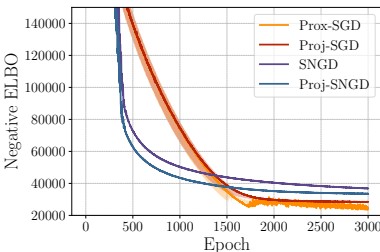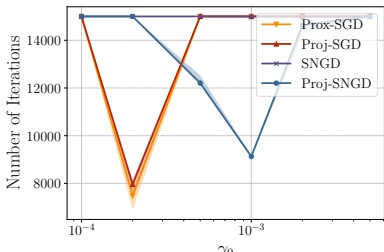

Figure 5: Euclidean and non-Euclidean algorithms on CIFAR-10 dataset. Left: Objective during optimization with tuned step size. Right: Number of iterations before the objective falls below $\ell(\omega) \leq 35000$ for different initial step sizes $\gamma_0$. Euclidean algorithms are more robust to step size tuning and achieve faster convergence in the initial phase of optimization, but non-Euclidean algorithms exhibit faster convergence after 3000 epochs.

### J.2 Experimental Results on Additional Datasets

In this section, we compare the performance of Euclidean and non-Euclidean algorithms on Madelon and CIFAR-10 dataset.

**Experiment on Madelon Dataset.** In this experiment, we compare non-Euclidean algorithms (Proj-SNGD and SNGD) with Euclidean algorithms on Madelon dataset. Details of implementation can be found in Section J.4.

We consider logistic regression on Madelon dataset ($n = 2600$, $d = 500$). We use mini-batches of size 2000 and set the step size $\gamma_t = \gamma_0/\sqrt{t}$, where $\gamma_0$ is a hyperparameter to be tuned. We run the algorithm for 1000 epochs (2000 iterations). The results of 5 independent runs are shown in Figure 4. Similar to the results of the MNIST experiment, we observe that Proj-SNGD slightly outperforms SNGD, and both non-Euclidean algorithms achieve faster convergence than Euclidean counterparts. Moreover, non-Euclidean algorithms, especially Proj-SNGD, are more robust to step size and admit larger step sizes.

**Experiment on CIFAR-10 Dataset.** In this experiment, we consider logistic regression on a subset of CIFAR-10 dataset with pictures of cats and dogs ($n = 10000$, $d = 3072$). We use mini-batches of size 2000 and set the step size $\gamma_t = \gamma_0/\sqrt{t}$, where $\gamma_0$ is a hyperparameter to be tuned. We run the algorithm for 3000 epochs (15000 iterations). The results of 5 independent runs are shown in Figure Figure 5. In the left panel of Figure 5, we observe that non-Euclidean algorithms converge faster during the initial phase (before 1500 epochs). However, Euclidean algorithms reach the optimum faster overall. In the right panel of Figure 5. Notably, non-Euclidean algorithms can accommodate larger step sizes and are the most robust to step size tuning.

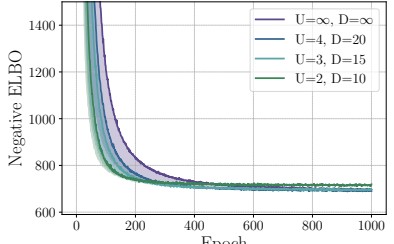 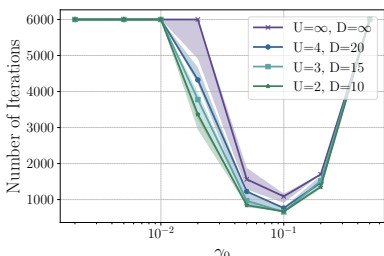

Figure 6: Convergence of non-Euclidean algorithms on MNIST dataset. Left: Objective during optimization with $\gamma_0 = 0.05$. Right: Number of iterations before the objective falls below $\ell(\omega) \leq 700$ for different initial step sizes $\gamma_0$.

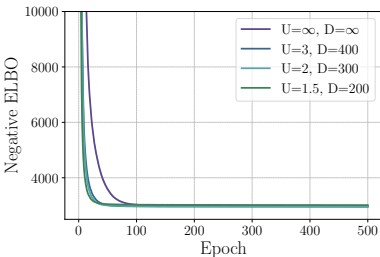 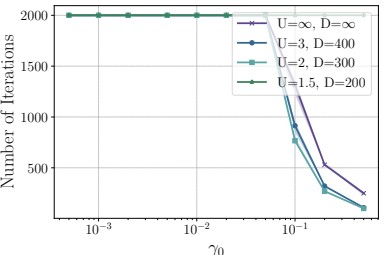

Figure 7: Convergence of non-Euclidean algorithms on Madelon dataset. Left: Objective during optimization with $\gamma_0 = 0.5$. Right: Number of iterations before the objective falls below $\ell(\omega) \leq 3000$ for different initial step sizes $\gamma_0$.

### J.3 Choice of $U$ and $D$ in Proj-SNGD

In this section, we examine the effect of projection in the Proj-SNGD algorithm.

**MNIST Dataset.** We compare four different settings: no projection ($U = \infty, D = \infty$), projection with $U = 4, D = 20$, $U = 3, D = 15$ and $U = 2, D = 10$. Smaller values of $U$ and $D$ yield a smaller smoothness coefficient and a larger hidden convexity coefficient, leading to stronger theoretical guarantees.

**Madelon Dataset.** For this dataset, we consider the following four different settings: no projection, projection with $U = 3, D = 400$, $U = 2, D = 300$ and $U = 1.5, D = 200$.

**CIFAR-10 Dataset.** For CIFAR-10 dataset, we adopt the same settings as those used for the MNIST dataset: no projection, projection with $U = 4, D = 20$, $U = 3, D = 15$ and $U = 2, D = 10$.

As shown in the left panel of Figure 6, we note that projection can accelerate convergence. However, when $U$ and $D$ are too small (e.g., $U = 2$ and $D = 10$), Proj-SNGD may converge to a sub-optimal solution. The right panel of Figure 6 confirms that smaller $U$ and $D$ lead to faster convergence. Therefore, a moderate choice of $U$ and $D$ provides both strong theoretical guarantees and favorable empirical performance. Similar phenomena can also be observed in Figures 7 and 8, where projection with suitable choices of $U$ and $D$ can accelerate and stabilize the training process. If $U$ and $D$ are too small, the optimal solution may fall outside the search space (see right panel of Figures 7 and 8), and Proj-SNGD will converge to a suboptimal solution. If they are too large, the algorithm will not benefit from projection (see SNGD without projection in Figures 6 to 8).

The slow convergence of SNGD without projection is due to the poor tolerance for large step sizes, which causes divergence in the initial phase. In contrast, projection stabilizes this in the initial phase and accelerates convergence (see the Poisson regression example in Figure 2). This effect is consistently observed across 3 different datasets. Moreover, this effect is consistent with our theoretical upper bounds for the relative smoothness parameter. This is because our upper bound

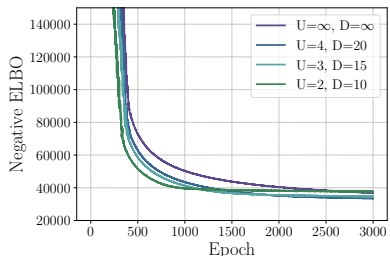
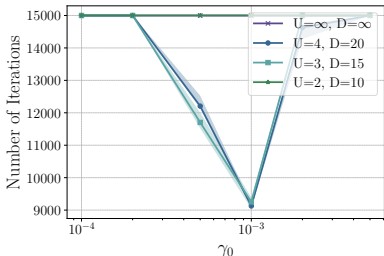

Figure 8: Convergence of non-Euclidean algorithms on CIFAR-10 dataset. Left: Objective during optimization with $\gamma_0 = 10^{-3}$. Right: Number of iterations before the objective falls below $\ell(\omega) \leq 35000$ for different initial step sizes $\gamma_0$.

on $\ell$ depends on the diameter of the compact set, indicating that relative smoothness may not hold globally, which leads to divergence behavior in the initial phase.

### J.4 Implementation Details

In all experiments involving Proj-SGD, we set $M = D$ (see Algorithm 3 for the definition of $M$) for a fair comparison with Proj-SNGD. We use $N = 2000$ samples from variational distribution $q$ to estimate the expected log-likelihood (see the definition of stochastic gradient estimator (42)).

In the left panel of Figure 3 (MNIST dataset), we set $\gamma_0 = 0.05$ for Proj-SNGD and SNGD, and $\gamma_0 = 0.01$ for Prox-SGD and Proj-SGD. In the left panel of Figure 4 (Madelon dataset), we set $\gamma_0 = 0.5$ for non-Euclidean algorithms, and $\gamma_0 = 0.01$ for Euclidean algorithms. In the left panel of Figure 5 (CIFAR-10 dataset), we set $\gamma_0 = 10^{-3}$ for non-Euclidean algorithms, and $\gamma_0 = 2 \times 10^{-4}$ for Euclidean algorithms.

All experiments, except for CIFAR-10, are conducted on an Apple M3 Pro CPU. On the MNIST dataset, training for 1000 epochs takes approximately 5 minutes, and on the Madelon dataset it takes about 1 minute. The CIFAR-10 experiment is run on an NVIDIA GeForce RTX 3090 GPU, requiring roughly 8 minutes for 3000 epochs.

