# OpenReview forum: "Natural Gradient VI: Guarantees for Non-Conjugate Models"
_NeurIPS.cc/2025/Conference — NeurIPS 2025 poster_

### Official Review · Reviewer_tPUM · 2025-06-18

**Clarity:** 3
**Significance:** 2
**Originality:** 3
**Rating:** 4
**Confidence:** 5

**Summary:**

The paper analyzes the convergence of natural gradient variational inference (NGVI) with stochastic gradients applied to non-conjugate models. Obtaining convergence guarantees for stochastic NGVI under realizable/verifiable assumptions has been an open problem for decades. The main challenge has been the fact that, under typical smoothness and log-concavity assumptions on the target, the ELBO objective is no longer convex and smooth with respect to the canonical variational parametrizations of NGVI. This paper addresses this open issue by constraining the variational parameters to a compact set. This enables the ELBO to be relatively smooth and even Polyak-Lojasiewicz (PL). In light of this, for log-smooth and log-Hessian-smooth likelihoods, the paper reports a $O(1/\sqrt{T})$ convergence rate for reducing the squared gradient norm of the ELBO. If the likelihood is additionally strongly log-concave, the paper reports a $O(1/T)$ rate for reducing ELBO. Some experimental results in favor of confining the parameters to a compact set are demonstrated.

**Questions:**

n/a

**Ethical Concerns:**

["NO or VERY MINOR ethics concerns only"]

**Final Justification:**

The authors resolved my original major concerns on the correctness of the experimental results and the peculiar Assumption 2. Therefore, I now lean towards accepting the paper.

**Limitations:**

The limitation section in the paper is appropriate.

**Paper Formatting Concerns:**

* [CE24] has a broken font
* [AZO14] is pointing to the ArXiv version when it was published at ICTS 2017.

**Quality:**

2

**Strengths And Weaknesses:**

### Strengths
* The paper demonstrates that the ELBO can be relatively smooth if the variational parameters are confined to a compact set. As I haven't seen this result before, this is new and potentially interesting.
* The fact that the ELBO can satisfy PL due to hidden convexity is interesting.

### Weaknesses
* The fact that the ELBO is relatively smooth under compactness is unsurprising. Compactness is very strong, and most regularity assumptions start to hold under this assumption due to continuity. However, in addition to this, the relative smoothness result requires both the log-likelihood and its gradient to be Lipschitz. The need for both at the same time is quite strong.
* It is unclear to me if Assumption 1 can be simply assumed away. In [1], Assumption 1 was explicitly established by studying special cases. However, that justification does not apply here since [1] did not study NGVI with Monte Carlo gradients. Given that the assumptions used in the paper are already strong enough, I would expect Assumption 1 to be verified explicitly.
* The experiments in Section 5 have a correctness issue. In particular, the implementation of Prox-SGD seems to be incorrect, which is why it appears that it is not converging in Figure 3. I checked the code, and the error appears to be that the stochastic gradient used for Prox-SGD is differentiating the whole ELBO $\nabla \ell(z)$ instead of just the energy term $\nabla_{w} \mathbb{E}_{z \sim q(\cdot; w)} -\log p(D|z)$. In Prox-SGD, the proximal operator handles the entropy term. Therefore, the gradient should only be taken on the energy term. Otherwise, the algorithm ends up minimizing the entropy term twice, which is what seems to be happening.
* It is unclear why Assumption 2 is needed and what it implies. Why has this to be assumed when projections to a compact set are already being used?

The problem of proving the convergence of stochastic NGVI on non-conjugate models is a genuinely difficult problem where very little progress has been made. Therefore, showing any useful theoretical results is a significant contribution. However, the set of assumptions are really strong to shed much light into the behavior of NGVI. Still, I think any theoretical result on this problem could serve as an initial step for future works. As such, I am in favor of accepting the submission. The reason for the rejection-leaning score is due to the correctness concern in the experiments and the unclear meaning of Assumption 2. If these concerns are addressed, I will happily raise my score.

## Major Comments
* The title is too non-descriptive. It is not obvious that this paper is about a theoretical analysis of the convergence of NGVI.
* Discuss the assumptions in the abstract, especially since they are strong. The fact that the domain is enforced to be bounded and both the log-likelihood and its gradient need to be Lipschitz continuous should be stated.  Citing the parametrization considered in the paper would also be useful.
* What is also not clear to me is why the paper focuses on joint likelihoods with an explicit prior-likelihood pair. Since the paper considers the non-conjugate setting, it should suffice to just consider a generic unnormalized target, which should simplify the setup and the notation. Similarly, the regularity results like Theorem 3.2  and Proposition 2,3 assume regularity of the likelihood $\log p(\mathcal{D}\mid z)$. This doesn't seem to be necessary. Why not just assume the regularity of the joint? (and equivalently, the "target unnormalized density")
* Line 211: Isn't this about "strong" convexity, not just convexity? If it is the case, this should be made more precise.
* Proposition 2: I think the constant $\mu_C$ can easily be improved. If the diagonal of a triangular matrix is strictly positive, then the diagonal elements correspond to its eigenvalues. Also, the eigenvalues coincide with the singular values. Then I think, for all $i$, \sigma_{\mathrm{max}}(G_i) \leq \max\left(1, 2C_{11}, \ldots, 2C_{dd}\right) \leq 1 + 2 \max_{j} C_{jj} \leq 1 + 2 \sqrt{D}$ holds?

## Minor Comments
* Line 26: BBVI, essentially non-conjugate ELBO maximization with SGD, was concurrently developed by [1,2] as well.
* Line 29: While I know that the term "Euclidean gradient descent" has been used in some works, I personally feel that this is not descriptive enough. (Which Euclidean space?) Gradient descent in the space of parameters is probably a better term in my opinion.
* Line 31: Cite something for NVGI. My understanding is that Amari's paper doesn't specifically describe VI with natural gradients. Therefore, NGVI should be attributed to some other paper.
* Line 38 "where the posterior is not contained in the variational family": This statement doesn't seem necessary.
* Line 42: Cite something for mirror descent.
* Line 72: Convexity was established in [2]. Domke 2020 generalized their result to strongly convex targets.
* Line 184: Why not change $U$, $D$ to $M$, $S$ so that $M$ corresponds to the bound on $\mu$ and $S$ corresponds to the bound on $\Sigma$.
* Line 297: The use of projected and proximal SGD was first proposed in [3].

1. Wingate, D., & Weber, T. (2013). Automated variational inference in probabilistic programming. arXiv preprint arXiv:1301.1299.
2. Titsias, M., & Lázaro-Gredilla, M. (2014, June). Doubly stochastic variational Bayes for non-conjugate inference. In International conference on machine learning (pp. 1971-1979). PMLR.
3. Domke, J. (2020, November). Provable smoothness guarantees for black-box variational inference. In International Conference on Machine Learning (pp. 2587-2596). PMLR.

---

> ### Author Rebuttal · Authors · 2025-07-31
>
> Thank you for taking the time to thoroughly review our manuscript and share constructive suggestions. We address each of your weaknesses and major comments in detail below.
>
> ### Weaknesses
>
> > The fact that the ELBO is relatively smooth under compactness is unsurprising. Compactness is very strong, and most regularity assumptions start to hold under this assumption due to continuity. However, in addition to this, the relative smoothness result requires both the log-likelihood and its gradient to be Lipschitz. The need for both at the same time is quite strong.
>
> We agree that compactness is a strong assumption, and that requiring both the log-likelihood and its gradient to be Lipschitz is restrictive. These conditions partly reflect the inherent complexity of NGVI under the relative smoothness framework. As illustrated in Appendix F, for univariate case, while it is possible to relax the compactness assumption on $\mu$, the necessary and sufficient conditions for relative smoothness involve higher-order (third and fourth order) derivatives of the log-probability, highlighting the technical difficulty of characterizing smoothness in this setting. We will explicitly address the limitations of these assumptions in the revision and emphasize that developing approaches to relax them is an important direction for future work.
>
> > It is unclear to me if Assumption 1 can be simply assumed away. In [1], Assumption 1 was explicitly established by studying special cases. However, that justification does not apply here since [1] did not study NGVI with Monte Carlo gradients. Given that the assumptions used in the paper are already strong enough, I would expect Assumption 1 to be verified explicitly.
>
> Thank you for raising this point, and we agree that Assumption 1 requires careful verification. In Appendix I, we have explicitly checked that Assumption 1 holds for logistic regression model. We will make this clarification more explicit in the revision.
>
> We did not find Assumption 1 in [1] and assume you meant to refer to some other paper. Could you please share the correct reference?
>
> > The experiments in Section 5 have a correctness issue.
>
> Thank you very much for catching this issue. You are correct that there was an error in the gradient computation for Prox-SGD. We have corrected this mistake and rerun all the experiments. The updated results on MNIST dataset (see below) show that Prox-SGD converges as expected and achieves performance comparable to Proj-SGD. We will include these corrected results in the revised version. Thank you again for pointing this out.
>
> We report the median negative ELBO after 1000 epochs (6000 iterations) across 5 runs with tuned step size; values in parentheses indicate the first and third quartiles. These results confirm that Prox-SGD converges with the corrected implementation and achieves performance comparable to Proj-SGD.
>
> |        | Prox-SGD                      | Proj-SGD                      | SNGD                          | Proj-SNGD                      |
> |:------:|:-----------------------------:|:-----------------------------:|:-----------------------------:|:------------------------------:|
> | $-\text{ELBO}$ | 691.07 (688.26, 691.75)        | 691.99 (689.44, 692.32)        | 696.27 (695.00, 696.91)        | 691.19 (690.95, 691.85)         |
>
> We also report the median number of iterations required for the loss to fall below 700 with tuned step size; values in parentheses indicate the first and third quartiles. The results below also show that Prox-SGD converges at a rate comparable to Proj-SGD. However, SNGD algorithms converge much faster than SGD algorithms.
>
> |               | Prox-SGD               | Proj-SGD               | SNGD                    | Proj-SNGD              |
> |:--------------|:----------------------:|:----------------------:|:-----------------------:|:-----------------------:|
> | **# Iterations** | 1905 (1843, 2050)       | 2121 (2109, 2169)       | 1087 (910, 1166)         | 764 (681, 783)          |
>
>
> > It is unclear why Assumption 2 is needed and what it implies. Why has this to be assumed when projections to a compact set are already being used?
>
> The key condition we need in the proof of Theorem 4.2 is $\omega^+=\omega_*$ (see Appendix H.3), where $$\omega^+\coloneqq\arg\min_{\omega' \in\tilde{\Omega}}\\{\langle\nabla\ell(\omega),\omega'\rangle+2\rho D_{A^\*}(\omega',\omega)\\},\quad \omega_*\coloneqq \arg\min_{\omega'\in\Omega}\\{\langle\nabla\ell(\omega),\omega'\rangle+2\rho D_{A^*}(\omega',\omega)\\}.$$When $\omega\in\partial \tilde\Omega$, this condition does not necessarily hold, so we impose Assumption 2 to avoid this scenario.
>
> However, it is possible to replace Assumption 2 with another algorithm-independent assumption. The intuition is that if the gradients $\nabla \ell(\omega)$ point towards the interior of $\tilde \Omega$ when $\omega\in\partial \tilde\Omega$, then for sufficiently large $\rho$, we have $\omega_*\in\tilde\Omega$, and hence $\omega^+=\omega_*$.
>
> Since $\nabla A(\eta)$ and $\nabla A^*(\omega)$ are inverse operators of one another (see Appendix C.1), (35) implies $$\omega_\*=(\nabla A^\*)^{-1}\left(\nabla A^\*(\omega)-\frac{1}{2\rho}\nabla\ell(\omega)\right)=\omega-\frac{1}{2\rho}\nabla^2 A(\eta)\nabla \ell(\omega)+o(1/\rho)=\omega-\frac{1}{2\rho}(\nabla^2 A^\*(\omega))^{-1}\nabla\ell(\omega)+o(1/\rho)$$as $\rho\to\infty$. Then we impose the following assumption.
>
> **Assumption 2'.** *For any $\omega\in\partial \tilde\Omega$, let $\mathbf{n}_\omega$ be the outward normal direction at $\omega$. Then it holds that$$\left<-(\nabla^2 A^\*(\omega))^{-1}\nabla\ell(\omega),\mathbf{n}_\omega\right><0.$$*
>
> Let $W\coloneqq\\{\omega_t:0\leq t\leq T\\}\cap \partial \tilde\Omega$ denote the set of iterates that lie on the boundary of $\tilde \Omega$. Note that $W$ is a finite set. If Assumption 2' is satisfied, there exists some $\varepsilon>0$ such that $$\left<-(\nabla^2 A^\*(\omega))^{-1}\nabla\ell(\omega),\mathbf{n}_\omega\right><-\varepsilon$$for all $\omega\in W$. Then we have $$\left<\mathbf{n}_\omega, \omega_*-\omega\right><-\frac{\varepsilon}{2\rho}+o(1/\rho)<0$$
>
> for some sufficiently large $\rho$. Hence $\omega_*\in\tilde\Omega$.
>
> In the following, we present an example where Assumption 2' holds. We consider a univariate Bayesian linear regression model with a single data point $(x, y)$, where $x,y,z\in \mathbb{R},p(z)=\mathcal{N}(0, 1)$ and $p(y|x,z)=\mathcal{N}(xz, 1)$. It can be shown that Assumption 2' is satisfied if the following set of inequalities holds:$$-U(x^2+1)<xy<U(x^2+1),\text{ and }D^{-1}<x^2+1<D.$$Therefore, by carefully choosing $U$ and $D$, Assumption 2' can be satisfied.
>
> ### Major Comments
>
> > The title is too non-descriptive. It is not obvious that this paper is about a theoretical analysis of the convergence of NGVI.
>
> We will revise our title to a more descriptive one reflecting the main theoretical contribution of the paper, e.g., ``Natural Gradient VI: Guarantees for Non-Conjugate Models''.
>
> > Discuss the assumptions in the abstract, especially since they are strong. The fact that the domain is enforced to be bounded and both the log-likelihood and its gradient need to be Lipschitz continuous should be stated. Citing the parametrization considered in the paper would also be useful.
>
> Thank you for this suggestion. We will revisit the abstract mentioning the key assumptions and the parameterization considered in the paper.
>
> > What is also not clear to me is why the paper focuses on joint likelihoods with an explicit prior-likelihood pair. Since the paper considers the non-conjugate setting, it should suffice to just consider a generic unnormalized target, which should simplify the setup and the notation. Similarly, the regularity results like Theorem 3.2 and Proposition 2,3 assume regularity of the likelihood $\log p(\mathcal{D}|z)$. This doesn't seem to be necessary. Why not just assume the regularity of the joint? (and equivalently, the "target unnormalized density")
>
> Thank you for this observation. We adopt the explicit prior–likelihood decomposition in order to leverage the structure of the KL term: by isolating $D_{\mathrm{KL}}(q(z)\|p(z))$ we can directly verify that this term is 1-1 relative smooth with respect to $A^*$ (see line 166). The remaining term is the expected negative log-likelihood $-\mathbb{E}_q[\log p(\mathcal{D}|z)]$, for which we establish separate regularity assumptions.
>
> > Line 211: Isn't this about "strong" convexity, not just convexity? If it is the case, this should be made more precise.
>
> Yes, you are correct. In this case, $\mu_H>0$, which corresponds to strong convexity rather than mere convexity.
>
> > Proposition 2: I think the constant $\mu_C$ can easily be improved. If the diagonal of a triangular matrix is strictly positive, then the diagonal elements correspond to its eigenvalues. Also, the eigenvalues coincide with the singular values. Then I think, for all $i$, $\sigma_{\mathrm{max}}(G_i)\leq\max\left(1,2C_{11},\ldots,2C_{dd}\right)\leq1+2\max_{j}C_{jj}\leq 1+2\sqrt{D}$ holds?
>
>
> Thank you for your thoughtful attempt to improve the bound. However, the argument does not apply in this case because $G_i$ is not symmetric, so its eigenvalues do not necessarily coincide with its singular values. The singular values must therefore be computed directly from the definition.
>
> ---
>
> We also thank the reviewer for the detailed suggestions on citations, terminology and presentation in minor comments. We have already included additional related work in Appendix B, where more papers on NGVI and mirror descent are cited.
>
> We hope our responses have addressed your concerns, and we will incorporate your constructive and thoughtful remarks in the revision, which we believe will greatly improve the overall quality of our work.
>
> ### References
>
> [1] David Wingate and Theophane Weber. Automated variational inference in probabilistic programming. arXiv preprint arXiv:1301.1299, 2013.

---

> > ### Comment · Reviewer_tPUM · 2025-08-01
> > **Response**
> >
> > I sincerely thank the authors for their clarifications. Given that my major concerns are resolved, I am happy to raise my score.
> >
> > > Thank you for this observation. We adopt the explicit prior–likelihood decomposition in order to leverage the structure of the KL term: by isolating $D_{\mathrm{KL}}(q(z)|p(z))$ we can directly verify that this term is 1-1 relative smooth with respect to $A^*$ (see line 166). The remaining term is the expected negative log-likelihood $-\mathbb{E}_q[\log p(\mathcal{D}|z)]$, for which we establish separate regularity assumptions.
> >
> > If this is the case, I would recommend using a simpler notation throughout the paper, and only use the prior-likelihood decomposition as a special case whenever needed. Especially given the notations in the paper are pretty dense already.
> >
> > > Thank you for your thoughtful attempt to improve the bound. However, the argument does not apply in this case because
> >  is not symmetric, so its eigenvalues do not necessarily coincide with its singular values. The singular values must therefore be computed directly from the definition.
> >
> > Yes, in general it is true that if the matrix is not symmetric, the eigenvalues may not coincide with the singular values. However, we have here a special case where they do: Again, $G_i$ is a *triangular matrix*. These are special because the diagonal entries exactly correspond to the unordered eigenvalues [Exercise 9, Ch 5., p. 258, 1]. Furthermore, since all of the eigenvalues are positive by design, the singular values coincide with the eigenvalues. We literally know *all* of the eigenvalues. Therefore, we obviously know all of the singular values. Does this make sense?
> >
> > 1. Friedberg, S. H., Insel, A. J., & Spence, L. E. (2002). Linear algebra. 4ed. Prentice Hall.

---

> > > ### Author Response · Authors · 2025-08-02
> > >
> > > Thank you for the constructive discussion.
> > >
> > > > If this is the case, I would recommend using a simpler notation throughout the paper, and only use the prior-likelihood decomposition as a special case whenever needed. Especially given the notations in the paper are pretty dense already.
> > >
> > > Thank you for the suggestion. We appreciate your concern about notation complexity and will consider simplifying the notation in the revision.
> > >
> > > > Furthermore, since all of the eigenvalues are positive by design, the singular values coincide with the eigenvalues. We literally know all of the eigenvalues.
> > >
> > > For a triangular matrix $G_i$, it is true that the eigenvalues of $G_i$ coincide with its diagonal entries. However, the absolute value of eigenvalues of $G_i$ do not necessarily match the singular values. Below is a counterexample:
> > >
> > > Let $$A=\begin{pmatrix}
> > >     1&0\\\\1&1
> > > \end{pmatrix}.$$The eigenvalues of $A$ are $\lambda_1(A)=\lambda_2(A)=1$. By definition, the singular values of $A$ are defined as the square root of eigenvalues of $A^\top A$. Since $$A^\top A=\begin{pmatrix}
> > >     2&1\\\\1&1
> > > \end{pmatrix},$$we have $\lambda_1(A^\top A)=\frac{3+\sqrt{5}}{2}, \lambda_2(A^\top A)=\frac{3-\sqrt{5}}{2}$. Thus, the singular values are $\sigma_{1,2}(A)=\sqrt{\frac{3\pm \sqrt{5}}{2}}\neq \lambda_{1,2}(A)$.

---

> > > > ### Comment · Reviewer_tPUM · 2025-08-02
> > > > **Response**
> > > >
> > > > Ah yes you are correct. I overlooked that part. Sorry for the confusion!

---

### Official Review · Reviewer_nVxV · 2025-06-27

**Clarity:** 3
**Significance:** 3
**Originality:** 3
**Rating:** 5
**Confidence:** 3

**Summary:**

The paper investigates the convergence properties of non-conjugate stochastic gradient descent on the variational inference ELBO leveraging recent work on the convergence of stochastic natural gradient descent (SNGD) for conjugate models but presently advanced to the non-conjugate setting. Notably, the proposed procedure includes a projection step (akin to how similarly a projection step of Proj-SGD is explored in [DGG23]) forming the proj-SNGD relying on theoretically derived results wrt. the smoothness properties of the associated ELBO and explores recent results from stochastic mirror descent of [FH24] to establish a convergence theory of the proposed proj-SNGD of O(1/sqrt(T)) with enhanced convergence of O(1/T) for concave log-likelihood functions. Notably, the manuscript also establishes non-trivial sufficient conditions for the relative smoothness of the ELBO function and properties wrt. hidden convexity (leveraging [FHH23]) in the present context for log-concave likelihood functions. On one synthetic and real dataset proj-SNGD is explored in practice and found to perform well when compared to standard SGD and SNGD notably with enhanced convergence and robustness to initial step size for the proj-SNGD procedure.

**Questions:**

Why are there no error bars on Figure 2 – the results appear somewhat anecdotal in the current presentation? -  it would be important here to see how the convergence behavior is across many initializations.

The experimentation is rather limited considering one synthetic and one real dataset. The paper would improve by establishing the convergence properties on more problems to highlight that the observed benefits especially of proj-SNGD when compared to SNGD can be observed in general. The gains are interesting but with the limited experimentation it could be established more convincingly by considering more problems than presently given in the main paper and small additional experimentation on MNIST given in the supplementary material.

It would further be interesting to increase the problem domains systematic from very smooth to less smooth and probe the strengths and limitations of the derived theoretical results relying on smoothness and projections as well as provide a general guidance on tuning U and D in the projection procedure as briefly investigated in the supplementary.

**Ethical Concerns:**

["NO or VERY MINOR ethics concerns only"]

**Final Justification:**

I find that the authors' responses in their rebuttal sufficiently clarifies and address my concerns. I appreciate the additional experimentation provided including the investigation of influence of initialization and inclusion of experimentation on additional data sets as requested. In light of this and the authors' rebuttals and answers to the concerns of the other reviewers I maintain my positive score.

**Limitations:**

Yes, the line of research is foundational for inference in the non-conjugate VI setting and positive and negative societal impacts are not so relevant to discuss in this work.

**Paper Formatting Concerns:**

No concerns in regards to the paper formatting.

**Quality:**

3

**Strengths And Weaknesses:**

Strenghs
* The paper is well-positioned in the literature
* The approach is novel and with a strong technical depth.
*  The problem considered is interesting and important providing theoretical results in the typical non-conjugate setting of VI which is a strongpoint.
* The theoretical contribution is non-trivial and explores recent results in the literature and combines several interesting insights wrt. the relative smoothness of the ELBO and convergence of the proposed proj-SNGD procedure.
* The limited experimentation supports the theoretical results and practical use of the proposed procedure.
* The established theory is in depth described and derived with an extensive supplementary material providing necessary details.

Weaknesses
* The main weakness of the paper is the experimentation which is limited and could be more convincing (see questions).


Quality:
The manuscript is well written, the results derived non-trivial, but the experimentation is limited and could be strengthened by considering more problems and datasets to highlight the general strength of the proposed framework which will also increase the impact and make the results more convincing.


Clarity:
The paper is generally well written and clear although the material covered extensive and with a high technical depth. There are a few minor typos such as

The realative smoothness -> the relative smoothness

I further found the example below lemma 3.1. unclear. Can you please clarify the steps in the equalities between lines 176 and 177. I do not understand how the right hand side is obtained from the expressions in the middle.

On line 126 please establish the PL inequality when first used and describe what PL abbreviates to make the manuscript more self-contained.


Significance:
The proposed framework establishes convergence properties of non-conjugate VI inference based on stochastic natural gradient descent which is an important and non-trivial contribution to the field. The experimentation is rather limited and the gains of the projection step wrt. convergence properties when comparing the proposed proj-SNGD to the SNGD appears to improve upon convergence and robustness but the gains are somewhat limited and it would be interesting to see if there are cases where substantial gains can be achieved by the proposed projection step and how general these gains can be observed across models and datasets.


Originality:
The manuscript provides new and original results and non-trivial theorems establishing smoothness and convergence properties of SNGD in the non-conjugate setting which can warrant publication.

---

> ### Author Rebuttal · Authors · 2025-07-31
>
> We sincerely appreciate the reviewer’s careful evaluation and insightful feedback. Below, we provide detailed responses to the identified weaknesses and questions.
>
> ### Weaknesses
>
> > There are a few minor typos such as The realative smoothness -> the relative smoothness
>
> Thank you for pointing this out. We will correct these typos.
>
> > Can you please clarify the steps in the equalities between lines 176 and 177. I do not understand how the right hand side is obtained from the expressions in the middle.
>
> Since $\mu=\xi,\sigma^2=\Xi-\xi^2$, the chain rule yields $\frac{\partial \mu}{\partial \xi}=1,\frac{\partial \sigma^2}{\partial \xi}=-2\xi, \frac{\partial \mu}{\partial \Xi}=0$ and $\frac{\partial \sigma^2}{\partial \Xi}=1$. Applying Lemma 3.1 together with the results above gives the formulas shown between lines 176 and 177.
>
> > On line 126 please establish the PL inequality when first used and describe what PL abbreviates to make the manuscript more self-contained.
>
> Thank you for the suggestion. The abbreviation "PL" first appeared in line 216. We introduced the full term “Polyak–Lojasiewicz” in line 152, and in the revision we will make it clear at this point that we will use the abbreviation “PL.”
>
> ### Questions
>
> > Why are there no error bars on Figure 2 – the results appear somewhat anecdotal in the current presentation? - it would be important here to see how the convergence behavior is across many initializations.
>
> In Figure 2, Poisson regression admits closed-form gradients (see line 1005 in Appendix E). Therefore, for a fixed initialization, the setup is deterministic, and error bars are not necessary.
>
> To examine the convergence behavior under different initializations, we conduct an additional experiment, where $\mu_0\sim \mathrm{Unif}([-3, 0])$ is sampled 10 times, and we compare the performance of SNGD and Proj-SNGD with different step sizes. We report the median number of iterations required for the function value gap to fall below $2\times 10^{-4}$; values in parentheses indicate the first and third quartiles. The results indicate that projection still accelerates convergence under different initializations. We will include these new results in the revised version of the paper.
>
> |            | Step size = 0.5      | Step size = 0.3      | Step size = 0.1       |
> |:----------:|:----------------------:|:----------------------:|:----------------------:|
> | **SNGD**   | 27 (20.5, 30)         | 33 (26.5, 35.5)       | 47 (47, 47.75)        |
> | **Proj-SNGD** | 9 (9, 9)            | 11 (11, 11)           | 47 (47, 47.75)        |
>
> For $\mu_0>0$, this effect is less pronounced, as SNGD exhibits more stable behavior in this regime.
>
> > The experimentation is rather limited considering one synthetic and one real dataset. The paper would improve by establishing the convergence properties on more problems to highlight that the observed benefits especially of proj-SNGD when compared to SNGD can be observed in general. The gains are interesting but with the limited experimentation it could be established more convincingly by considering more problems than presently given in the main paper and small additional experimentation on MNIST given in the supplementary material.
>
> We have conducted additional experiments on new datasets to further highlight the advantages of Proj-SNGD over SNGD.
>
> 1. Madelon dataset [1] ($n=2600,d=500$). For SNGD and Proj-SNGD algorithms with different choices of $U$ and $D$, we report the median number of iterations required for the loss to fall below 3000 (denoted as “NA” if not achieved in 2000 iterations); values in parentheses indicate the first and third quartiles (similar to Figure 4).
>
> |                      | $\gamma_0=0.1$           | $\gamma_0=0.2$           | $\gamma_0=0.5$           |
> |:--------------------:|:-------------------------:|:-------------------------:|:-------------------------:|
> | $U=\infty,D=\infty$ | 1314 (1224, 1358)        | 530 (528, 546)           | 250 (228, 264)           |
> | $U=3,D=400$         | 914 (852, 956)           | 320 (318, 328)           | 108 (108, 108)           |
> | $U=2,D=300$         | 766 (756, 812)           | 270 (262, 270)           | 100 (98, 106)            |
> | $U=1.5,D=200$       | NA                       | NA                        | NA                        |
>
> 2. A subset of the CIFAR-10 dataset [2] containing images of cats and dogs ($n=10000,d=3072$). Similarly, we report the median number of iterations required for the loss to fall below 35000 (denoted as “NA” if not achieved in 15000 iterations); values in parentheses indicate the first and third quartiles.
>
> |                      | $\gamma_0 = 5 \times 10^{-4}$    | $\gamma_0 = 10^{-3}$          |
> |:--------------------:|:--------------------------------:|:-----------------------------:|
> | $U = \infty, D = \infty$ | NA                             | NA                            |
> | $U = 4, D = 20$          | 12206 (12138, 12528)            | 9127 (9073, 9177)             |
> | $U = 3, D = 15$          | 11703 (11522, 12007)            | 9253 (9039, 9387)             |
> | $U = 2, D = 10$          | NA                               | NA                             |
>
> The results above suggest that, with properly chosen hyperparameters $U$ and $D$, Proj-NGVI can outperform NGVI.
>
> > It would further be interesting to increase the problem domains systematic from very smooth to less smooth and probe the strengths and limitations of the derived theoretical results relying on smoothness and projections as well as provide a general guidance on tuning U and D in the projection procedure as briefly investigated in the supplementary.
>
> Thank you for raising this practical point. An illustrative experiment on the effect of the domain for the MNIST dataset is presented in Figure 4, Appendix J.2. Based on this, a broad guideline for the choice of $U$ and $D$—intended as intuition rather than a strict prescription—is as follows:
>
> (1) $U$ and $D$ should not be too small, otherwise the optimal solution may fall outside $\tilde \Omega$. For instance, in the left panel of Figure 4, Proj-SNGD with $U=2,D=10$ converges to a suboptimal solution.
>
> (2) $U$ and $D$ should not be too large, otherwise the algorithm will not benefit from projection.
>
> In our experiments, we run a small number of iterations with several candidate values of $U$ and $D$, and select the sets of parameters that lead to faster convergence.
>
> ---
>
> We appreciate the reviewer for thoughtful remarks and suggestions, which we believe will improve the overall quality of the manuscript.
>
> ### References
>
> [1] Isabelle Guyon. Madelon. UCI Machine Learning Repository, 2004. DOI:https://doi.org/10.24432/C5602H.
>
> [2] Alex Krizhevsky. Learning multiple layers of features from tiny images. Technical report, 2009.

---

### Official Review · Reviewer_Kcs6 · 2025-06-29

**Clarity:** 3
**Significance:** 3
**Originality:** 2
**Rating:** 4
**Confidence:** 3

**Summary:**

The paper studies on theoretical aspects of stochastic natural gradient variational inference. It connects NGVI with the stochastic mirror descent and advance theoretical understanding in the case of non-conjugate likelihood settings and provide non-asymptotic error bounds.

**Questions:**

1. While the finite error bounds in Theorem 4.1 and Theorem 4.2 are established, they are conditional upon the choice of the step size. The current theorem only demonstrates the error bound for some step size (not necessarily any step size schedule). Can you further elaborate on influence of the step size choice in error bounds and their rate? Will it hold regardless of how large the initial step size is chosen?

2. In the projection step introduced in the Section 4.1., one needs to calibrate the projection radius. How should practitioners decide projection radiuses? Can you perhaps elaborate more on the influence of the choice of projection radiuses? I understand that the authors have stated moderate values of U and D work empirically, but it's not clear what they mean by moderate values. Perhaps more extensive numerical experiments would be beneficial to justify the projection step.

**Ethical Concerns:**

["NO or VERY MINOR ethics concerns only"]

**Limitations:**

Not Applicable.

**Paper Formatting Concerns:**

No major formatting issue.

**Quality:**

3

**Strengths And Weaknesses:**

Strengths: The paper provides a solid theoretical understanding of NGVI, expanding the existing works to the setting where the variational loss can be non-convex. Characterizing a hidden convexity properties of the variational loss, it extends theoretical insights on the convergence of NGVI. The organization of the paper is well-structured with clear exposition.

Weaknesses: No major significances, while I found some incomplete sentences to be edited, e.g., page 2, line 72, "In the Euclidean set up, ..." and page 5 line 167, "the Hessian of the two functions coincide".

Also, I suggest providing more references behind characterizations they are using. For instance, the duality of the natural parameter in the exponential family and the dual expectation parameter are stated without providing further references.

---

> ### Author Rebuttal · Authors · 2025-07-31
>
> We thank the reviewer for the thoughtful and helpful feedback. We will address each of the weaknesses and questions in detail.
>
> ### Weaknesses
>
> > I found some incomplete sentences to be edited, e.g., page 2, line 72, "In the Euclidean set up, ..." and page 5 line 167, "the Hessian of the two functions coincide".
>
> Thank you for pointing this out. We will revise these sentences in the revision.
>
> > Also, I suggest providing more references behind characterizations they are using. For instance, the duality of the natural parameter in the exponential family and the dual expectation parameter are stated without providing further references.
>
> Thank you for your suggestion. Most of the content in Section 2.1 is based on [1], which provides detailed discussion of the duality between the natural and expectation parameters in exponential families. We will revise the paper to make this clearer.
>
> ### Questions
>
> > While the finite error bounds in Theorem 4.1 and Theorem 4.2 are established, they are conditional upon the choice of the step size. The current theorem only demonstrates the error bound for some step size (not necessarily any step size schedule). Can you further elaborate on influence of the step size choice in error bounds and their rate? Will it hold regardless of how large the initial step size is chosen?
>
> Our theorems use specific step size schemes, which are standard in non-convex optimization literature. For Theorem 4.1, the step-size is constant and is provided in line 1251, Appendix H.2: $$\gamma_t=\gamma=\min\left\\{\frac{1}{2L},\sqrt{\frac{\lambda_0}{V^2LT}}\right\\}.$$ For Theorem 4.2, we follow the same step size scheme as Theorem 4.7 in [2], which we did not explicitly mention in the paper:$$\gamma_t=\begin{cases}
>     \frac{1}{2L},&\text{if }t\leq T/2\text{ and }T\leq\frac{6L}{\mu_B},\\\\
>     \frac{6}{\mu_B(t-\lceil T/2\rceil)+12L}, &\text{otherwise}.
> \end{cases}$$ In particular, it is important in both cases that $\gamma_t \leq 1/(2L)$ to avoid divergence. We acknowledge that this could cause confusion and will revise the main text to clearly state the step size schemes.
>
> > In the projection step introduced in the Section 4.1., one needs to calibrate the projection radius. How should practitioners decide projection radiuses? Can you perhaps elaborate more on the influence of the choice of projection radiuses? I understand that the authors have stated moderate values of $U$ and $D$ work empirically, but it's not clear what they mean by moderate values.
>
> By "moderate values", we mean that:
>
> 1. $U$ and $D$ should not be too small, otherwise the optimal solution may fall outside $\tilde \Omega$. For instance, in the left panel of Figure 4 on page 52, Proj-SNGD with $U=2,D=10$ converges to a suboptimal solution.
>
> 2. $U$ and $D$ should not be too large, otherwise the algorithm will not benefit from projection.
>
> In our experiments (see Appendix J.2), we run a small number of iterations with several candidate values of $U$ and $D$, and select the sets of parameters that lead to faster convergence.
>
> > Perhaps more extensive numerical experiments would be beneficial to justify the projection step.
>
> We have conducted additional experiments on new datasets to further justify the projection step.
>
> 1. Madelon dataset [3] ($n=2600,d=500$). For SNGD and Proj-SNGD algorithms with different choices of $U$ and $D$, we report the median number of iterations required for the loss to fall below 3000 (denoted as “NA” if not achieved in 2000 iterations); values in parentheses indicate the first and third quartiles (similar to Figure 4).
>
> |                      | $\gamma_0=0.1$           | $\gamma_0=0.2$           | $\gamma_0=0.5$           |
> |:--------------------:|:-------------------------:|:-------------------------:|:-------------------------:|
> | $U=\infty,D=\infty$ | 1314 (1224, 1358)        | 530 (528, 546)           | 250 (228, 264)           |
> | $U=3,D=400$         | 914 (852, 956)           | 320 (318, 328)           | 108 (108, 108)           |
> | $U=2,D=300$         | 766 (756, 812)           | 270 (262, 270)           | 100 (98, 106)            |
> | $U=1.5,D=200$       | NA                       | NA                        | NA                        |
>
> 2. A subset of the CIFAR-10 dataset [4] containing images of cats and dogs ($n=10000,d=3072$). Similarly, we report the median number of iterations required for the loss to fall below 35000 (denoted as “NA” if not achieved in 15000 iterations); values in parentheses indicate the first and third quartiles.
>
> |                      | $\gamma_0 = 5 \times 10^{-4}$    | $\gamma_0 = 10^{-3}$          |
> |:--------------------:|:--------------------------------:|:-----------------------------:|
> | $U = \infty, D = \infty$ | NA                             | NA                            |
> | $U = 4, D = 20$          | 12206 (12138, 12528)            | 9127 (9073, 9177)             |
> | $U = 3, D = 15$          | 11703 (11522, 12007)            | 9253 (9039, 9387)             |
> | $U = 2, D = 10$          | NA                               | NA                             |
>
> The results above suggest that projection with appropriately chosen hyperparameters can accelerate the convergence of NGVI.
>
> ---
>
> Thank you again for the insightful questions and helpful suggestions. We hope our responses have addressed your concerns, and we will revise the manuscript accordingly to improve the clarity and completeness of our manuscript.
>
> ### References
>
> [1] Martin J Wainwright, Michael I Jordan, et al. Graphical models, exponential families, and variational inference. Foundations and Trends in Machine Learning, 1(1–2):1-305, 2008.
>
> [2] Ilyas Fatkhullin and Niao He. Taming nonconvex stochastic mirror descent with general bregman divergence. In International Conference on Artificial Intelligence and Statistics, pages 3493–3501. PMLR, 2024.
>
> [3] Isabelle Guyon. Madelon. UCI Machine Learning Repository, 2004. DOI:https://doi.org/10.24432/C5602H.
>
> [4] Alex Krizhevsky. Learning multiple layers of features from tiny images. Technical report, 2009.

---

> > ### Comment · Reviewer_Kcs6 · 2025-08-07
> > **official comment**
> >
> > Thank you for addressing all my concerns and questions in detail. I appreciate authors' effort in further numerical experiments, but I am a bit concerned that its sensitivity to the choice of radius is more substantial than I expected. Why does it get NA in CIFAR-10 case, for U = \infty and D = \infty? My understanding is that for small values, NA means you cannot get closer to the true optimum? Can you clarify these?

---

> > > ### Author Response · Authors · 2025-08-08
> > >
> > > Thank you for the question.
> > >
> > > An “NA” indicates that the loss does not reach the threshold in 15000 iterations. For SNGD without projection, the “NA” occurs because convergence is too slow; however, with more iterations, it would eventually reach the true optimum. To show this, we rerun the CIFAR-10 experiment for 5000 epochs (25000 iterations) with a different random seed, and we report the median number of iterations required for the loss to fall below 35000 (denoted as “NA” if not achieved in 25000 iterations)
> > >
> > > |                      | $\gamma_0 = 5\times 10^{-4}$      | $\gamma_0 = 10^{-3}$       |
> > > |----------------------|-----------------------------------|----------------------------|
> > > | $U=\infty, D=\infty$ | 17633 (17188, 18007)              | 16937 (16699, 16957)       |
> > > | $U=4, D=20$          | 12131 (11997, 12347)              | 9328 (8938, 9363)          |
> > > | $U=3, D=15$          | 11563 (11487, 11634)              | 9418 (9037, 9657)          |
> > > | $U=2, D=10$          | NA                                | NA                         |
> > >
> > > This suggests that SNGD without projection can still reach the true optimum if trained for slightly longer.
> > >
> > > The slow convergence of SNGD without projection is due to the poor tolerance for large step sizes, which causes divergence in the initial phase. In contrast, projection stabilizes this in the initial phase and accelerates convergence (see the Poisson regression example in Figure 2). This effect is consistently observed across 3 different datasets. Moreover, this effect is consistent with our theoretical upper bounds for the relative smoothness parameter. This is because our upper bound on $\ell$ depends on the diameter of the compact set, indicating that relative smoothness may not hold globally, which leads to divergence behavior in the initial phase.

---

> > > > ### Comment · Reviewer_Kcs6 · 2025-08-08
> > > > **official comment**
> > > >
> > > > Thanks for addressing my questions. While I find the methodology interesting, performance seems to depend quite a lot on the choice of projection radius. And it seems a systematic approach for estimating (even a crude estimate) the projection radius would further strengthen the methodology. Therefore, I would like to retain my original score.

---

### Official Review · Reviewer_icec · 2025-07-01

**Clarity:** 3
**Significance:** 3
**Originality:** 2
**Rating:** 4
**Confidence:** 4

**Summary:**

The paper proposes a theoretical study of Stochastic Natural Gradient Descent in the particular setting of Variational Inference, assuming a Gaussian prior and mean-field Gaussian variational family. The study begins by establishing some properties of the loss landscape (coerciveness, relative smoothness and hidden convexity). Convergence guarantees are then provided under fairly strong assumptions.

**Questions:**

* Lines 284-286: “empirical results indicate that […] even if Assumption 2 does not hold, the performance of Proj-SNGD is no worse and sometimes better than SGD.” Can you expand on this?
* Is the big O characterization of  $L$ given by Theorem 3.2 practically relevant?
* I don’t really understand how $(35)$ is obtained. What are the steps that lead to this equation?
* In the Poisson regression experiment, what is the function value gap?
* What is the difference between Figures 1 and 2? Does projecting achieve the same effect as changing the initialization in this example?

**Ethical Concerns:**

["NO or VERY MINOR ethics concerns only"]

**Final Justification:**

This paper develops theoretical results on NGVI in non-conjugate settings, assuming a Gaussian prior and variational family. While some theoretical arguments are interesting and insightful, strong assumptions besides Gaussianity restrict the applicability of the guarantees obtained. The experiments conducted by the authors during the rebuttal period convincingly justify their method. However, this is mainly a theoretical paper, which is why I decided to keep my original score of 4 - Borderline accept.

**Limitations:**

Yes.

**Paper Formatting Concerns:**

None.

**Quality:**

3

**Strengths And Weaknesses:**

**Strengths**

The results overall are quite insightful, and the authors use a wide range of concepts and techniques/previously established results to support their analysis and prove their claims. The loss landscape analysis is of particular interest.

**Weaknesses**
* The main limitation, as emphasized by the authors, is Assumption 2. Unlike Assumptions 1 and 3, it cannot be checked directly and is susceptible to not being satisfied even in the restricted setting considered in the paper.
* While the paper is well-structured, its clarity could be improved. This is partly because each result relies on its own set of technical assumptions. A dedicated section outlining all the assumptions would significantly improve readability and be a valuable addition. It would be helpful to have more than just the two (three, counting the appendices) assumptions explicitly stated in the paper.
* The results are limited to specific problems in a purely Gaussian setting. They do not seem to be easily generalizable to other settings, as a significant part of the arguments rely on the properties of Gaussian distributions.
* There is no explicit construction of the step-size sequence.

**Additional remarks**
* Given how much effort went into writing the appendices, a summary of the results already established and used in the proofs (when the proof consists in checking the application conditions of some theorem) would have been a nice touch.
* In Section 2.2, $I$ denotes both the Fisher Information Matrix and the identity matrix.
* Line 864: “1-relatively smoothness” -> 1-relative smoothness
* Line 960: “the objective is coerciveness […] listed above” -> the objective is coercive […] listed below.

---

> ### Author Rebuttal · Authors · 2025-07-30
>
> We thank the reviewer for the careful reading and valuable feedback. Below, we address each of the weaknesses and questions in detail.
>
> ### Weaknesses
>
> > The main limitation, as emphasized by the authors, is Assumption 2. Unlike Assumptions 1 and 3, it cannot be checked directly and is susceptible to not being satisfied even in the restricted setting considered in the paper.
>
> We agree that Assumption 2 is primarily a technical condition and can be challenging to verify beforehand. The key condition we need in the proof of Theorem 4.2 is $\omega^+=\omega_*$ (see Appendix H.3), where $$\omega^+\coloneqq\arg\min_{\omega' \in\tilde{\Omega}}\\{\langle\nabla\ell(\omega),\omega'\rangle+2\rho D_{A^\*}(\omega',\omega)\\},\quad \omega_*\coloneqq \arg\min_{\omega'\in\Omega}\\{\langle\nabla\ell(\omega),\omega'\rangle+2\rho D_{A^*}(\omega',\omega)\\}.$$When $\omega\in\partial \tilde\Omega$, this condition does not necessarily hold, so we impose Assumption 2 to avoid this scenario.
>
> However, it is possible to replace Assumption 2 with another algorithm-independent assumption. The intuition is that if the gradients $\nabla \ell(\omega)$ point towards the interior of $\tilde \Omega$ when $\omega\in\partial \tilde\Omega$, then for sufficiently large $\rho$, we have $\omega_*\in\tilde\Omega$, and hence $\omega^+=\omega_*$.
>
> Since $\nabla A(\eta)$ and $\nabla A^*(\omega)$ are inverse operators of one another (see Appendix C.1), (35) implies $$\omega_\*=(\nabla A^\*)^{-1}\left(\nabla A^\*(\omega)-\frac{1}{2\rho}\nabla\ell(\omega)\right)=\omega-\frac{1}{2\rho}\nabla^2 A(\eta)\nabla \ell(\omega)+o(1/\rho)=\omega-\frac{1}{2\rho}(\nabla^2 A^\*(\omega))^{-1}\nabla\ell(\omega)+o(1/\rho)$$as $\rho\to\infty$. Then we impose the following assumption.
>
> **Assumption 2'.** *For any $\omega\in\partial \tilde\Omega$, let $\mathbf{n}_\omega$ be the outward normal direction at $\omega$. Assume that$$\left<-(\nabla^2 A^\*(\omega))^{-1}\nabla\ell(\omega),\mathbf{n}_\omega\right><0.$$*
>
> Let $W\coloneqq\\{\omega_t:0\leq t\leq T\\}\cap \partial \tilde\Omega$ denote the set of iterates that lie on the boundary of $\tilde \Omega$. Note that $W$ is a finite set. If Assumption 2' is satisfied, there exists some $\varepsilon>0$ such that $$\left<-(\nabla^2 A^\*(\omega))^{-1}\nabla\ell(\omega),\mathbf{n}_\omega\right><-\varepsilon$$for all $\omega\in W$. Then we have $$\left<\mathbf{n}_\omega, \omega_*-\omega\right><-\frac{\varepsilon}{2\rho}+o(1/\rho)<0$$
>
> for some sufficiently large $\rho$. Hence $\omega_*\in\tilde\Omega$.
>
> In the following, we present an example where Assumption 2' holds. We consider a univariate Bayesian linear regression model with a single data point $(x, y)$, where $x,y,z\in \mathbb{R},p(z)=\mathcal{N}(0, 1)$ and $p(y|x,z)=\mathcal{N}(xz, 1)$. It can be shown that Assumption 2' is satisfied if the following set of inequalities holds:$$-U(x^2+1)<xy<U(x^2+1),\text{ and }D^{-1}<x^2+1<D.$$Therefore, by carefully choosing $U$ and $D$, Assumption 2' can be satisfied. We will include the derivation of this fact in the revision.
>
> > While the paper is well-structured, its clarity could be improved. This is partly because each result relies on its own set of technical assumptions. A dedicated section outlining all the assumptions would significantly improve readability and be a valuable addition. It would be helpful to have more than just the two (three, counting the appendices) assumptions explicitly stated in the paper.
>
> Thank you for your suggestion. We will add a list of assumptions and a summary of key external results used in the proofs, especially when the proof mainly involves verifying these conditions. This will make the paper easier to follow and more readable.
>
> > The results are limited to specific problems in a purely Gaussian setting. They do not seem to be easily generalizable to other settings, as a significant part of the arguments rely on the properties of Gaussian distributions.
>
> We acknowledge that our analysis relies on properties of the Gaussian family. We choose this setting due to its analytical simplicity and its widespread use in practice. This restriction is standard in the literature (e.g., [1,2]), as relaxing it would require fundamentally different techniques. We view extending the results beyond the Gaussian setting as an important direction for future work.
>
> > There is no explicit construction of the step-size sequence.
>
> Thank you for the comment. Our theorems use specific step size schemes, which are standard in non-convex optimization literature. For Theorem 4.1, the step-size is constant and is provided in line 1251, Appendix H.2: $$\gamma_t=\gamma=\min\left\\{\frac{1}{2L},\sqrt{\frac{\lambda_0}{V^2LT}}\right\\}.$$ For Theorem 4.2, we follow the same step size scheme as Theorem 4.7 in [3], which we did not explicitly mention in the paper:$$\gamma_t=\begin{cases}
>     \frac{1}{2L},&\text{if }t\leq T/2\text{ and }T\leq\frac{6L}{\mu_B},\\\\
>     \frac{6}{\mu_B(t-\lceil T/2\rceil)+12L}, &\text{otherwise}.
> \end{cases}$$ In particular, it is important in both cases that $\gamma_t \leq 1/(2L)$ to avoid divergence. We acknowledge that this could cause confusion and will revise the main text to clearly state the step size schemes.
>
> ### Questions
>
> > Lines 284-286: “empirical results indicate that […] even if Assumption 2 does not hold, the performance of Proj-SNGD is no worse and sometimes better than SGD.” Can you expand on this?
>
> In our experiments (Figures 2 and 3), Assumption 2 is not satisfied. This can be inferred from the different loss behaviors of SNGD and Proj-SNGD: such differences indicate that projection occurred during Proj-SNGD, and when projection occurs, the iterate must lie on the boundary of $\tilde\Omega$. We will make this explicit in the revision to avoid confusion. Despite this, Proj-SNGD converges faster and demonstrates greater robustness to step size tuning. These observations suggest that Proj-SNGD can outperform SGD algorithms even when Assumption 2 does not hold.
>
> > Is the big O characterization of $L$ given by Theorem 3.2 practically relevant?
>
> The Big-O characterization of $L$ in Theorem 3.2 is primarily of theoretical interest, as it provides a conservative estimate of the real smoothness parameter. It offers insight into how $L$ scales with problem parameters, including $d,U,D$ and the properties of the log likelihood function, which can guide step size selection in practice. In particular, the dependence on dimension $d$ is at most linear, and the dependence on $U$ and $D$ is polynomial of degree at most 3. This characterization is practically relevant because $L$ directly influences the convergence rate of Proj-SNGD, and implies the polynomial iteration complexity in these parameters.
>
> > I don’t really understand how (35) is obtained. What are the steps that lead to this equation?
>
> (35) is obtained by explicitly solving the optimization problem in line 1323. By definition, the Bregman divergence equals $D_{A^\*}(\omega',\omega)=A^\*(\omega')-A^\*(\omega)-\left<\nabla A^\*(\omega),\omega'-\omega\right>.$ Substituting it into the problem in line 1323 and applying the first-order optimality condition yields $\nabla\ell(\omega)+2\rho(\nabla A^\*(\omega')-\nabla A^\*(\omega))=0$, which coincides with (35). The second derivative $2\rho \nabla^2 A^\*(\omega')\succ 0$ because $A^\*$ is strictly convex.
>
> > In the Poisson regression experiment, what is the function value gap?
>
> Function value gap at iteration $t$ is the difference between the function value at iteration $t$ and the optimal value, i.e., $\ell(\omega_t)-\ell^*$.
>
> > What is the difference between Figures 1 and 2? Does projecting achieve the same effect as changing the initialization in this example?
>
> Figures 1 and 2 correspond to the same experimental setting. The left panel of Figure 1 is identical to the left panel of Figure 2. The right panel of Figure 1 illustrates the performance of SNGD under a different initialization, and the right panel of Figure 2 shows the performance of our proposed Proj-SNGD with the original initialization. Both projection and changing the initialization in this example can stabilize and accelerate the training process, but the training trajectories of the two are not identical. The main insight from this experiment is that the objective is less well-behaved in the nonconvex setting (Figure 1), and that our Proj-SNGD converges even with initializations for which original SNGD initially diverges (Figure 2).
>
> ---
>
> We also thank the reviewer for the helpful suggestions on clarity and presentation in the additional remarks, and we will incorporate these suggestions in the revision to improve readability. We believe that the reviewer’s feedback will help us improve the quality of the manuscript.
>
> ### References
>
> [1] Kaiwen Wu and Jacob R. Gardner. Understanding Stochastic Natural Gradient Variational Inference. In International Conference on Machine Learning, pages 53398-53421. PMLR, 2024.
>
> [2] Navish Kumar, Thomas Möllenhoff, Mohammad Emtiyaz Khan, and Aurelien Lucchi. Optimization guarantees for square-root natural-gradient variational inference. Transactions on Machine Learning Research, 2025.
>
> [3] Ilyas Fatkhullin and Niao He. Taming nonconvex stochastic mirror descent with general bregman divergence. In International Conference on Artificial Intelligence and Statistics, pages 3493–3501. PMLR, 2024.

---

### Note · Authors · 2025-08-14

We thank the reviewers for their particularly in-depth reviews and productive discussions. We truly appreciate this effort as we believe the discussions were very helpful to further improve our paper. Since some discussions came out quite lengthy and technical, we want to take this opportunity to summarize the key discussion points and improvements.

1. Encouraged by the reviewers' criticism of our Assumption 2, we reexamined the proof of our Proj-SNGD's fast convergence under hidden convexity and analyzed why Assumption 2 is needed. Since it is challenging to verify it a priori, we proposed an alternative, verifiable Assumption 2', see details below in the discussion. This new alternative assumption replaces our Assumption 2 and it is algorithm independent, which makes it possible to verify for a specific problem. In particular, we demonstrated that it holds in a Bayesian linear regression example. Following the reviewer's suggestion, we will replace our previous Assumption 2 with this new Assumption 2' in the next revision.

2. Reviewer tPUM noticed a subtle implementation issue in a baseline algorithm from the prior work (Prox-SGD). We fixed this issue and reran the experiments: the updated results confirm that its convergence behavior is consistent with theoretical expectations and is comparable to Proj-SGD, while our proposed Proj-SNGD still achieves faster convergence and greater robustness to step size tuning. We appreciate reviewer tPUM for catching this.

3. We conducted extra experiments with additional ablations of projection radius and added more variety of datasets, including larger CIFAR-10 dataset. We provide insightful illustrations of the effect of projection in accelerating and stabilizing NGVI, which experimentally confirms the need to restrict the diameter of the search space. The results are consistent across different datasets including large scale datasets.

In summary, our work analyzes the loss landscape of NGVI and provides the first convergence guarantees for non-convex NGVI, both with and without convexity of the negative log likelihood. We are glad that the reviewers appreciate that our work makes an important theoretical contribution to this challenging and important topic. We also support our findings by empirical results on multiple datasets, testing our algorithmic novelty in practice via ablation studies. We thank the reviewers one more time for their insightful comments, which helped to improve the quality of our work.

---

### Decision · Program_Chairs · 2025-09-17

**Decision:**

Accept (poster)

**Comment:**

This paper analyzes stochastic nature gradient descent in the context of variational inference. In particular, it considers the setting of non-conjugate models, where the loss is nonconvex. In this process, a modified NGVI algorithm is proposed that includes non-Euclidean projections, and the convergence of this method is established. The paper also provides an empirical study on regression problems.

Reviewers initially had mixed opinions on the paper. The main reason for this was because of the limited setting of the theoretical results: reviewers pointed out many assumptions were quite strong, e.g., relying on Gaussianity with additional restrictions, such as compactness and bounded derivatives. However, after the discussion phase with the authors, reviewers were more positive about accepting the paper, as the authors cleared up some questions about the assumptions and experiments; the authors also provided additional experiments.

During the reviewer discussion phase, the reviewers agreed that despite the limited applicability of the results, the paper provides valuable insights into stochastic NGDVI. Thus, I recommend accepting this paper. We encourage the authors to revise the paper according to the discussion with the reviewers.